# Spatial tumor immune heterogeneity facilitates subtype co-existence and therapy response in pancreatic cancer

Lukas Klein [1], Mengyu Tu[1], Niklas Krebs[2], Laura Urbach[1], Daniela Grimm [1], Muhammad Umair Latif [1], Frederike Penz[1], Anna Blandau[3], Xueyan Wu [1], Rebecca Diya Samuel [1], Stefan Küffer[4], Florian Wegwitz[3], Nathan Chan[5], Kazeera Aliar [5], Foram Vyas[5,6], Uday Kishore[7], Elisabeth Hessmann[1,8,9], Andreas Trumpp [10,11], Elisa Espinet [10,11,12,13], Argyris Papantonis [4,8,9], Rama Khokha[5,6], Volker Ellenrieder[1,8,9], Barbara T. Grünwald[5,14,15] & Shiv K. Singh [1,8,9,15] ✉

Pancreatic ductal adenocarcinoma (PDAC) displays a high degree of spatial subtype heterogeneity and co-existence, linked to a diverse microenvironment and worse clinical outcome. However, the underlying mechanisms remain unclear. Here, by combining preclinical models, multi-center clinical, transcriptomic, proteomic, and patient bioimaging data, we identify an interplay between neoplastic intrinsic AP1 transcription factor dichotomy and extrinsic macrophages driving subtype co-existence and an immunosuppressive microenvironment. ATAC-, ChIP-, and RNA-seq analyses reveal that JUNB/AP1- and HDAC-mediated epigenetic programs repress pro-inflammatory signatures in tumor cells, antagonizing cJUN/AP1 signaling, favoring a therapy-responsive classical neoplastic state. This dichotomous regulation is amplified via regional TNF-α+ macrophages, which associates with a reactive phenotype and reduced CD8+ T cell infiltration in patients. Consequently, combined preclinical anti-TNF-α immunotherapy and chemotherapy reduces macrophages and promotes CD3+/CD8+ T cell infiltration in basal-like PDAC, improving survival. Hence, tumor cell-intrinsic epigenetic programs, together with extrinsic microenvironmental cues, facilitate intratumoral subtype heterogeneity and disease progression.

The molecular heterogeneity in neoplastic and stromal immune cells renders PDAC prognosis dismal and therapy challenging. PDAC has become the third leading cause of cancer-related death with a 5-year survival rate of 12%[1]. Presently, gemcitabine/nab-paclitaxel and modified FOLFIRINOX are the main therapeutic options in PDAC, though therapy resistance and local as well as distal recurrences are common[2,3]. Transcriptome analyses identified two clinically relevant PDAC subtypes; the basal-like (BL) or squamous subtype is linked to therapy resistance and worse

patient outcome, whereas the classical (CLA) subtype shows better clinical outcome[4–6].

Subtype-based screening of a small cohort of PDAC patients has shown potential prognostic as well as predictive benefits[6–8], and hence, a number of current clinical studies (e.g., NCT05314998) are designed to translate these subtypes into the clinical setting[9]. However, it has become clear that the CLA and BL subtypes are not discrete states of individual PDAC tumors, but rather co-exist and contribute to significant intratumoral heterogeneity that is poorly understood. Multi-

scale transcriptomic and imaging-based profiling revealed a widespread co-existence of CLA and BL subtypes within PDAC patient tumors, as well as hybrid/co-expressor states that are implicated as transition mechanism between the subtype extremes[10–15], emphasizing the complex tumor cell plasticity. Moreover, the extent of subtype co-existence increases in advanced disease, especially in metastatic samples[12,13,16]. This significantly associates with poor overall prognosis, making precision-based therapies for PDAC patients a major challenge[13,16,17]. Thus, understanding the underlying mechanisms of intratumoral subtype plasticity may improve subtype-based prediction and therapeutic response for PDAC patients.

While the precise mechanisms that drive this spatial plasticity in PDAC unclear, it appears that both tumor cell-intrinsic and extrinsic factors play a role. A major extrinsic driver of malignant cell phenotypic plasticity in PDAC is the regional tumor immune microenvironment (TiME), which enables the acquisition of therapy resistance and aggressive behavior[14,18,19]. A recent study showed that TGF-β can promote BL subtype specificity and therapy resistance by regulating neoplastic cell-intrinsic transcriptional programs[20]. Furthermore, TNF-α, secreted by macrophages, and other cell types, along with signaling events mediated by IFN-α/γ, can drive the BL subtype-specific transcriptional program and promote PDAC aggressiveness[21–24]. Notably, this transcription-based subtype plasticity is independent of genetic alterations[20,21]. It is currently unknown whether plasticity imposed by microenvironmental cues or other factors is responsible for intratumoral subtype heterogeneity and how such factors might promote a specific cell-type identity and response to therapy.

PDAC neoplastic cells show a remarkable capacity to change their phenotypic identity through transcriptional and epigenetic regulation. Lineage-specific transcription factors (TFs) and epigenetic co-regulators are considered a key hallmark of PDAC subtype specificity and disease progression[4,21,25–28]. For instance, MYC, TP63, and AP1 are crucial TFs in squamous/BL and inflammatory PDAC subtypes, while TFs such as GATA6 drive the therapy-responsive CLA neoplastic identity[7,21,29–31]. Notably, the AP1 inflammatory TF family drives a strong response to external stimuli such as growth factors and cytokines and regulates key cellular processes including differentiation and growth, also in the context of tumor biology[32–34]. The JUN/AP1 factors furthermore exhibit substantial heterogeneity in gene expression in the distinct PDAC subtypes. For instance, while cJUN/AP1high PDAC patients exhibit resistance to gemcitabine/nab-paclitaxel chemotherapy, earlier relapse, and a BL phenotypic state[21,35], JUNB/AP1 expression is linked to low-grade/CLA-like PDAC[16,21,25,36,37]. This study thus addresses the question whether and how AP1 heterogeneity could regulate intratumoral subtype plasticity, inflammatory programs, and therapy response in PDAC.

We report a spatially regulated dichotomy in the AP1 transcriptional programs (JUNB vs. cJUN) in PDAC subtype plasticity via both tumor cell intrinsic and extrinsic mechanisms. We show that JUNB/AP1- and HDAC-mediated epigenetic and transcriptional networks restrict macrophage infiltration in the TME. These spatial CD68+/TNF-α+ macrophages promote intratumoral subtype co-existence by destabilizing CLA-like epithelial and promoting a BL phenotypic state. Mechanistically, the loss of JUNB-mediated repressive functions is linked to TNF-α signaling, CLA-to-BL neoplastic transition, and poorer outcomes in PDAC patients. Notably, macrophage-derived TNF-αhigh expression marked a reactive stroma with low CD3+/CD8+ T cell counts in PDAC patient tumors. Combined anti-TNF-α and standard chemotherapy reduced CD68+/TNF-α+ macrophages and restored CD3+/CD8+ T cell infiltration, improving the overall outcome in preclinical models. These molecular insights may help define therapeutic vulnerabilities and subtype-based precision therapy strategies cognizant of intratumoral subtype co-existence in PDAC.

## Results

### JUNB associates with GATA6high CLA identity in PDAC patients

In our previous study, the AP1 pathway was found to significantly influence the subtype identity of PDAC through tumor cell-intrinsic and extrinsic mechanisms[21]. Due to their ability to integrate extrinsic inflammatory signals and intrinsic transcriptional programs[21,25,34,36], the AP1 transcription factors (TF) JUNB and cJUN are particularly important in the context of intratumoral subtype plasticity in PDAC. In this study, the initial focus was on JUNB/AP1, as it has been implicated in the identity of low-grade or CLA-like PDAC[16,21,25,37]. Using our JUNB ChIP-seq and ATAC-seq analysis of low-grade or CLA-like PDAC cell lines, we observed an enrichment of pathways related to 'cell adhesion' and 'developmental morphogenesis' (Fig. 1a), supporting the notion that JUNB could promote CLA-like epithelial features and/or early pancreatic differentiation. Hence, we analyzed flow cytometry-sorted epithelial-specific (EPCAM+/CD45−/CD31−) transcriptomes of 31 PDAC patients[22] (Fig. 1b) to investigate a link between JUNB and epithelial-specific features in PDAC. Indeed, our findings revealed a notable association between JUNB expression in the epithelial compartment of patients and the enrichment of classical (CLA) epithelial phenotype signatures such as "CLA-A" and "CLA-B" subtype signatures, as defined by Chan-Seng-Yue et al.[12] (Fig. 1c). Next, we investigated a possible association of JUNB and the bona fide marker of low-grade, CLA subtype PDAC, GATA6. We observed a significant positive correlation between epithelial-specific JUNB and GATA6 expression (Fig. 1d). To further investigate the link between JUNB and a GATA6high CLA-like epithelial cell state, we analyzed 32 treatment-naive, resected PDAC patient tissues. Using triple-IF staining for JUNB, GATA6, and the epithelial marker pan-cytokeratin (panCK; Fig. 1e, Supplementary Fig. 1a), we confirmed a strong relationship between JUNB and GATA6 expression in individual epithelial (panCK+) cells. Our findings revealed that an increased JUNB-positive fraction correlates with high GATA6:JUNB double-positive cells in well-differentiated epithelial neoplastic cells (Fig.1f). Notably, the majority of GATA6+ neoplastic (panCK+) cells also expressed JUNB (62.5%). Furthermore, in pancreatic orthotopic tumor models, low-grade/well-differentiated tumors not only displayed restricted GATA6 and JUNB expression, but also showed a significantly higher number of GATA6:JUNB double-positive cells compared to high-grade/poorly differentiated tumors (Supplementary Fig. 1b,c).

Next, we analyzed a clinical sample from the PMCC cohort of 105 PDAC patient tissues where JUNB and GATA6 protein expression in malignant neoplastic epithelial cells was carefully annotated (Supplementary Fig. 1d–g). Interestingly, JUNB expression was not uniform across each tumor, but rather displayed varying degrees of intratumoral spatial heterogeneity. Since multiple TMA cores were available for each patient, we were able to quantify regional expression differences and observed intratumoral co-existence of JUNBhigh and JUNBlow regions in 32.4% of patients (Supplementary Fig. 1e). We thus tested association of JUNB with the epithelial differentiation markers GATA6 and E-cadherin (ECAD) both at the patient level as well as in individual TMA cores (Fig. 1g). JUNBhigh vs. JUNBlow patient samples showed significantly elevated GATA6 expression (Fig. 1h–j). Importantly, patient-paired analysis showed that GATA6 expression was higher in JUNBhigh than JUNBlow regions within the same patient (Fig. 1j). The epithelial adhesion molecule ECAD showed an analogous elevated expression in JUNBhigh patients and samples, though it was only significant at the regional level (Fig. 1k–m). Accordingly, we observed an overall positive correlation of JUNB and GATA6 expression within patients and within individual TMA cores (Supplementary Fig. 1f, g).

Given positive association between epithelial JUNB and GATA6 in various cohorts of resected early-stage PDAC specimens, we proceeded to investigate whether this association was maintained at later stages. We examined the epithelial-enriched RNA expression dataset

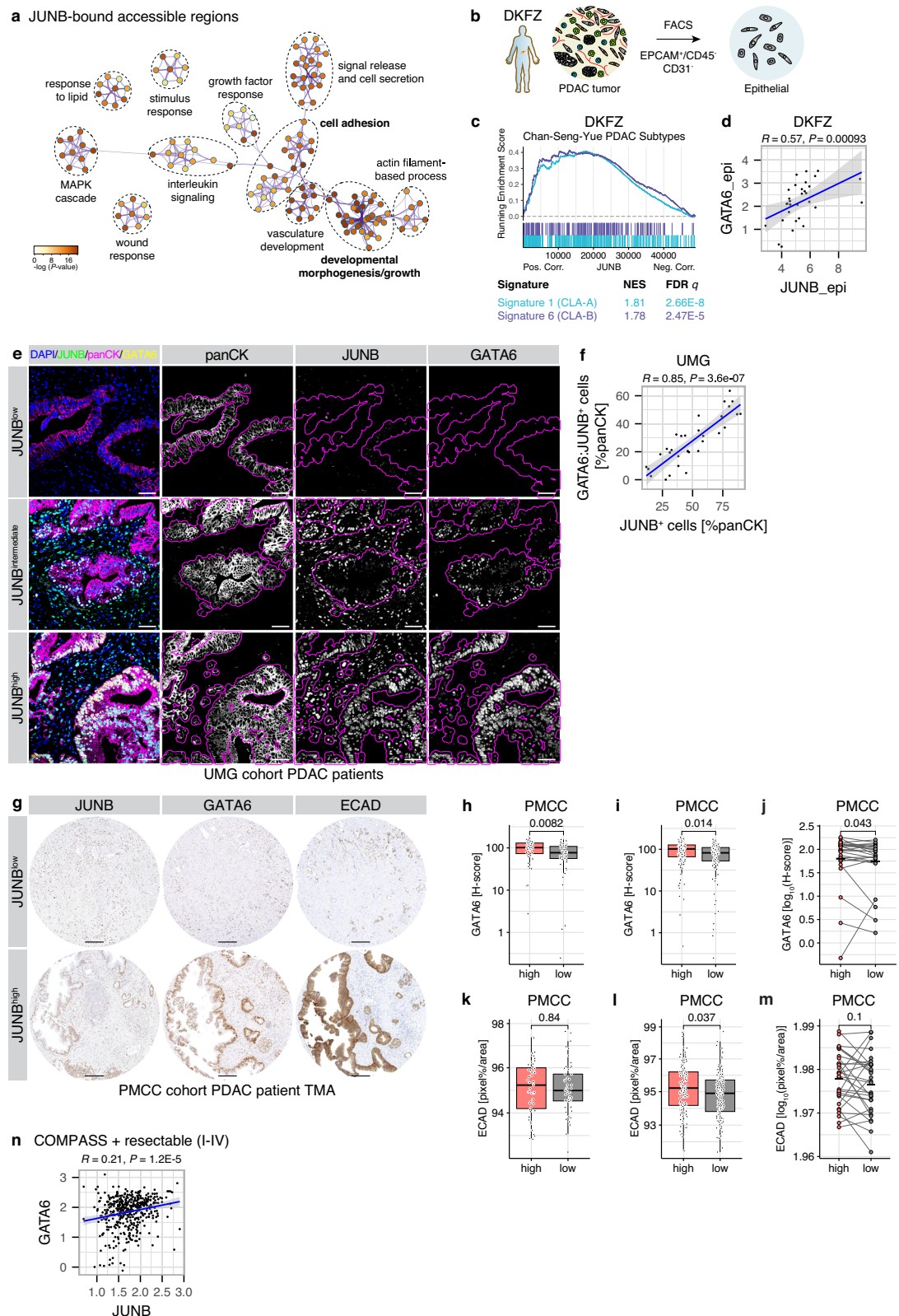

from the COMPASS trial[7], containing data of LCM-enriched epithelia from both early and advanced stage patients (stage I–IV). In this cohort, *JUNB* once again demonstrated a strong correlation with *GATA6* expression (Fig. 1n), further strengthening its association with the CLA-like epithelial state in PDAC.

## JUNB-mediated transcriptional repression affects the CLA phenotype, inflammation, and clinical outcome

As higher JUNB levels were linked to a GATA6-positive, CLA-like phenotype in PDAC neoplastic cells (Fig. 1), we investigated the underlying transcriptional mechanism. We utilized our ChIP-seq data for JUNB (as

**Fig. 1 | Neoplastic JUNB expression associates with GATA6 in PDAC patients.**
**a** Meta-analysis of enriched pathways in regions accessible in CAPAN1 and CAPAN2 (ATAC-seq) and bound in CAPAN1 by JUNB (ChIP-seq). Node color indicates significance, link width the number of gene overlaps between gene sets. **b** Epithelial-specific RNA-seq of resected PDAC patients in the Deutsche Krebsforschungszentrum (DKFZ) was generated by fluorescence-activated cell sorting (FACS) of EPCAM⁺/CD45⁻/CD31⁻ cells. **c** Gene set enrichment analysis for Chan-Seng-Yue PDAC subtypes[12] in genes correlating with JUNB in epithelial compartment-sorted transcriptomes of **b**. Normalized enrichment score (NES) and FDR *q* value are indicated. **d** Correlation analysis for epithelial-specific JUNB and GATA6. Linear regression with 95% CI, as well as Spearman's *R* and associated *P* value. *n* = 31 patients. **e** IF for JUNB, GATA6, and pan-cytokeratin (panCK) in resection tissue of therapy-naive PDAC patients at representative region with high, intermediate, and low epithelial JUNB expression in the University Medical Center Göttingen (UMG) cohort. Epithelial area is overlaid on greyscale images in magenta, based on

panCK⁺ cell classification. In the overlay, blue: DAPI, green: JUNB, magenta: panCK, yellow: GATA6. Scale bar 50 µm. **f** Quantification of (**e**) for JUNB⁺ and GATA6:JUNB double-positive epithelial (panCK⁺) cells, plotted as in (**d**). *n* = 32 patients. **g–m** IHC analysis in 105 PDAC patients of the Princess Margaret Cancer Centre (PMCC) for epithelial JUNB expression. **g** IHC for JUNB, GATA6 and ECAD in cores classified as JUNB^low and JUNB^high. Scale bar 200 µm. **h–m** Quantification of (**g**), for GATA6 (**h–j**) and ECAD (**k–m**) in JUNB^low and JUNB^high expression per patient (**h, k**), per TMA core across all patients (**i, l**) and in heterogeneous patients showing matched levels in JUNB^low and JUNB^high cores (**j,m**). **h,j**, high, *n* = 51; low, *n* = 52 patients. **i**, high, *n* = 106, low, *n* = 112 cores. **k, m**, high, *n* = 51; low, *n* = 49 patients. **l** high, *n* = 123; low, *n* = 119 cores. Boxplots show 25th to 75th percentile with median as box and highest and lowest value in 1.5 times interquartile range as whiskers. Two-tailed Wilcoxon rank sum test. **n** Correlation analysis for JUNB and GATA6 in LCM-enriched epithelia of COMPASS trial patients (stage I–IV), plotted as in (**d**). *n* = 439 patients. Source data are provided as a Source Data file.

in Fig. 1a), together with publicly available H3K27ac[25] data to determine the potential direct regulatory effects of JUNB on lineage TFs of the low-grade/CLA-like PDAC. JUNB binds not only on itself (Fig. 2a), but crucially also on a potential downstream enhancer of *GATA6* (Fig. 2b). Other CLA-associated factors such as *HNF1B* and *FOXA1* are also bound directly by JUNB in CLA-like PDAC cells at intronic and promoter regions, respectively (Fig. 2c,d). Both JUNB binding (Fig. 2e) and H3K27ac occupancy (Fig. 2f) were validated via ChIP-qPCR in CLA cells, which showed the strongest binding of JUNB at the *GATA6* locus. Next, regulatory effects of JUNB were investigated in a global approach by integrating JUNB-bound regions by ChIP-seq with differential expression upon silencing of JUNB (siJUNB) compared to control siRNA (siCtrl; Supplementary Fig. 2a). Genes directly bound by JUNB showed a higher fold change in gene expression than all genes (Fig. 2g). As further illustrated in Fig. 2h, JUNB-bound genes were preferentially upregulated upon silencing, suggesting direct repression by JUNB. Gene ontology analysis of genes directly bound by JUNB indicated that pathways involved in cell migration/stemness, inflammatory signaling, as well as histone deacetylase (HDAC) targets were enriched upon JUNB silencing in PDAC cells (Fig. 2i). Notably, these genes contained major BL-specific driver genes, such as *CD9*, *MYC*, *TP63*, and *cJUN* (Supplementary Table 3), indicating a direct tumor cell-intrinsic repression of BL features. In accordance, JUNB silencing in established (CAPAN2, CFPAC-1) as well as in PDX-derived (JUNB^high GCDX62) CLA cell lines led to a more invasive state (Supplementary Fig. 2b–j), which is a characteristic of BL PDAC cells.

We then investigated how these JUNB regulatory effects impact the overall clinical outcomes in PDAC patients. To enhance the global applicability of the genes directly regulated by JUNB, which were identified through combined ChIP- and RNA-seq in CAPAN1 cells (Fig. 2h), we analyzed the correlation of *JUNB* to each of these 146 genes in a panel of 46 PDAC cell lines from the Cancer Cell Line Encyclopedia[38,39] (CCLE). We focused the analysis on genes that are negatively associated with *JUNB* (Fig. 2j). Subsequently, the 37-gene "JUNB repression signature" retained the enrichment of pathways consistently, as identified in Fig. 2i, related to PDAC aggressiveness and inflammatory TNF-α signatures (Supplementary Fig. 2k). We then used gene set variation analysis (GSVA) for the stratification of patients by this signature (Supplementary Fig. 2l–o). In the combined TCGA, QCMG, Puleo, and Zhou datasets (total *n* = 652), this revealed a clear overall survival benefit for low expression of the JUNB repression signature (16 vs 24 months in upper vs lower quartile; hazard ratio = 1.49, 95% CI = 1.13–2.0; Fig. 2k). Particularly, a stark difference was noted in the progression-free survival rate among the TCGA patients, with a hazard ratio = 2.5, 95% CI = 1.36–4.4, for high expression of the signature (Fig. 2l). Additionally, the JUNB repression score was lowest with lower AJCC stage (Supplementary Fig. 2p) and showed a trend towards lower histological grade (Supplementary Fig. 2q). Together, these data suggest that JUNB-dependent repression of BL-associated

pro-inflammatory drivers, such as TNF-α signaling, confer improved survival in PDAC patients.

### JUNB antagonizes cJUN and cytokine expression utilizing HDAC1

Upon silencing JUNB in CLA cell line, we noted significant enrichment of inflammatory response and TNF-α signaling hallmark signatures, as well as TGF-β signaling and IFN-γ response (Supplementary Fig. 3a–d). This finding is particularly interesting given recent studies linking the TNF-α signaling pathway to therapy-induced plasticity and macrophage-driven inflammatory response in PDAC patients[17,21,40]. Therein, the inflammatory cJUN TF, an AP1 family member of JUNB, is a crucial mediator of such pro-inflammatory signaling and BL phenotypic state[21,35].

To understand whether JUNB-dependent repression of inflammatory pathways is associated with cJUN signaling, we validated the expression of core inflammatory factors in the major inflammatory signatures that were repressed by JUNB (Supplementary Fig. 3a–d). Our results showed an expected upregulation of several interleukins (e.g., *IL1B*, *IL6*) and C-X-C/C-C motif chemokines upon JUNB silencing, both in RNA-seq (Fig. 3a) and qPCR (Fig. 3b). Notably, *cJUN* itself and its downstream target *CCL2* were upregulated upon JUNB silencing (Fig. 3b). Furthermore, ChIP-seq data of JUNB and H3K27ac indicated strong JUNB binding in the absence of H3K27ac at the loci for *cJUN* (Fig. 3c), *IL1B* (Fig. 3d) and *CXCL9/10/11* (Fig. 3e), suggesting repression.

The mechanism through which JUNB exerts transcriptional repression in PDAC is poorly understood. TFs rely on additional epigenetic co-regulators to exert their regulatory functions on lineage gene expression. A previous study identified key epigenetic regulators, such as HDAC1, as crucial in determining PDAC subtype heterogeneity[26]. Particularly, in the gene signatures directly repressed by JUNB (see Fig. 2i, Supplementary Fig. 2k) and in GSEA in JUNB silencing transcriptome data (Fig. 3f), HDAC target signatures were found to be enriched. Therefore, we hypothesized that HDAC1 may be involved in JUNB-mediated transcriptional repression of BL-associated inflammatory lineage signatures. To determine whether HDAC1 cooperates with JUNB in transcriptional repression, we investigated protein-protein interaction, which confirmed a direct binding between JUNB and HDAC1 (Fig. 3g,h). Importantly, targeted ChIP followed by qPCR analysis further validated significant binding of both JUNB (Fig. 3i) and HDAC1 (Fig. 3j) at the repressed loci, which suggests that HDAC1 deacetylates and thereby represses these inflammatory genes.

In order to assess the impact of JUNB on the suppression of inflammatory genes through HDAC1 on a genome-wide scale, we performed ChIP-seq for HDAC1 and H3K27ac following JUNB silencing or with control siRNA (Supplementary Fig. 3e,f). A pronounced loss of HDAC1 occupancy was noticed upon depletion of JUNB, indicating that JUNB is necessary to maintain HDAC1 recruitment to the genome. Overlap of the HDAC1-loss sites with the established JUNB binding regions (as used in Fig. 1a) revealed 1454 sites that are significantly

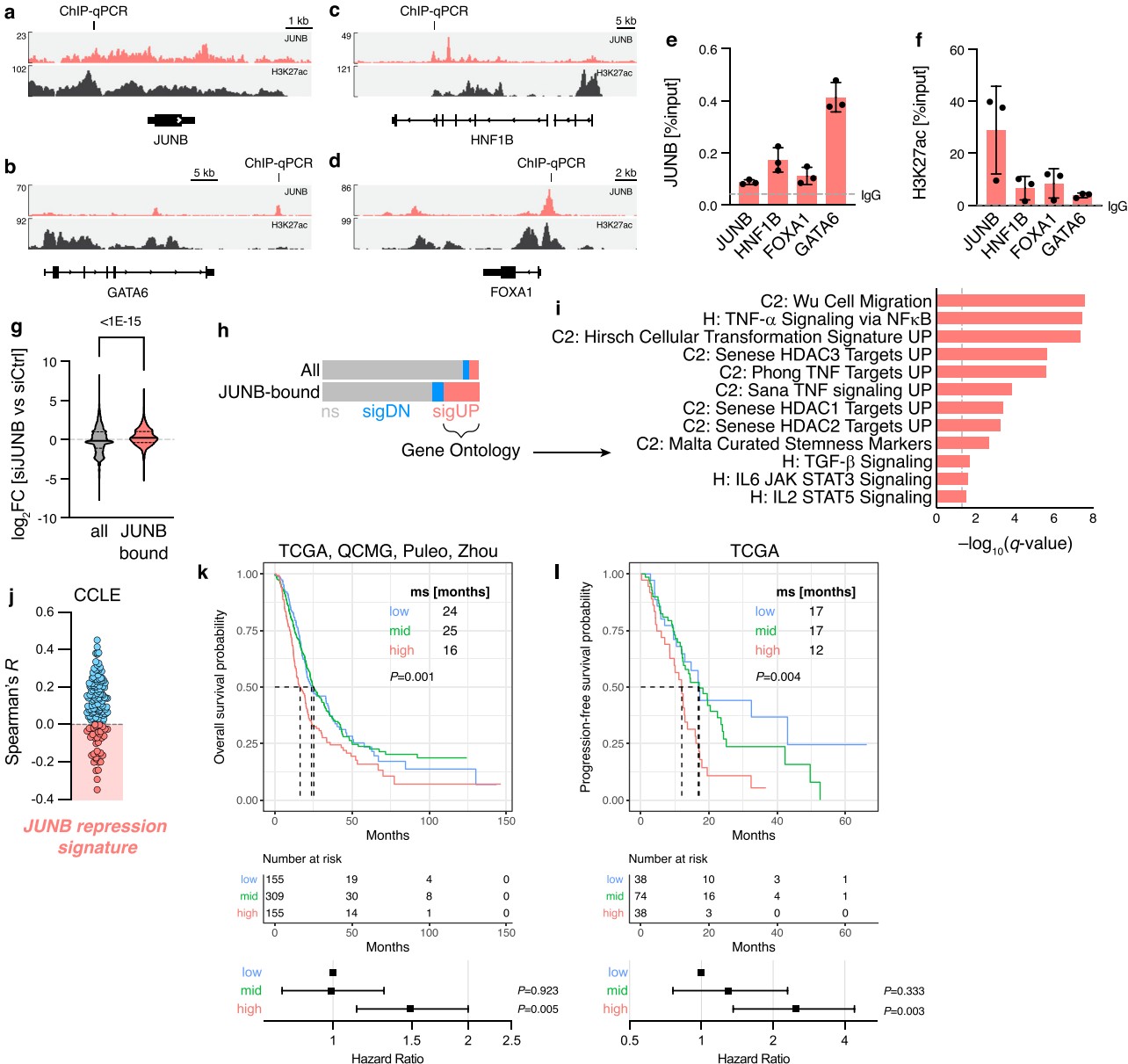

**Fig. 2 | Prognostic relevance of JUNB-repressed inflammatory signaling.** Coverage of previously published[21] JUNB ChIP-seq data in CAPAN1, as well as publicly available H3K27ac[25] data, for loci of JUNB (**a**), GATA6 (**b**), HNF1B (**c**), and FOXA1 (**d**). ChIP-qPCR validation regions are indicated. ChIP-qPCR for regions indicated in (**a**–**d**), showing signal relative to input for JUNB (**e**) and H3K27ac (**f**) pulldown with mean ± s.d. and average IgG isotype control. $n = 3$ biological replicates. **g**–**i** Integration of RNA-seq data performed after JUNB silencing (siJUNB; $n = 3$ biological replicates) or control siRNA (siCtrl; $n = 2$ biological replicates) in CAPAN1, with ChIP-seq for JUNB. **g** Violin plot of $\log_2$ fold change (FC) in siJUNB RNA-seq data for all ($n = 36.740$) or JUNB-bound ($n = 698$) genes. Median and quartiles are indicated. Two-tailed Student's $t$-test with Welch's correction. **h** As in (**g**), showing the number of genes that display a significant upregulation (sigUP) or downregulation (sigDN), or no significant change (ns). **i** Gene ontology analysis of significantly upregulated, JUNB-bound genes following JUNB silencing with $-\log_{10}(q$-value) indicated. Hallmark (H) and curated (C2) signature collections of the Molecular Signature Database (MSigDB) are shown. **j** Spearman correlation of genes as in (**i**) with JUNB in 46 PDAC cell lines of the Cancer Cell Line Encyclopedia (CCLE). Negatively associated genes (red) form the JUNB repression signature. **k** Overall survival, numbers at risk, and hazard ratio in TCGA ($n = 150$), Puleo ($n = 288$), QCMG ($n = 96$), and Zhou ($n = 85$) patients stratified by JUNB repression signature (**j**) score. Top: Kaplan-Meier survival analysis for the lower/upper quartiles ($n = 155$ patients each) and mid-group ($n = 309$ patients) for JUNB repression signature scores. Median survival (ms) is indicated. Log-rank test. Bottom: Cox proportional hazard. Hazard ratio (to lower quartile) with 95% CI. $P$ values are shown right. **l** As in (**k**), for progression-free survival in the TCGA cohort. Source data are provided as a Source Data file.

enriched for TNF-α pathway signature (Fig. 3k, l), highlighting the importance of HDAC1 in maintaining repression of these BL inflammatory processes with JUNB. As a likely response to the loss in HDAC1, we observed increased H3K27 acetylation at 3589 JUNB-bound sites upon siJUNB, which are also strongly enriched in inflammatory response signatures, particularly TNF-α signaling (Supplementary Fig. 3g,h). This genome-wide analysis further confirms the role of JUNB in repressing TNF-α signaling and its associated BL inflammatory

genes, such as the macrophage recruiting factor CCL2, and its regulator, the TF cJUN.

Consistent with our RNA expression and ChIP-qPCR data, we found protein expression of cJUN induced in both established and PDX-derived CLA-like PDAC cell lines (Supplementary Fig. 3i,j). We also observed an upregulation CCL2 in CLA-like PDAC cell lines (Fig. 3m, n, Supplementary Fig. 3k), which is a direct target of cJUN[21], indicating a direct repression of cJUN by JUNB. This was further

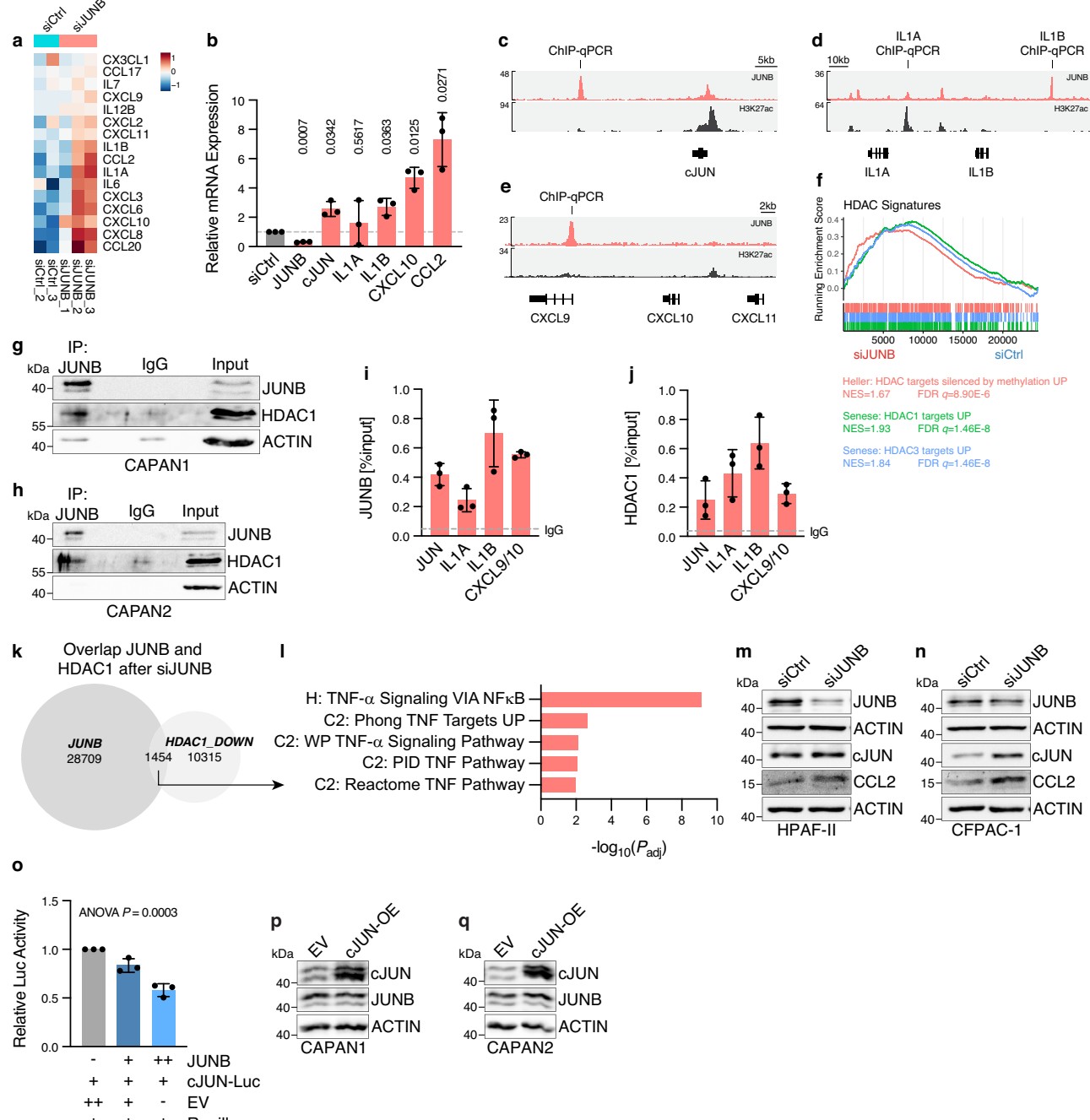

**Fig. 3 | JUNB-HDAC1 complex represses inflammatory signals and cJUN.**
**a** Heatmap showing expression of cytokines present in the core enrichment of the gene sets shown in Supplementary Fig. 3a–d, for JUNB silencing (siJUNB; *n* = 3 biological replicates) versus control siRNA (siCtrl; *n* = 2 biological replicates) in CAPAN1 cells. Cell color indicates z score. **b** qRT-PCR analysis for indicated target genes in siJUNB conditions (red), normalized to siCtrl (gray), in CAPAN1. Relative mRNA expression with mean ± s.d. shown. *n* = 3 biological replicates. Two-tailed Student's t-test with Welch's correction. Coverage of JUNB ChIP-seq data in CAPAN1[21], as well as publicly available H3K27ac[25] data, for loci of cJUN (**c**), IL1A/B (**d**), and CXCL9/10/11 (**e**). ChIP-qPCR validation regions are indicated. **f** Gene set enrichment analysis for curated signatures (C2) of the Molecular Signature Database (MSigDB) for siJUNB versus siCtrl in CAPAN1 cells. Normalized enrichment score (NES) and FDR *q* value are indicated. Immunoblot for JUNB, HDAC1, and β-actin after JUNB pulldown, IgG isotype control or input in CAPAN1 (**g**) and CAPAN2 (**h**). *n* = 3 biological replicates. **i, j**, ChIP-qPCR for regions indicated in (**c–e**), showing signal relative to input for JUNB (**i**) and HDAC1 (**j**) pulldown with

mean ± s.d. and average IgG isotype control. *n* = 3 biological replicates. **k, l**, ChIP-seq analysis for JUNB in control cells (as in **c–e**) and HDAC1 with siJUNB or siCtrl. **k** Overlap of JUNB binding regions and regions where HDAC1 is significantly lost upon siJUNB ("HDAC1_DOWN"). **l** GREAT analysis of the overlapping regions of (**k**) with −log₁₀(*P*adj) for binomial test indicated. Hallmark (H) and C2 signatures of MSigDB are shown. Representative immunoblot for JUNB, cJUN, CCL2, and β-actin in HPAF-II (**m**) and CFPAC-1 (**n**) after siJUNB or siCtrl. *n* = 3 biological replicates. **o** Dual-luciferase reporter assay for cJUN promoter firefly luciferase (Luc) constructs in CAPAN2 cells with varying concentrations of JUNB overexpression plasmids (or EV controls). Relative Luc activity to *Renilla* luciferase control with mean ± s.d. shown. One-way ANOVA. n = 3 biological replicates. Immunoblot for JUNB, cJUN, and β-actin in CAPAN1 (**p**) and CAPAN2 (**q**) cells with overexpression of cJUN (cJUN-OE) or empty vector (EV) control. *n* = 3 biological replicates. Source data are provided as a Source Data file.

confirmed by dual luciferase reporter assays for the promoter of *cJUN*, which showed that JUNB was able to directly downregulate *cJUN* (Fig. 3o, Supplementary Fig. 3l,m), suggesting that cJUN plays an antagonistic role, activating BL subtype-associated inflammatory genes. To determine the molecular and functional differences between cJUN and JUNB in subtype plasticity, we analyzed cJUN-bound regions by ChIP-seq upon cJUN overexpression (cJUN-OE) in JUNB[high] CLA cell lines (Supplementary Fig. 3n,o). Notably, unlike JUNB, cJUN-bound genes showed no preference for up- or down-regulation in RNA-seq data following HA-tagged cJUN-OE compared to empty vector (EV) controls (Supplementary Fig. 3n), underlining their diverging functions in PDAC subtype plasticity. Gene ontology analysis of up- and downregulated genes (Supplementary Fig. 3o) showed that the aggressiveness-associated pathways repressed by JUNB (e.g., "cell migration"; Fig. 2i) were directly activated by cJUN (Supplementary Fig. 3p). In particular, BL-associated EMT and TNF-α gene signatures were enriched among cJUN-bound genes (Supplementary Fig. 3p), indicating that cJUN may attenuate CLA-associated functions by antagonizing JUNB signaling. Interestingly, however, the expression of JUNB remained unchanged when cJUN was over-expressed in CLA cell lines (Fig. 3p, q). This led us to hypothesize that cJUN may employ indirect pathways, such as microenvironmental extrinsic factors, to antagonize JUNB-dependent signaling. Together, these findings suggest that JUNB restricts BL pro-inflammatory pro-grams via HDAC1-mediated transcriptional repression.

### Antagonistic roles of AP1 factors determine regional immune recruitment

Detailed gene expression analysis has shown potential immune-modulatory effects of the antagonistic JUNB-cJUN interplay (Fig. 3). To investigate the opposing functions of JUNB and cJUN in the spatial TME, we orthotopically implanted both cJUN-OE and EV control CLA-like PDAC cell lines into the pancreas of immunodeficient nude mice (Fig. 4a). Initially, we confirmed the expected high nuclear cJUN expression levels compared to EV in cJUN-OE PDAC tumors (Supplementary Fig. 4a,b). Intriguingly though, not all ductal cells in the HA-cJUN-OE tumors exhibited cJUN expression (Supplementary Fig. 4a). There were no obvious tumor histological differences between the groups, however, we interestingly observed a trend towards higher immune infiltrations (Fig. 4b). This was further supported by an increase in CD45[+]/CD68[+] and TNF-α[+]/CD68[+] macrophages in cJUN-OE CLA-derived tumors (Supplementary Fig. 4c-e), in line with the pro-inflammatory effects mediated by cJUN.

As JUNB attenuated the expression of cJUN (Fig. 3m–o, Supplementary Fig. 3i-m), we then assessed whether cJUN[low] areas in this heterogeneous tumor model exhibited high JUNB expression. This presented a valuable opportunity to investigate spatial effects of the AP1 heterogeneity in PDAC. Using whole slide images of IHC for HA-cJUN (Fig. 4c) and IF for JUNB (Fig. 4d) in serial sections, we marked and quantified "hotspot" regions of high cJUN and high JUNB expression. This revealed not only reduced HA-cJUN[+] cells in JUNB hotspots, but JUNB[+] cells were vice versa depleted in cJUN hotspot areas (Fig. 4e). Since cJUN-OE did not lead to a direct repression of JUNB expression (Fig. 3p, q), we hypothesized that microenviron-mental factors may affect JUNB expression and shape PDAC plasti-city. Previous studies have reported that microenvironmental factors such as TNF-α or TGF-β can influence subtype specificity[20,21]. As mentioned, we observed higher TNF-α[+] macrophages in cJUN-OE tumors (Supplementary Fig. 4c-e). Therefore, we sought to deter-mine whether regional CD68[+] macrophage infiltrations, particularly surrounding cJUN[+] hotspot area, might destabilize JUNB expression (Fig. 4f). Indeed, the average distance of CD68[+] macrophages to cJUN hotspots was lower than to JUNB hotspots (448.5 vs 645.3 µm; Fig. 4g). When analyzing the distance of each cell to either hotspot, it

was observed that significantly more CD68[+] macrophages were clo-ser to the cJUN hotspot (Fig. 4h). Upon further examination of the type of macrophages using IHC in the same manner, it was noted that M2-like (CD163[+]) macrophages are located closer to cJUN hotspots (Fig. 4i,j), while the opposite is true for M1-like (CD86[+]) macrophages, which are nearer to JUNB hotspots (Fig. 4k, l). Notably, M2 macro-phages are linked to the BL subtype and poor prognosis in PDAC patients[14,17,41]. Mechanistically, these M2 macrophages are likely recruited through CCL2 secretion at cJUN hotspots, as CCL2[+] cells in IF staining of the cJUN-OE tissues were also found significantly closer to cJUN hotspots (Fig. 4m, n).

Thus, cJUN[+] cell clusters appeared to preferentially recruit M2 macrophages via CCL2, which was restricted in JUNB[+] areas, potentially affecting AP1 heterogeneity through spatial inflammatory cues. Col-lectively, these results suggest that regional inflammatory macro-phages can influence neoplastic stability by influencing AP1 transcriptional programs.

### TNF-α destabilizes CLA-like neoplastic state and shapes local TME heterogeneity

It has been shown that TNF-α or TGF-β can influence subtype specificity[20,21], yet the impact of inflammatory cell-derived TNF-α on AP1 heterogeneity is unknown. Thus, to determine whether TNF-α destabilizes the CLA-like epithelial state (potentially by affecting JUNB signaling) and gradually promotes BL plasticity through neo-plastic co-existence, we analyzed its impact on transcriptional sig-natures in vitro and in vivo. Among genes upregulated in RNA-seq data of CLA-like PDAC cells upon exogenous TNF-α treatment, we found *JUNB*, together with a shift in established CLA and BL subtype markers (Fig. 5a). To further test this observation in vivo, we utilized a CLA-derived orthotopic murine model which was treated for three weeks with exogenous TNF-α. These tumors were then subjected to comprehensive cell type-specific transcriptome analysis. Alignment of bulk RNA-seq of these tumors to human (implanted neoplastic epithelial cells) or murine (host stromal cells) reference genomes and XenofilteR-based removal of the opposite species reads allowed generation of virtually microdissected, compartment-specific tran-scriptomes (Fig. 5b). Within the tumor cell-specific data, GSEA showed a strong enrichment of TNF-α signaling pathways (Fig. 5c), confirming that tumor cells reacted to the exogenous treatment. In accordance with the in vitro data, TNF-α treatment led to repression of CLA gene signatures in vivo (Fig. 5d). The stromal population also responded to the TNF-α treatment with induction of TNF-α signaling (Fig. 5e), along with a significant remodeling of the stromal immune populations, as determined by MCPcounter (Fig. 5f). Specifically, cytotoxic T cell, as well as B cell signatures were reduced after TNF-α treatment. Further, CIBERSORTx-based deconvolution analysis revealed a significant increase in M2 macrophages (Supplementary Fig. 5a), consistent with the observations in the cJUN-OE tumors (see Fig. 4). PDAC patients with a high expression of the JUNB repression signature genes (see Fig. 2) also showed a reduced T and B cells, along with an increase in monocytic lineage cells by MCPcounter (Fig. 5g, Supplementary Fig. 5b). Further deconvolution analysis by CIBERSORTx confirmed a decrease in CD8 T cells, as well as naïve and memory B cells, and particularly an increase in M2 macrophages, with no significant changes in monocytes, M0 or M1 macrophages (Fig. 5h, Supplementary Fig. 5c). As shown above, the JUNB repres-sion signature was directly associated with TNF-α signaling as well (Supplementary Fig. 2k). Hence, we suspected that JUNB expression might be reduced in response to TNF-α, potentially contributing to the indirect cJUN-CCL2-dependent attenuation of JUNB in the spatial TiME. Indeed, IF staining for JUNB (Fig. 5i) confirmed a reduction in nuclear neoplastic JUNB intensity upon TNF-α treatment (Fig. 5j). In addition, the expression of the CLA subtype marker ECAD and

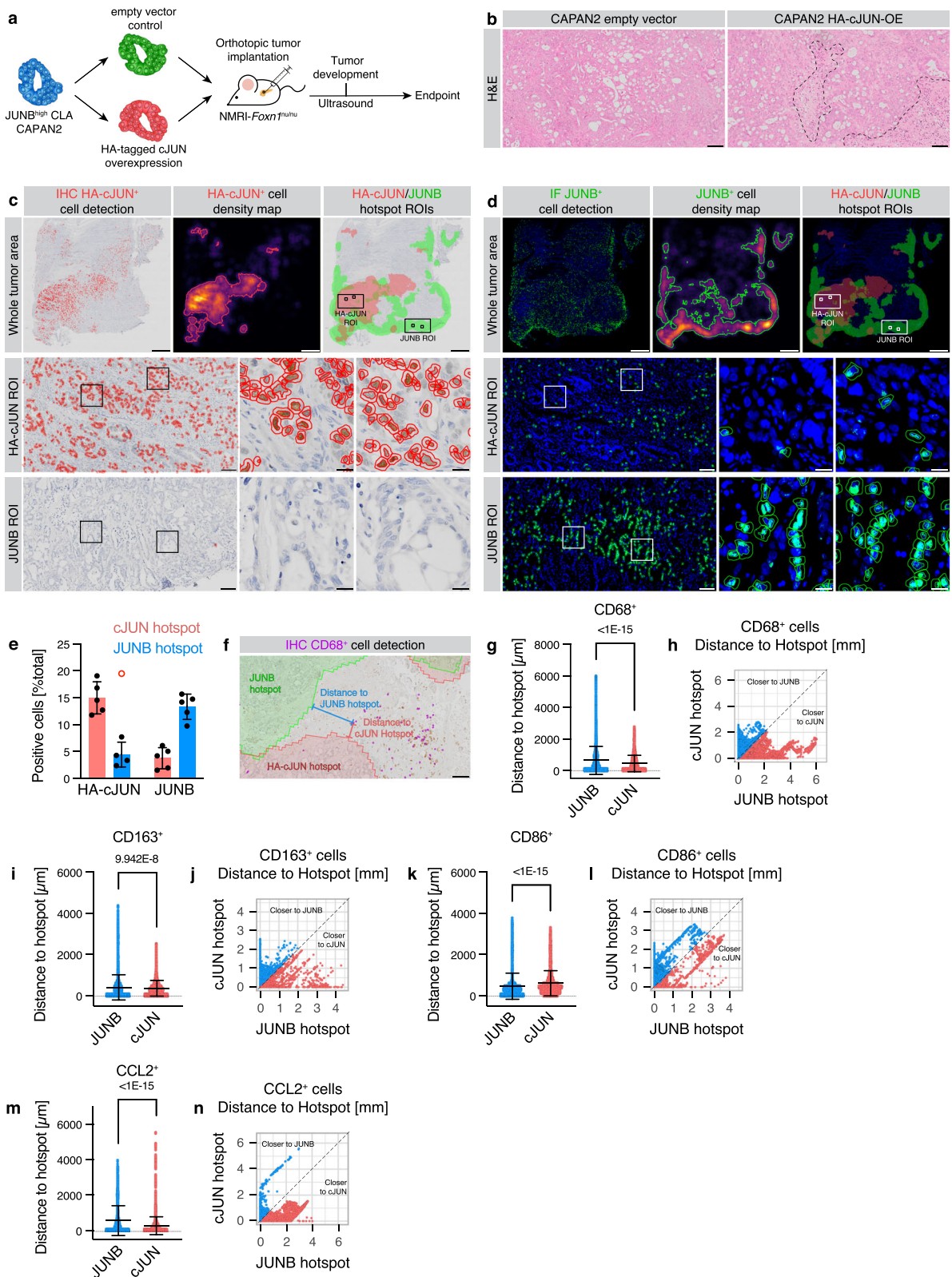

nuclear GATA6 were also significantly reduced (Fig. 5k–m). Furthermore, an increase in M2-like macrophages (CD163[+]) was validated in the TNF-α-treated tissues, corroborating the findings of the in silico CIBERSORTx analysis (Fig. 5n, o). Together, these results suggest that TNF-α directly influences the remodeling of the immune microenvironment and the plasticity of neoplastic cells, thus fostering neoplastic co-existence and PDAC aggressiveness.

## TNF-α promotes reactive spatial TiME heterogeneity in PDAC patients

Exogenous TNF-α treatment affected the TiME as well as neoplastic cell plasticity in experimental models (Fig. 5). Next, we set out to correlate TNF-α expression with spatial TME functions such as recruitment of CD68[+] macrophages or CD3[+], CD4[+], CD8[+] T cell infiltrations at histological levels in PDAC patients. We analyzed TNF-α expression and its

**Fig. 4 | Regional AP1 heterogeneity determines macrophage recruitment.**
**a** NMRI-*Foxn1*[nu/nu] mice were orthotopically transplanted with CAPAN2 cells with stable HA-tagged cJUN overexpression (HA-cJUN-OE) or empty vector (EV) control. **b** H&E staining of CAPAN2 HA-cJUN-OE and EV tumors. Immune infiltrates are indicated. Scale bar 100 μm. *n* = 8 animals. **c-n**, QuPath-based analysis of HA-cJUN-OE tumors for HA-cJUN IHC, DAPI-JUNB IF, CD68 IHC, CD163 IHC, CD86 IHC, and DAPI-CCL2 IF. IHC for the HA tag of cJUN (**c**) or IF for JUNB (**d**) in serial sections. Nuclear-positive cell detections (red/green) are indicated. Density maps of positive cells were created and thresholded to derive hotspot regions for HA-cJUN+ and JUNB+ cells, respectively, in the same tumors. Whole tumor overviews as well as a HA-cJUN (mid) and a JUNB (bottom) hotspot ROIs are shown. Scale bar: tumor overview, 1 mm; large ROI, 100 μm; ROI insert, 20 μm. **e** Quantification of (**c**, **d**) for HA-cJUN+ and JUNB+ cells relative to the total number of detected cells in cJUN (red) and JUNB (blue) hotspot regions, with mean ± s.d. shown. One outlier is indicated

(red circle), which was excluded for mean and s.d. *n* = 5 tumors. **f** IHC for CD68 in HA-cJUN-OE tumors, with CD68+ cell detection (purple), as well as JUNB and HA-cJUN hotspots. Exemplary 2D distance measurement strategy which was used in (**g-n**) is shown. Scale bar 100 μm. Distance analysis of CD68+ (**g**), CD163+ (**i**), CD86+ (**k**), or CCL2+ (**m**) cells to JUNB or HA-cJUN hotspots. Scatter plots show each individual cell, with mean ± s.d. Two-tailed Student's *t*-test with Welch's correction. **g**, *n* = 14774 CD68+ cells from *n* = 5 tumors, with a total of *n* = 1,003,639 cells analyzed. **i**, *n* = 7691 CD163+ cells from *n* = 4 tumors, with a total of *n* = 654,297 cells analyzed. **k** *n* = 8384 CD86+ cells from *n* = 4 tumors, with a total of *n* = 632,792 cells analyzed. **m** *n* = 14,925 CCL2+ cells from *n* = 5 tumors, with a total of *n* = 745,208 cells analyzed. **h, j, l, n** As in (**g, i, k, m**) showing the shortest distances of each cell towards both the HA-cJUN and JUNB hotspots. Source data are provided as a Source Data file.

effects in 105 PDAC patients of the PMCC cohort by IHC. Overall, TNF-α levels were highly heterogeneous, with 46.9% of tumors displaying strong spatial variation in its expression (Fig. 6a). A major source of TNF-α in the TiME are macrophages[21,42]; indeed, TNF-α[high] samples exhibited higher CD68 scores, both globally as well as in the individual samples (Fig. 6b–d). Importantly, TNF-α-dependent regional remodeling of the TiME seen in mice was recapitulated in patients, as the lymphocyte populations, particularly CD8+ T cells, were significantly reduced in TNF-α[high/int] compared to TNF-α[low] patients (Fig. 6e, Supplementary Fig. 6a–g). Recently, we have shown that the heterogeneous PDAC ecosystem self-organizes into 'deserted' and 'reactive' sub-tumor microenvironments (subTMEs), which leads to intratumoral zonation with co-existing immune-cold and immune-hot regions in human PDAC[43]. In accordance with these diverse tumor ecosystems, regional TNF-α expression was strongly increased within the immune-rich 'reactive' subTME regions (Fig. 6f–h), which supports a BL inflammatory phenotypic state in PDAC patients[43]. Together, these results indicate that high TNF-α levels are involved in an immunosuppressive TME that supports a BL state in PDAC.

**Targeting TNF-α during chemotherapy leads to favorable TiME reorganization**
Finally, we tested whether targeting TNF-α could shift tumors towards a favorable clinical state, given the central role of TNF-α in shaping PDAC subtype co-existence by influencing the JUNB-cJUN dichotomy. Anti-TNF-α monotherapy is not effective in aggressive PDAC[21]. Similarly, gemcitabine (GEM) chemotherapy alone or in combination with paclitaxel is essentially ineffective in *Kras*[G12D];*p53*[R172H];*Pdx1-Cre* (KPC)-derived murine PDAC models[23,29]. Thus, we tested whether combination of GEM with TNF-α inhibition may enhance treatment response. We utilized a highly aggressive KPC-derived orthotopic model and treated the animals with a combination of GEM plus anti-TNF-α monoclonal antibody therapy (Fig. 7a). This significantly prolonged overall survival from 19 to 32 days (Fig. 7b). While tumors maintained comparable gross histology (Fig. 7c), analysis of CD45, CD68, and TNF-α by IF showed a significant reduction in the number of CD45+/CD68+ macrophages, as well as reduction in CD45+/TNF-α+ cells (Fig. 7d–f). Notably, this also resulted in a significant increase in CD3+ as well as cytotoxic CD8+ T cells in the TME (Fig. 7g–i). Thus, TNF-α-dependent macrophage recruitment appeared to be halted, leading to a less immunosuppressive TiME in GEM plus anti-TNF-α-treated PDAC tumors, which highlights the important role of TNF-α in shaping the immune landscape to aid tumor growth and survival. The central findings are summarized in Fig. 7j.

## Discussion
PDAC is a highly heterogeneous disease, not only due to the intratumoral co-existence of neoplastic subtypes, but also in terms of its complex, overabundant TME. This subtype co-existence is increased during disease progression and negatively impacts both the predictive

and prognostic utility of the transcriptomic subtypes. However, specific regional drivers of subtype identity and their potential relationship with the heterogeneous TME are currently unknown.

Here, we investigated the role of neoplastic AP1-mediated epigenetic and transcriptional programs in shaping the local inflammatory TiME, which in turn is critical for intratumoral subtype plasticity and PDAC aggressiveness[10–15,20,40,44]. We report that AP1 transcription factors (JUNB/AP1 vs. cJUN/AP1) hold a dichotomous role in maintaining both the plasticity and stability of CLA-like and BL neoplastic cells via intrinsic epigenetic and transcriptional regulation of lineage gene expression as well as extrinsic inflammatory processes. Integrated bioimaging and epithelial-specific transcriptome analyses of PDAC patients showed that high JUNB expression is associated with GATA6+ CLA identity. Mechanistically, neoplastic JUNB positively controls the regulation of CLA-specific lineage factors (HNF1B and GATA6), while epigenetically repressing BL-specific inflammatory immune regulators (cJUN), which is critical for the maintenance of the CLA-like neoplastic state. Intriguingly, this JUNB-mediated epithelial CLA-like state is not stable, but highly plastic in response to inflammatory cues (i.e., TNF-α) as characterized by loss of JUNB+/GATA6+ cells in preclinical models and in PDAC patient specimens. These findings clearly indicate that loss of JUNB/AP1-dependent gene regulation leads to destabilization of CLA-like neoplastic state, induces a CD68+/TNF-α+ macrophage-driven inflammatory response in the TiME and, thereby, promotes a BL invasive state via complementary intrinsic and extrinsic mechanisms. Specifically, the TNF-α-mediated inflammatory response associated with low JUNB expression destabilizes CLA-like neoplastic state by promoting a CLA-to-BL transition via epigenetic transcriptional reprogramming in PDAC. Notably, the maintenance of a CLA-like epithelial state or the suppression of pro-inflammatory factors by JUNB requires the epigenetic co-regulator HDAC1. Our study emphasizes the importance of previous work[26], which has demonstrated how the interaction of epigenetic co-regulators like HDAC1 with lineage-specific TFs can affect PDAC heterogeneity. These results unveil a key reciprocal interdependence between neoplastic (intrinsic) and local microenvironmental (extrinsic) factors that influence subtype plasticity/instability and thereby promote PDAC heterogeneity and aggressiveness.

Emerging evidence shows that cJUN/AP1 TF plays an important role in tumor inflammation, chemotherapy response, and tumor recurrence in PDAC patients[21,35]. Complementarily, we here show that JUNB/AP1 acts as a counterpart to promote a favorable CLA phenotypic state in PDAC. Of note, JUNB/AP1-mediated transcriptional programs can also confer tumor-promoting functions in other cancer types[45–47]; JUN/AP1 TFs are highly context dependent and may co-operate for target gene transcription[47–49] or oppose one another[50]. Our data indicates that in PDAC, JUN/AP TFs exert antagonistic roles, with JUNB directly repressing cJUN and cJUN-regulated cytokine secretion (i.e., CCL2), thereby inhibiting recruitment of CD68+/TNF-α+ M2 macrophages in the TiME. Thus, JUNB not only maintains a CLA subtype

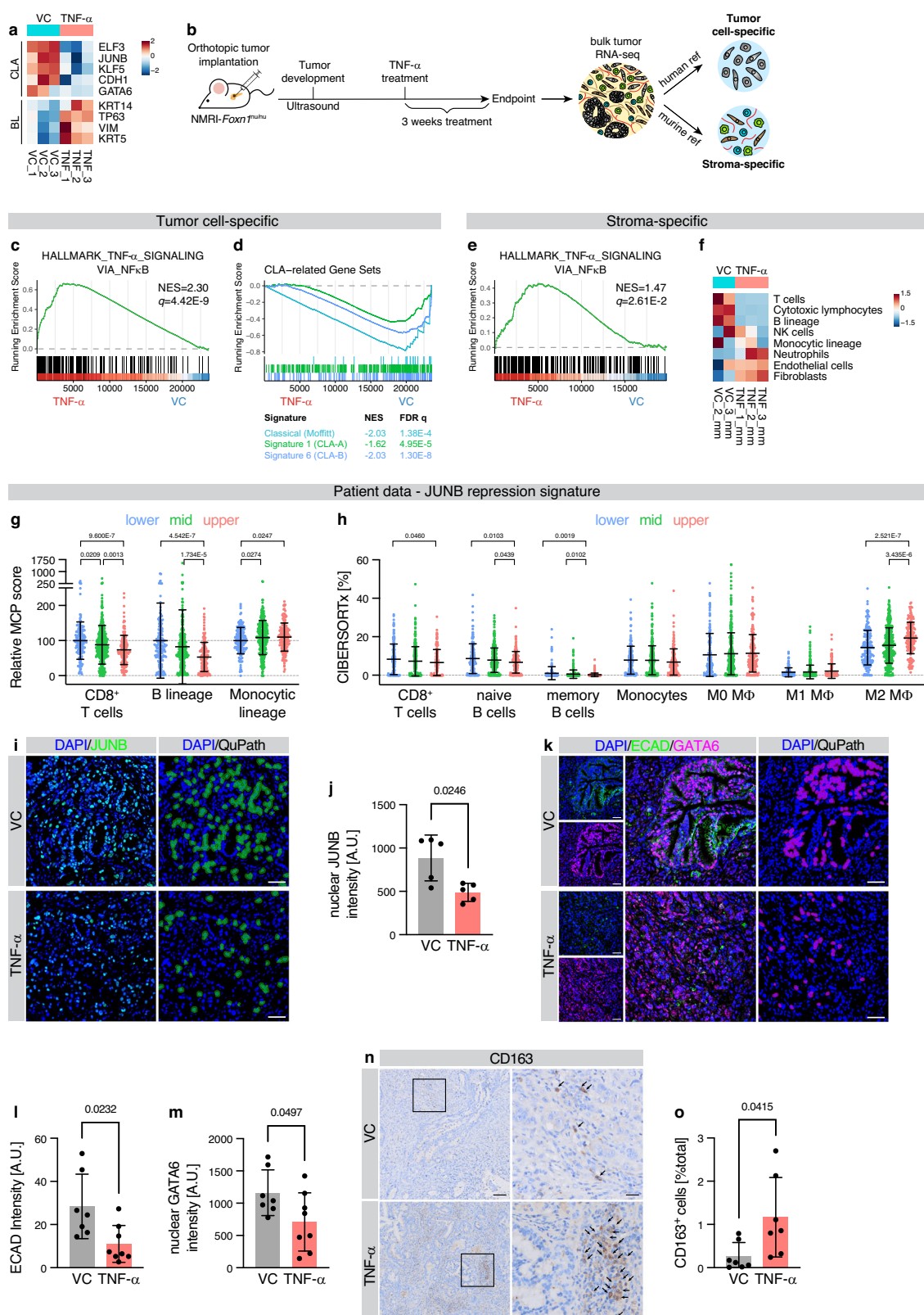

lineage with high expression of epithelial differentiation factors such as *GATA6*, but also directly represses drivers of disease aggressiveness (e.g., *MYC*) as well as inflammatory pathways (e.g., M2 macrophages) associated with poorer clinical outcome in PDAC patients[17]. In this context, our model of CLA-derived orthotopic tumors with cJUN-OE provides a key experimental benefit in that it recapitulates the intratumoral subtype co-existence seen in PDAC patients. Specifically, this

model revealed significant infiltration of CD68[+]/TNF-α[+] M2-like macrophages in the spatial tumor neighborhood of cJUN[high] but not cJUN[low] neoplastic cells, indicating that cJUN exploits regional macrophages to attenuate JUNB expression and thereby suppress CLA-like neoplastic state. These findings were crucial in revealing the mechanisms maintaining AP1 dichotomy: direct (JUNB repressing cJUN) and indirect (cJUN repressing JUNB via local macrophages), which regulate the

**Fig. 5 | TNF-α disrupts CLA subtype identity and anti-tumor immunity.**
**a** Heatmap of CLA and BL PDAC identity genes, in previously published[21] RNA-seq data of CAPAN1 cells treated with TNF-α or vehicle control (VC) for 18 h. Cell color indicates z score. $n = 3$ biological replicates. **b**–**f** Virtually microdissected RNA-seq data of orthotopically transplanted CAPAN1 tumors in NMRI-*Foxn1*[nu/nu] mice treated with TNF-α or VC for 3 weeks. $n = 3$ tumors; one stroma-specific transcriptome was excluded from the analysis. **b** Deconvolution of bulk RNA-seq to generate tumor (human) and stromal (murine) cell-specific transcriptomes (Methods). Tumor cell-specific transcriptome. Gene set enrichment analysis (GSEA) for Hallmark signatures of the Molecular signature database (MSigDB) (**c**) and PDAC subtype signatures (**d**), for TNF-α versus VC. Normalized enrichment score (NES) and FDR $q$-value are indicated. **e** As in (**c**), for stroma-specific transcriptome. **f** MCPcounter analysis in stroma-specific transcriptome. Cell color indicates z score. **g** Relative MCPcounter scores for the indicated lineages in $n = 652$ patients of the TCGA, QCMG, Puleo, and Zhou cohort, separated into quartiles based on the JUNB

repression signature score (as in Fig. 2k, l). MCPcounter scores were min–max normalized and standardized to the mean of the lower JUNB repression signature score group. Mean ± s.d. shown. **h** As in (**g**), but applying CIBERSORTx for deconvolution. Mean ± s.d. for CIBERSORTx percentages shown. **i** IF for JUNB in orthotopically transplanted CAPAN1 tumors treated with TNF-α or VC, with cell detection for nuclear JUNB⁺ cells. Scale bar 50 μm. **j** Quantification of **i**, for per-animal average nuclear JUNB intensity with mean ± s.d. shown. $n = 5$ animals. **k** As in (**i**), for ECAD and GATA6 staining and cell detection for nuclear GATA6⁺ cells. **l** Quantification of **k** for per-animal average ECAD intensity per FOV with mean ± s.d. shown. **m** As in (**l**), for nuclear GATA6 intensity. **l**, **m**, VC, $n = 7$ animals; TNF-α, $n = 8$ animals. **n** As in (**i**), for CD163 IHC staining. Arrows indicate positive cells. Scale bar: overview, 100 μm; insert, 25 μm. **o** Quantification of (**n**) for per-animal percentage of CD163⁺ cells with mean ± s.d. shown. $n = 7$ animals. **g**, **h**, **j**, **l**, **m**, **o** Two-tailed Student's $t$-test with Welch's correction. Source data are provided as a Source Data file.

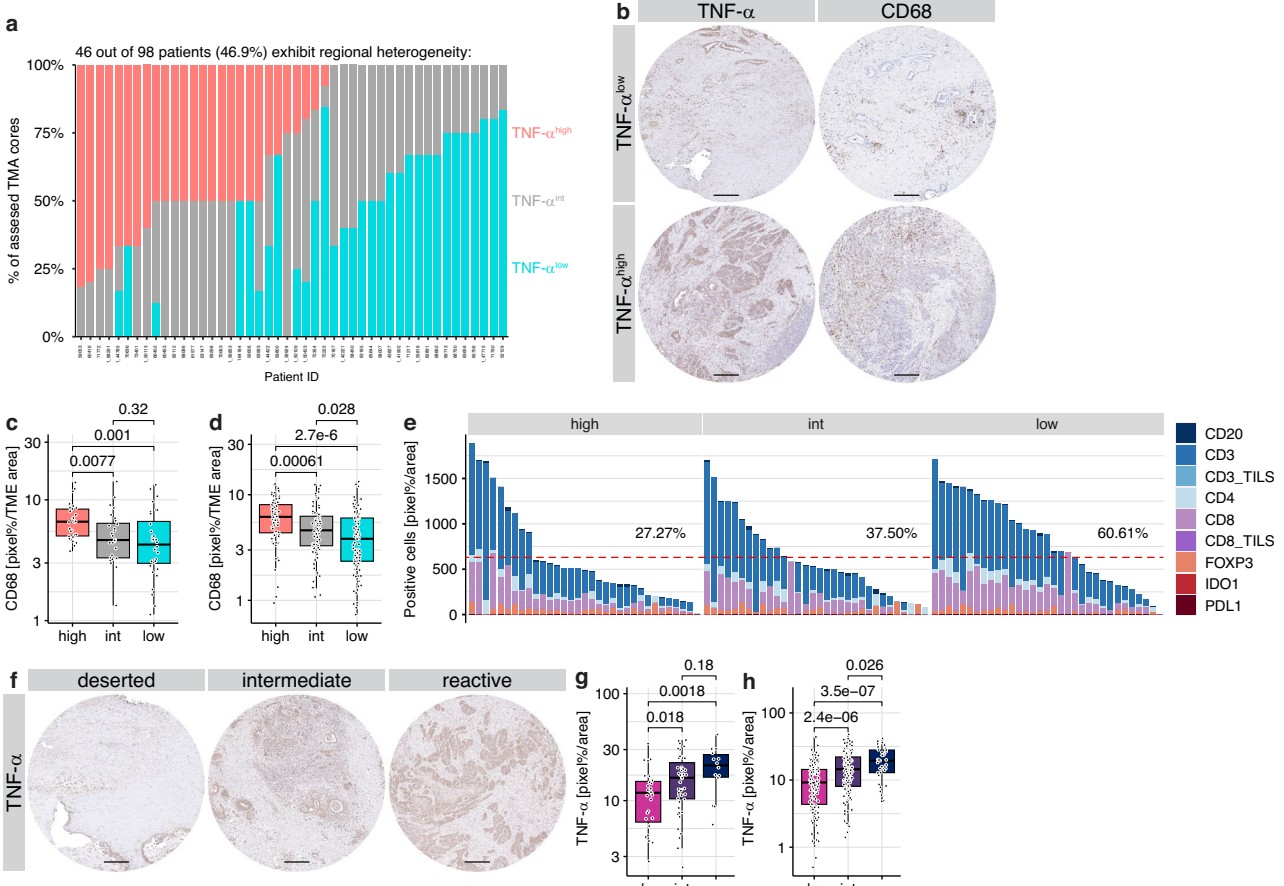

**Fig. 6 | Spatial TNF-α expression promotes macrophage infiltration and T-cell exclusion. a**–**h** IHC analysis in 105 PDAC patients of the Princess Margaret Cancer Centre (PMCC) for TNF-α expression. **a** Spatial heterogeneity of TNF-α expression within different TMA cores of each patient. **b** IHC for TNF-α and CD68 in cores classified as TNF-α[low] and TNF-α[high]. Scale bar 200 μm. **c**, **d** Quantification of (**b**), in TNF-α[low], TNF-α[intermediate] (TNF-α[int]), and TNF-α[high] expression per patient (**c**) and per TMA core across all patients (**d**). **c** TNF-α[high], $n = 30$ patients; TNF-α[int], $n = 28$ patients; TNF-α[low], $n = 32$ patients. **d** TNF-α[high], $n = 87$ cores; TNF-α[int], $n = 92$ cores; TNF-α[low], $n = 96$ cores. **e** Lymphoid compartment distribution in TNF-α[low/int/high]

patients. Line and percentages denote patients above a third of the maximum value. **f** Representative IHC staining of TMA cores for TNF-α in deserted, intermediate, and reactive subTMEs. Scale bar 200 μm. Quantification for TNF-α per patient (**g**) or TMA core (**h**) is classified as deserted (des), intermediate (int), and reactive (rea). **g** des, $n = 41$ patients; int, $n = 45$ patients; rea, $n = 11$ patients. **h** des, $n = 120$ cores; int, $n = 134$ cores; rea, $n = 39$ cores. Boxplots show 25th to 75th percentile with median as box and highest and lowest value in 1.5 times interquartile range as whiskers. Two-tailed Wilcoxon rank sum test. Source data are provided as a Source Data file.

divergent expression and functions of the AP1 TFs in a reciprocal manner with the spatial TiME.

Altogether, this string of insights provides a potential mechanistic foundation for several recent studies that showed a high degree of heterogeneity in the neoplastic and stromal immune compartments in human PDAC, including hybrid/intermediate/co-expressor CLA/BL

subtype states that exist in naive and therapy-treated PDAC tumors[10–15,17,20,40,44]. We propose that extrinsic regional TNF-α plays an essential role in destabilizing CLA-like neoplastic state by promoting BL cJUN/AP1-mediated transcriptional programs. Compartment-specific transcriptomic profiling of TNF-α-treated tumors demonstrated that TNF-α strongly affects both neoplastic-specific as well as

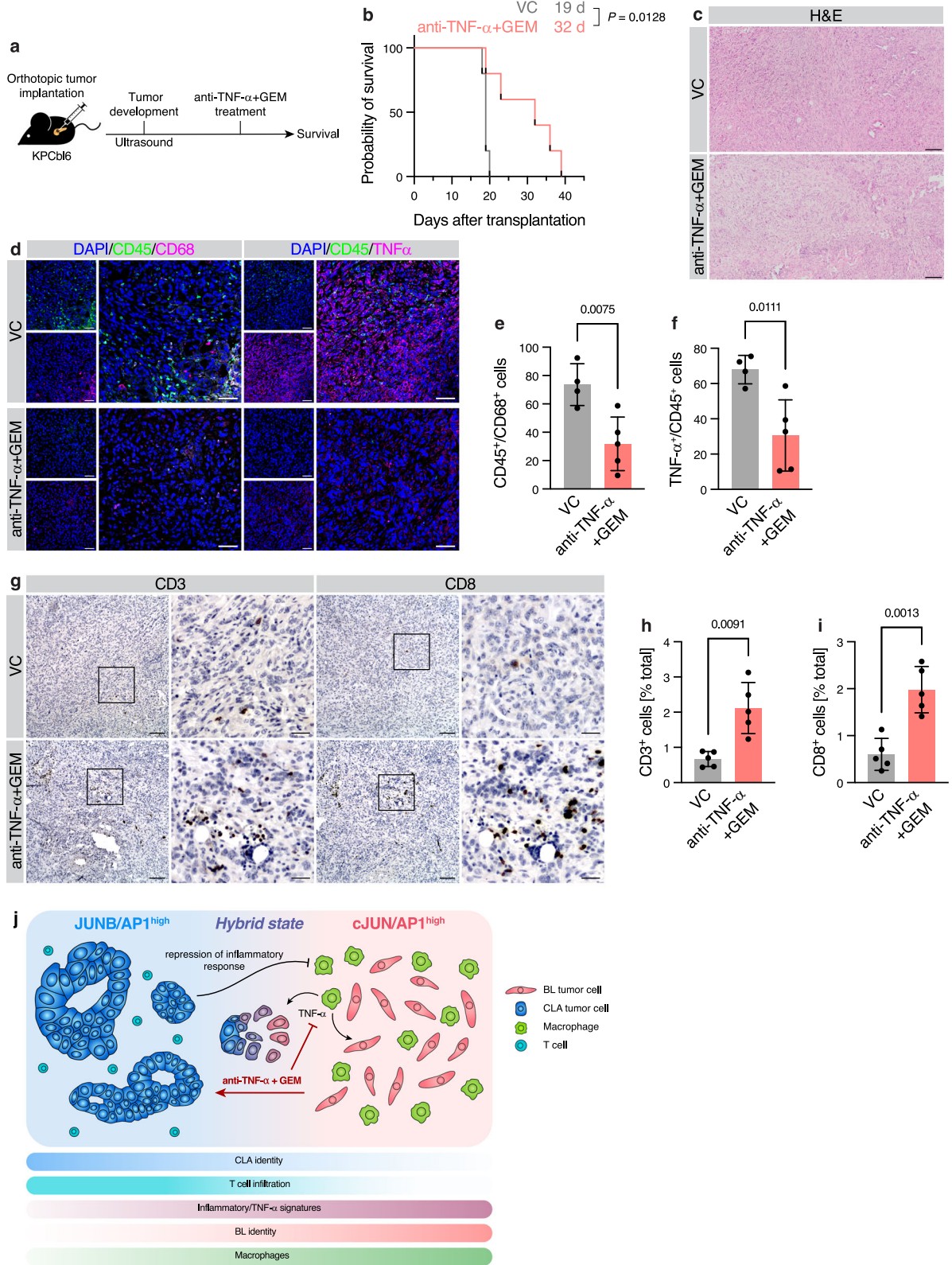

immunosuppressive stromal gene expression. In the neoplastic compartment, TNF-α treatment induced loss of CLA-like cell identity, whereas in the TME it resulted in depletion of T and B cell signatures. In line, in a large cohort of PDAC patient tumors, we observed a significant reduction of CD3[+], CD4[+], and CD8[+] T cell infiltrations particularly in TNF-α[high] tumors, whereas CD68[+] macrophages were strongly elevated, underlining how recruitment of inflammatory CD68[+]

macrophages to BL TNF-α[high] regions simultaneously leads to an immunosuppressive TME. Furthermore, we also found strongly enhanced TNF-α expression in reactive subTME regions[43], strengthening TNF-α as a major link between the TME and the BL inflammatory subtype, as reactive subTMEs provide the organizational framework for an inflamed, poorly differentiated and aggressive tissue state with CK5[high]/GATA6[low] neoplastic cells. To explore the therapeutic potential

**Fig. 7 | Targeting of TNF-α during chemotherapy restores anti-tumor immunity and prolongs survival. a** KPC cells were orthotopically implanted into syngeneic C57BL6/J mice and treated with an anti-TNF-α antibody in combination with gemcitabine (GEM) chemotherapy, or vehicle control (VC). **b** Kaplan-Meier survival analysis of **a**. Median survival indicated. Log-rank test. VC, $n = 5$ animals. **c** H&E staining of anti-TNF-α + GEM and VC tumors. Scale bar 100 μm. **d** IF for CD45 with CD68, and CD45 with TNF-α, in anti-TNF-α + GEM and VC tumors. Scale bar 50 μm. Quantification of (**d**) for CD45/CD68 (**e**) and TNF-α/CD45 (**f**) double-positive cells.

Per-animal average counts per FOV with mean ± s.d. shown. VC, $n = 4$ animals; anti-TNF-α + GEM, $n = 5$ animals. **g**, IHC for CD3 and CD8 in orthotopically transplanted anti-TNF-α + GEM and VC tumors. Scale bar: overview, 100 μm; insert, 30 μm. Quantification of (**g**) for CD3$^+$ (**h**) and CD8$^+$ (**i**) cells. Per-animal average percentage of positive cells with mean ± s.d. shown. $n = 5$ animals. **e**, **f**, **h**, **i** Two-tailed Student's $t$-test with Welch's correction. **j** Model of AP1 dichotomy in PDAC subtype co-existence and immune recruitment. Source data are provided as a Source Data file.

of targeting TNF-α, we treated a highly aggressive KPC tumor model with an anti-TNF-α monoclonal antibody plus GEM. Combined anti-TNF-α and chemotherapy substantially improved the survival, reduced the infiltration of CD45$^+$/CD68$^+$/TNF-α$^+$ macrophages, and induced recruitment of CD3$^+$ and CD8$^+$ T cells in the TME. This identifies TNF-α-specific functions that appear to determine an immunosuppressive regional TiME and PDAC aggressiveness. In accordance, the PRINCE trial suggests that higher CD4$^+$/8$^+$ T cell levels relate to a better response to immune checkpoint inhibition with chemotherapy. Elevated TNF-α signaling has a negative impact on therapy response in metastatic PDAC patients, emphasizing its role in immunosuppression[51]. Combining anti-TNF-α therapy with chemotherapy has shown significant benefits in metastatic lung cancer patients[52]. This approach is now being used with nivolumab and anti-TNF-α therapy in resectable lung cancer patients (NCT04991025). Future studies targeting TNF-α pathways combined with immunotherapies may offer important therapeutic options in PDAC therapy.

In sum, our comprehensive analysis of the dichotomous role of the AP1 TFs in PDAC subtype heterogeneity has shed light on the mechanism of JUNB/HDAC-dependent suppression of BL-associated inflammatory responses. JUNB signaling can be regionally repressed through macrophage-secreted TNF-α, destabilizing favorable CLA-like state and inducing T cell exclusion, which highlights the important role of the single cytokine TNF-α in shaping an immunosuppressive TiME and tumor aggressiveness. Thus, therapeutically shifting the balance from T cell$^{low}$/M2 macrophage$^{high}$/BL towards T cell$^{high}$/macrophage$^{low}$/CLA-like state through inhibition of TNF-α with GEM chemotherapy may provide a valuable strategy to enhance anti-tumor immunity and treatment response in PDAC.

## Methods
### Ethical approval
All experiments conducted in this study adhere to relevant ethical regulations. Specifically, all animal experiments were performed at the University Medical Center Göttingen (UMG) in accordance with the guidelines approved by the Central Animal Experimental Authority and its ethics review board (permission numbers: 15/2057, 14/1634, 18/2953). The ethics review board of UMG approved the generation of the PDX mouse model (permission no. 70112108). Regarding patient data, the resection material from the UMG patient cohort is derived from the Molecular Pancreatic Cancer Program (MolPAC) at UMG. This material is part of the study titled "Klinische und molekulare Evaluation von Patienten mit Pankreasraumforderungen im Rahmen des Pankreasprogramms der UMG (MolPAC)," which was approved by the ethics committee of UMG under approval number "11/5/17." For the Princess Margaret Cancer Centre (PMCC) cohort tumors were obtained from the UHN Biospecimens Program after collection at Princess Margaret Cancer Centre (Toronto, Canada) and have been previously discussed in studies[43,53,54]. All patients provided written informed consent for the molecular characterization of their tumor samples and for follow-up on their clinical information and were approved by University Health Network Research Ethics Board (case numbers 03-0049, 08-0767, 15-9596, 17-6106, 16-5380). The DKFZ cohort samples are part of the Department of General, Visceral and Transplantation Surgery, University of Heidelberg (HIPO-project approved by the ethical

committee of the University of Heidelberg; case number S-206/2011 and EPZ-Biobank Ethic Vote no. 301/2001).

Further characteristics and experimental procedures of patient cohorts are detailed hereafter.

### Preclinical animal experiments
Eight to ten-week-old male NMRI-*Foxn1*$^{nu/nu}$ or C57BL/6 J mice were used for orthotopic transplantation. $1 \times 10^6$ cells (CAPAN1, MiaPaCa2, CAPAN2-EV, or CAPAN2-cJUN) were transplanted orthotopically into NMRI-*Foxn1*$^{nu/nu}$ mice, or $3.5 \times 10^4$ KPC cells into syngeneic C57BL/6J mice, each in 15 μL culture medium under isoflurane anesthesia. Tumor growth was monitored by weekly ultrasound. The TNF-α treatment model in orthotopic CAPAN1 tumor model has been described previously[21]. In brief, mice were treated intraperitonially (i.p.) with TNF-α (0.4 mg kg$^{-1}$; 300-01 A; PeproTech) or control (aqua dest.) three times per week when tumor size reached ~300 mm$^3$, for a total of three weeks. For the anti-TNF-α plus gemcitabine treatment model, C57BL/6J mice bearing orthotopic KPC tumors were treated with 10 μg g$^{-1}$ anti-TNF-α antibody (BioLegend) in combination with 100 mg/kg gemcitabine (Sigma-Aldrich) three times a week for three weeks. Preclinical studies were terminated when mice displayed exclusion criteria, e.g., body weight loss >20%, tumor volume of >1 cm$^3$, or overall poor clinical presentation. Tissues were fixed in formalin and embedded in paraffin for histological analysis.

### Human PDAC tissue and tumor microarray IHC analysis
PDAC resection tissue utilized for IF staining of the UMG cohort included 32 chemotherapy-naïve, resected primary PDAC cases, with 23 female and 9 male patients, 5 stage I, 19 stage II, and 8 stage III, with a mean age at diagnosis of 72.2 years (age range 49–86 years). IF staining was performed as described below.

The PDAC tissue microarray (TMA) cohort from PMCC comprises resectable primary pancreatic tumor specimens from 105 treatment-naïve patients diagnosed with PDAC. These included 50 female and 65 male patients, 9 stage I and 96 stage II, with a mean age at diagnosis of 65.8 years (age range 42–84 years). Out of the 105 patients, specimens from a TMA were utilized in the study. To create the TMA, a pathologist identified the optimal area for coring reviewed sections from paraffin blocks. After marking the areas, 1.2 mm tissue cores were manually punched and transferred into recipient paraffin blocks. In addition to the tumor samples, cores of benign pancreatic, renal, pulmonary, and hepatic tissues were included for control purposes and to aid in TMA slide orientation. Multiple tumor cores were arrayed (two-to four-fold redundancy) from the paraffin blocks and spread across multiple slides for the final tissue microarray. Additionally, for a subset of $n = 98$ patients, tumor specimens were processed by Laser Capture Microdissection (LCM) to enable epithelial-enriched RNA-seq profiling. This analysis revealed that $n = 36$ tumors exhibited a basal-like profile, while $n = 62$ exhibited a classical profile, as determined using the clustering method described previously[5]. In addition to the stainings described in the original publication[43], IHC was performed for JUNB (Cell Signaling #3753) and TNF-α (abcam ab1793) manually according to standard laboratory procedures using antigen retrieval by Tris-EDTA buffer solution (pH 9.0). IHC staining was quantified using QuPath bioimage analysis software[55]. Individual TMA cores were registered, and patient

identifiers were superimposed for analysis. Malignant epithelial regions were manually annotated for JUNB, Cytokeratin 19, CDH1 and GATA6 expression (see also Supplementary Fig 1d), while the *Simple Tissue Detection* tool was used to select tissue for TNF-α expression analysis. Residual normal pancreas epithelium, nerves, large blood vessels, tissue folds, and stain artifact were manually excluded. Pixel-based detection parameters were set and optimized for each stain. "Positive pixel percentage" within the annotated epithelial regions was determined as the fraction of positive pixels within all detected pixels and as indicated, either averaged across all cores per patient (patient level). For display and statistical testing, samples were then stratified into groups ("high", "intermediate", "low") via the top, intermediate, and bottom 33% of the obtained JUNB and TNF-α positive pixel percentage values, respectively.

### Flow cytometry-sorted compartment RNA-seq

The compartment-sorted patient transcriptome data has been published previously[22]. In brief, tumor tissue of untreated patient with partial pancreatoduodenectomy at the Department of General, Visceral and Transplantation Surgery, University of Heidelberg was subjected to fluorescence-activated cell sorting with compartment-specific markers (for epithelial EPCAM+/CD45−/CD31−) and subsequently RNA-sequenced in the sorted fractions. This cohort included 31 PDAC cases, with 17 female and 14 male patients, 4 stage IIA, 23 stage IIB, 3 stage IV, and one without stage information available, with a mean age at diagnosis of 65.79 years (age range 41–83 years).

### Patient data analysis and study approval

For gene expression and survival analysis, publicly available datasets[4,17,56,57] were used. TCGA[57] and QCMG[4] datasets were accessed via cBioPortal[58,59], Puleo[56] dataset was accessed via the ArrayExpress database with accession number E-MTAB-6134b, the Zhou[17] dataset from Supplementary Tables of the original publication. For the TCGA cohort, only 150 curated PDAC cases as described previously[60] were retained for analysis. The expression values were plotted as z scores (TCGA), RMA-normalized probe intensities (of the highest-expressing probe for one gene; Puleo), log(RSEM) (QCMG), log2(TPM + 1) (compartment-sorted patients), or logR-normalized counts (Zhou). Survival data was available for 288 of the total 309 patients of the Puleo cohort, 85 of 97 patients for Zhou, and all 96 and 150 patients for QCMG and TCGA respectively.

R v4.2.0 was utilized for the analysis. For correlation analysis, Spearman's $R$ and $P$ value, as well as linear regression with 95% CI are indicated in the figures, plotted using the ggscatter function of the ggpubr package v0.5.0. GSEA analysis of compartment-sorted patient data was performed as described for RNA-seq data, using a gene list sorted by Spearman's $R$ for correlation of all genes to JUNB as input. JUNB repression signature scores were calculated using the GSVA package[61] v1.44.5 for expression data of each cohort individually. MCPcounter scores were determined with the MCPcounter package as above. For merging the MCPcounter scores of the different cohorts, scores were min-max normalized and standardized to the mean of the lower quartile of the JUNB repression signature scores. CIBERSORTx analysis[62] was performed using the CIBERSORTx website (https://cibersortx.stanford.edu) and cell fractions imputed using the LM22 signature matrix[63]. Patient survival analysis was performed using the survival v3.5-5 and ggsurvfit v0.3.0 packages.

### Cell culture

Established human PDAC cell lines CAPAN1, CAPAN2 and MiaPaCa2 were purchased from ATCC (Manassas, VA) and authenticated (CAPAN1, RRID:CVCL_0237), (CAPAN2, RRID:CVCL_0026), (MiaPa-Ca2:CVCL_0428) by the German Collection of Microorganisms and cell culture GmbH (DSMZ). PDAC cell lines were cultured in RPMI 1640 or DMEM (Thermo Fisher Scientific) with 10% (v/v) fetal calf serum (FCS;

Th. Geyer). CFPAC1 (RRID:CVCL_1119) and HPAF-II (RRID:CVCL_0313) PDAC cell lines were provided by Gioaccino Natoli (Humanitas University) and authenticated by the IEO Tissue Culture Facility. CFPAC-1 cells were maintained in IMDM with 10% FBS, while HPAF-II cells were maintained in Eagle's minimum essential medium with 10% FBS, 1 mM sodium pyruvate, and 1% non-essential amino acids. Murine cells derived from the *Kras*^G12D;*p53*^R172H;*Cre* mouse model ("KPC cells") were maintained in DMEM with 10% FCS and 1% non-essential amino acids. Patient-derived primary cell line GCDX62 was maintained in a 3:1 mixture of Keratinocyte-SFM (KSF; Thermo Fisher Scientific; supplemented with 2% (v/v) FCS, 1% (v/v) Penicillin-streptomycin, bovine pituitary extract (BPE), and human epidermal growth factor) and RPMI 1640 containing 10% (v/v) FCS. cJUN overexpression (cJUN-OE; construct pMSCV-cJUN, no. 34898, addgene) and empty vector (EV; construct MSCV, no. 68469, addgene) control cell lines of CAPAN2 and GCDX62 were generated previously[21], and were maintained in their normal growth medium supplemented with 1 µg/mL puromycin.

### siRNA transfection

For transient knockdown experiments, $5 \times 10^5$ cells were seeded in 6-well plates and immediately transfected with a mixture of 10 µL Lipofectamine2000 (Thermo Fisher Scientific), 6 µL of 20 µM target-specific siRNA (or non-targeting siRNA as control), and 200 µL Opti-MEM (Thermo Fisher Scientific) after 15 min incubation of the transfection mixture at room temperature (RT). Cell culture medium was changed after 24 h, and protein or RNA extracted 48-72 h after transfection.

### Immunoblotting

Cells were washed with phosphate-buffered saline (PBS) and lysed in whole cell lysis (WCL) buffer supplemented with cOmplete protease inhibitor cocktail (Roche Diagnostics), 100 µM sodium orthovanadate (NaO; Sigma-Aldrich) and 100 µM phenylmethylsulfonyl fluoride (PMSF; Sigma-Aldrich) for 30 min on ice. Cell lysates were centrifuged at $17,000 \times g$ for 20 min at 4 °C, and supernatants were collected. Proteins were diluted to 1 µg/µL in WCL and Laemmli buffer and boiled for 8 min at 95 °C. Denatured samples were resolved by SDS-PAGE on 10 or 15% gels and transferred onto nitrocellulose membranes. After blocking in 5% (w/v) milk powder in TRIS-buffered saline with 0.1% (v/v) Tween 20, primary antibodies were incubated overnight at 4 °C. Secondary HRP-linked antibodies were subsequently incubated at RT for 1 h. Next, membranes were developed with ECL solution using a ChemoStar imager (Intas Science Imaging Instruments). Antibodies are listed in Supplementary Table 1.

### Co-immunoprecipitation

Cells were seeded in 10 cm dishes and harvested by scraping in 1.5 mL ice-cold PBS. The cell suspension was centrifuged at $500 \times g$ for 5 min at 4 °C. The resulting pellet was resuspended in lysis buffer containing Triton X-100. After the cells were completely lysed, the lysates were centrifuged at $17,000 \times g$ for 20 min at 4 °C and the supernatant transferred into new tubes. 500 µg of protein were added to washed agarose protein A beads (Merck Millipore) and incubated on a rotating wheel for 1 h at 4 °C. Afterwards, beads were removed by centrifugation and precleared lysates were incubated with the target antibodies overnight at 4 °C. The following day, 50 µL of washed agarose A beads were added to the lysates and incubated on a rotating wheel for 2 h at 4 °C. The beads/antibody/target complexes were washed twice with WCL buffer and twice with PBS with cOmplete protease inhibitor cocktail. Finally, the complexes were resuspended in 65 µL 2x Laemmli buffer and incubated for 8 min at 95 °C. These samples were processed for immunoblotting as described above. To avoid the heavy chain signal of the pulldown antibody in case pulldown and primary immunoblot antibody were of the same species, an anti-light-chain antibody raised in another species was used prior to using the secondary antibody. Antibodies are listed in Supplementary Table 1.

## RNA isolation and quantitative real-time PCR

Total RNA was extracted using TRIzol reagent (Invitrogen) according to the manufacturer's protocol. Briefly, cells were washed with ice-cold PBS and collected in 800 μL TRIzol reagent, followed by addition of 200 μL chloroform after a short incubation. The solution was vortexed for 5 s to mix it thoroughly. After incubation at RT for 5 min, samples were centrifuged at 17,000 × g for 15 min at 4 °C. Next, the upper aqueous phase was transferred into a new 1.5 mL tube. Subsequently, 500 μL of isopropanol was added and mixed. After centrifugation at 17,000 × g for 30 min at 4 °C, the resulting pellet was washed twice with 75% ethanol. Finally, the dried pellet was dissolved in 30 μL of nuclease-free water. 1 μg RNA was used for cDNA synthesis using the iScript cDNA Synthesis Kit (BioRad) according to the manufacturer's instructions. 5 μL of SYBR green (BioRad) and 0.25 μL of forward and reverse primers each were mixed with 1 μL of cDNA. Quantitative real-time PCR (qRT-PCR) was performed using the StepOnePlus Real-Time System (Applied Biosystems). Relative quantification values were calculated with the associated StepOnePlus software, using *XS13* as reference control gene. Primer sequences are listed in Supplementary Table 2.

## Chromatin immunoprecipitation

For chromatin immunoprecipitation followed by quantitative real-time PCR (ChIP-qPCR), cells were seeded in 10 cm culture dishes and fixed in 1% PFA (Thermo Fisher Scientific) in PBS for 20 min at RT. After quenching with 1.25 M glycine, cells were washed in PBS and scraped in 1.5 mL cold Nelson buffer with cOmplete protease inhibitor cocktail, 100 μM NaO and PMSF, and 10 mM sodium fluoride. Following centrifugation at 12,000 × g for 2 min at 4 °C and washing with Nelson buffer with inhibitors, nuclear pellets were snap-frozen in liquid nitrogen. Nuclear pellets were lyzed in Gomes lysis buffer with cOmplete protease inhibitor cocktail, 100 μM NaO and PMSF, and 10 mM sodium fluoride with 0.1% SDS (for JUNB, HDAC1) or 0.5% SDS (for H3K27ac). After lysis for 15 min at 4 °C, cells were sonicated on a Bioruptor Pico (Diagenode) for 3–8 cycles with 30 s ON/OFF. Sonication efficiency was validated by agarose gel electrophoresis. Thereafter, lysates were pre-cleared with 15 μL washed protein A/G magnetic beads (Thermo Fisher Scientific) and then incubated overnight with the primary pulldown antibodies or isotype control (Supplementary Table 1). For pulldown of antibody-antigen complexes, 30 μL of washed beads are added to lysates and incubated for 2 h at 4 °C. Finally, lysates are washed with Gomes lysis buffer, Gomes wash buffer, and TE buffer. Corresponding input and pulldown samples are then RNase A and proteinase K digested. DNA was isolated by phenol-chloroform-isoamyl alcohol method. RT-qPCR was performed as described above, increasing total PCR cycles to 55. Primer sequences are listed in Supplementary Table 2.

## RNA-seq and ChIP-seq analysis

For RNA-seq, library preparation and sequencing for CAPAN1 cells subjected to JUNB silencing (siJUNB) or non-targeting control (siCtrl) were performed as described previously[21]. In brief, cDNA libraries were made using 500 ng total RNA and TruSeq RNA Library Prep kits (Illumina; RS-122-2001/2). cDNA concentration was measured using Qubit (Thermo Fisher; Q32854) and fragment sizes were confirmed by Bioanalyzer (Agilent; 5067-4626). Single-end, 50 bp sequencing was performed on a HiSeq2000 (Illumina) at the next-generation sequencing (NGS) Integrative Genomics Unit at UMG. For comparability, previously published RNA-seq data of TNF-α-treated CAPAN1 cells was analyzed analogously using the following pipeline.

Raw reads were quality-checked using FastQC, aligned to hg38 reference genome using STAR[64] v2.7.3a and counted per gene using htseq-count[65] v0.11.3. Downstream analysis was conducted in R v4.2.0 and Bioconductor[66] packages v3.15. Variance stabilization was performed using RUVSeq[67] v1.30.0 function RUVs. Differential gene expression was performed using DESeq2[68] v1.36.0. Gene set enrichment analysis[69] (GSEA) was conducted with clusterProfiler[70] v4.4.4, using signatures of the Molecular Signature Database[71], as well as custom signatures based on published PDAC subtype classifications[4,5,12,72]. Expression heatmaps were created by the pheatmap package v1.0.12.

For tissue RNA of TNF-α or VC-treated tumors of orthotopically implanted CAPAN1 cells, RNA was purified using Direct-ZOL RNA Miniprep (Zymo Research). Library preparation and sequencing (paired-end, 150 bp read length) were performed by Novogene. To allow deconvolution of human (tumor cell) and murine (host stroma) compartments, reads were aligned using STAR v2.7.3a to human reference hg38, as well as murine reference mm39. Subsequently, the XenofilteR[73] package v1.6 in R v4.1.0 was used to remove either murine reads from the human alignment or vice versa. After filtering, samples were processed as above.

To estimate the relative abundance of stromal cells, murine reads were processed with the MCPcounter[74] package v1.2.0 in R v4.2.0.

JUNB and cJUN ChIP-seq, as well as ATAC-seq, were conducted previously[21], accessible at GSE179781. Further, previously published RNA-seq data for GCDX62-cJUN-OE and GCDX62-EV was utilized, which is accessible at GSE173121. For meta-pathway analysis of JUNB-bound and accessible regions, genes were annotated to JUNB-bound and ATAC-seq peak regions using GREAT and analyzed using Metascape[75] (https://metascape.org/). For integration of RNA- and ChIP-seq data, consensus ChIP-seq peaks were annotated to genes using the R package rGREAT[76] v3.0.0, and their fold change in the corresponding RNA-seq extracted. Gene ontology (GO) analysis was then performed for the significantly up- or downregulated genes of this subset using the clusterProfiler package as above. The list of JUNB-bound, significantly upregulated genes in CAPAN1 is available in Supplementary Table 3. The overlap of JUNB-bound genes (as per rGREAT) with genes that are significantly upregulated upon siJUNB in the RNA-seq were subsequently analyzed for their correlation to JUNB in 46 pancreatic cancer cell lines of the CCLE[36,37] dataset, which was downloaded from cBioPortal. Those genes that had an inverse association with JUNB (negative Spearman's R) were termed the JUNB repression signature; the full list is available in Supplementary Table 4.

For the ChIP-seq for HDAC1 and H3K27ac in CAPAN1 cells with siJUNB or siCtrl, ChIP has been performed as detailed above. DNA libraries for sequencing were constructed using Kapa Hyper Prep (KR0961) and double size-selected (Nucleomags, Macherey-Nagel). Library quality was controlled using the Agilent DNA High Sensitivity DNA Kit and Agilent Bioanalyzer 2100. After concentration measurement using Qubit dsDNA HS Assay, libraries were multiplexed to 10 nM in EB buffer. Libraries were sequenced by BGI using a DNBSEQ-G400 to 100 bp paired-end. One input control sample per condition was performed and inputs for siJUNB and siCtrl were merged. Raw reads were quality-checked using FastQC, adapters trimmed using Trimmomatic v0.36, aligned to hg38 reference genome using bowtie2 v2.3.4.1, low-quality alignments, not properly paired reads, and chrM filtered using samtools v1.9, blacklist regions removed using bedtools v2.29.1, and duplicates removed using PICARD v 2.20.2. Peaks were then called using MACS2 v2.1.2. Differential binding sites were determined using DiffBind v3.4.11 in R v4.1.0. Subsequent downstream analysis, including peak overlap using ChIPpeakAnno v3.36.1, as well as gene ontology analysis using clusterProfiler v4.10.1, was performed in R v4.3.2.

## Reporter assay

For reporter assays, the Dual-Luciferase Reporter Assay System (Promega) was used. $5 \times 10^4$ cells were seeded into 24-well plates. The following day, cells were transfected with 200 ng of pJC6-GL3 (#11979, Addgene; cJUN), which contains the cJUN promoter linked to the firefly luciferase gene. Additionally, cells were co-transfected with different concentrations of JUNB-overexpressing plasmids (Paul Dobner,

University of Massachusetts), 15 ng of pRL-null *Renilla* luciferase control reporter (#E2271, Promega; for background normalization), and compensated with empty vector control (pCMV2c, David Russell, UTSW) to ensure equal total amounts of transfected plasmids. Reporter, overexpression, *Renilla* luciferase and empty vector plasmid mixture was added to Lipofectamine 2000 (Thermo Fisher Scientific) and 50 μL of Opti-MEM (Thermo Fisher Scientific) and incubated at RT for 10 min. Thereafter, 50 μL of the transfection mixture was added to each well for 24 h. Before the measurement, cells were lysed with 1x Passive Lysis Buffer (Promega) for 10 min on a shaker at RT. Subsequently, 30 μL of the cell lysate was transferred into a white 96-well plate and 30 μL of firefly luciferase substrate (Promega) was added. The luminescence was measured using a LUmo microplate reader (Autobio Diagnostics). Next, 30 μL of the Stop & Glo reagent (Promega) was added to the previous solution and luminescence measured again for the *Renilla* signal. Firefly luciferase luminescence signal was finally divided by the *Renilla* luciferase luminescence and normalized to the control sample.

### Transwell invasion assay

For invasion assays of silenced cells, siRNA transfection was performed 24 h before as described above. 8 μm porous cell culture inserts (Falcon) were coated in type I collagen (Enzo; diluted 1:62.5 in 0.1 M HCl) for 2 h. Then, silenced or control cells were seeded in 50 μL Matrigel solution (Corning) and solidified for 30 min before adding normal culture medium to the inserts and wells. After 48 h incubation, medium was aspirated, Matrigel removed, and membranes fixed in 4% PFA. After washing with PBS, membranes were stained with DAPI for 1 min and finally mounted on glass slides in Immu-Mount (Thermo Fisher Scientific). Invaded cells were imaged using a DMi8 fluorescence microscope (Leica) and quantified manually using ImageJ Fiji[77]. For each biological replicate, two independent inserts were evaluated.

### Hematoxylin and eosin staining

Hematoxylin and eosin (H&E) staining was performed as follows. Formalin-fixed paraffin-embedded tissues were cut into 4 μm thin sections. Tissue sections were incubated in xylene for 1 h and rehydrated in decreasing concentrations of ethanol and finally water. Then, sections were stained with hematoxylin for 8 min, followed by bluing for 7 min under running tap water. Subsequently, sections were briefly incubated in mild acetic acid solution and transferred into eosin/acetic acid solution for 3 min. Lastly, slides were dehydrated in an increasing ethanol series and mounted with Roti-Mount (Carl Roth).

### Immunofluorescence and immunohistochemical staining

Immunofluorescence (IF) staining was performed as follows. Slides were deparaffinized in xylene and rehydrated followed by antigen retrieval by boiling in citrate buffer (pH 6.0). Sections were blocked in 1% (w/v) bovine serum albumin (BSA; Sigma) in phosphate buffer (PB) containing 0.4% Triton X-100. After washing five times with PB, sections were incubated with primary antibodies at 4 °C overnight. After six PB washes, fluorophore-coupled secondary antibodies were incubated at 4 °C for 2 h. Subsequently, sections were washed in PB, stained with DAPI, and mounted in Immu-Mount (Thermo Fisher Scientific).

For immunohistochemistry (IHC) staining, the VECTASTAIN ABC-HRP Kit and ImmPACT DAB Substrate Kit (Vector Laboratories) was used. Deparaffinization and antigen retrieval were performed as above. Then, slides were fixed on a slide holder and incubated with 3% hydrogen peroxide solution for 10 min prior to blocking in 10% BSA. Next, the slides were incubated with primary antibodies overnight at 4 °C. The next day, slides were washed three times with PBS with 0.1% Tween20 (Sigma; PBST) and incubated with secondary antibodies for 1 h followed by AB complex incubation. Afterward, slides were washed with PBST and stained with DAB solution and further placed in deionized water to stop the reaction. Slides were then stained with hematoxylin for 7 min and blued under running water. Lastly, the slides were dehydrated and fixed as described for H&E staining.

### Image acquisition and analysis

For bright-field applications (H&E, IHC), images were acquired using either an Olympus BX43 light microscope or the Olympus VS120 virtual slide microscope. IF staining was imaged using either an Olympus IX81 confocal fluorescence microscope, or the Olympus VS120 virtual slide microscope.

Quantification of IF and IHC images was performed either using ImageJ Fiji[77] (by manually counting positive cells, measuring the fluorescence intensity of the entire field of view, or using semi-automatic macros[78]), or using QuPath[55] v0.4.3. In QuPath, cell detection was performed using the built-in detection feature, obtaining intensity measurements per cell, per nucleus or per cytoplasm, which were used for positive cell quantification using single measurement classifications (or compound classifiers for double-positive cells). For IHC images, positive cell detection using optical density for nuclei detection and deconvoluted DAB intensity for positive cells was used.

For hotspot analysis of HA-cJUN IHC and JUNB IF, density maps of positive cells were created and above a threshold defined as high-density areas ("hotspots"), which were then transferred to the respective other slide image to derive positive cells within cJUN or JUNB hotspot regions. For distance analysis of hotspots to CD68+ cells, cell detection was performed as above, the hotspot regions transferred to the CD68 staining image, and the 2D distance to annotation tool of QuPath used.

### Statistics and reproducibility

GraphPad Prism version 8.0.2 as well as R above version 4 were used for statistical analysis. The comparison of independent groups was performed using an unpaired Student's *t*-test with Welch's correction. One-way analysis of variance (ANOVA) was used for more than three conditions of a single factor. Survival data were analyzed using the log-rank test. Results were considered significant with a *P* value below 0.05, as indicated in the figures. Spearman correlation coefficient with a two-tailed *P* value was used for the correlation data. No statistical method was used to predetermine the sample size. For animal experiments, mice were randomized to implanted cell lines (if applicable) as well as to treatment groups (if applicable). The Investigators were not blinded to allocation during experiments and outcome assessment. For sequencing analyses, samples were removed if they did not pass overall quality control assessment. No other data was excluded from the analysis.

### Reporting summary

Further information on research design is available in the Nature Portfolio Reporting Summary linked to this article.

## Data availability

For this study, the Molecular Signatures Database (https://www.gsea-msigdb.org/gsea/msigdb/) database was used. Previously published[21,25] ChIP- and ATAC-seq data are available at Gene Expression Omnibus (GEO) under accession codes GSE173159 and GSE64560. ChIP- and RNA-seq data generated in this study has been deposited at GEO with the accession code GSE276324. FACS-sorted epithelial patient tumor RNA-seq data[22] are available at EGA under accession code EGAS00001004660. Access is restricted due to patient data protection but can be made available through the DKFZ-HIPO Data Access Committee of Heidelberg Center for Personalized Oncology (hipo_daco@dkfz-heidelberg.de) using the Data Access Request via the EGA DAC Portal. Patient tumor LCM-enriched transcriptome and proteome data are available at EGA under accession code EGAS00001002543 and from UCSD's MASSive database under accession code MSV000086812 [https://massive.ucsd.edu/

ProteoSAFe/dataset.jsp?task=
35d2ed0cbcd04045adceeb4866e478a3]. Data access can be
requested from the PanCuRx Translational Research Initiative using
the OICR Data Access Agreement as detailed on EGA. Processed
proteome data is detailed in the previous publication[43] Table S7
[https://ars.els-cdn.com/content/image/1-s2.0-S0092867421011053-
mmc6.xlsx]. Source data are provided with this paper.

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

## Acknowledgements

This study was supported by the Deutsche Krebshilfe (70112999 and 70115054; Max-Eder Program), the Fritz-Thyssen Stiftung (10.20.2.038MN and 10.23.2.021MN) and the Wilhelm-Sander-Stiftung (2021.159.1 and 2021.159.2) to S.K.S. This study was also supported by KFO 5002 grant of the DFG to E.H., A.P., V.E., and S.K.S. N.K. received funding from the DFG (413501650) and an UAEU Start up UPAR grant (12F043) awarded to UK for this study. The processing of human PDAC patient data was made possible by the Dietmar-Hopp Foundation and the BioRNSpitzen cluster 'Molecular- and Cell-based Medicine'; the German Ministry of Science and Education (BMBF) e:Med program for systems biology (PANC-STRAT consortium, grant no. 01ZX1305C and 01ZX1605C). The PancoBank collection and processing of human specimens were supported by Heidelberger Stiftung Chirurgie. We thank E. Sidhu, L. Schüürhuis, K. Reutlinger, S. Mercan, and W. Kopp for their technical assistance, as well as J. Todorovic for help with the UMG patient cohort. We sincerely thank the EPZ-Biobank and Department of General and Visceral Surgery of the University Hospital Heidelberg, particularly N.A. Giese, T. Hackert, O. Strobel, and M. Büchler for their collaboration involving fresh primary PDAC specimens.

## Author contributions

L.K., M.T., and S.K.S. designed the overall study. S.K.S and L.K. wrote the manuscript. M.T., B.T.G. and U.K. edited the manuscript. M.T. performed preclinical studies. M.T., L.K., D.G., M.U.L., S.K., and F.P. analyzed tissue of patients and preclinical models. M.T., L.U., X.W., R.D.S., and F.P. performed in vitro experiments. A.B. and F.W. performed ChIP experiments, L.K. performed bioinformatic analysis of ChIP-, ATAC-, and RNA-seq data from in vitro and in vivo models, as well as analysis of patient expression data, with AP aiding ChIP- and ATAC-seq data analysis. E.E. performed RNA-seq of flow cytometry-sorted patient specimens, E.E. and L.K. analyzed the data, with E.E. and A.T. aiding in data

interpretation. E.H. provided human PDX and patient specimens. S.K.S., E.H., and V.E. critically analyzed murine and human histopathological examinations and data interpretation. N.K., N.C., K.A., F.V., and B.T.G. performed patient IHC analysis. B.T.G. and R.K. helped with experimental design and data interpretation. SKS supervised the project and interpreted the data.

## Funding

## Competing interests
The authors declare no competing interests.

## Additional information

[1]Department of Gastroenterology, Gastrointestinal Oncology and Endocrinology, University Medical Center Göttingen, Göttingen, Germany. [2]Department of Medical Oncology, West German Cancer Center, University Hospital Essen, Essen, Germany. [3]Department of Gynecology and Obstetrics, University Medical Center Göttingen, Göttingen, Germany. [4]Institute of Pathology, University Medical Center Göttingen, Göttingen, Germany. [5]Princess Margaret Cancer Centre, University Health Network, Toronto, ON, Canada. [6]Department of Medical Biophysics, University of Toronto, Toronto, ON, Canada. [7]Department of Veterinary Medicine and Zayed Center of Health Sciences, United Arab Emirates University, Al Ain, UAE. [8]Clinical Research Unit KFO5002, University Medical Center Göttingen, Göttingen, Germany. [9]Comprehensive Cancer Center, Lower Saxony, Göttingen and Hannover, Germany. [10]Division of Stem Cells and Cancer, DKFZ, Heidelberg, Germany. [11]HI-STEM—Heidelberg Institute for Stem Cell Technology and Experimental Medicine gGmbH, Heidelberg, Germany. [12]Department of Pathology and Experimental Therapy, School of Medicine, University of Barcelona, L'Hospitalet de Llobregat, Barcelona, Spain. [13]Molecular Mechanisms and Experimental Therapy in Oncology Program (Oncobell), Institut d'Investigació Biomèdica de Bellvitge, L'Hospitalet de Llobregat, Barcelona, Spain. [14]Department of Urology, West German Cancer Center, University Hospital Essen, Essen, Germany. [15]These authors jointly supervised this work: Barbara T. Grünwald, Shiv K. Singh. ✉e-mail: shiv.singh@med.uni-goettingen.de

