## [Peer Review File · Nature Communications]

REVIEWER COMMENTS

Reviewer #1 (Remarks to the Author): Expert in pancreatic cancer genomics and therapy

Klein et al. report a novel spatially regulated dichotomy in the AP1 transcriptional programs (JUNB vs. cJUN) that shapes regional subtype identity in PDAC via combining preclinical models, multi-center clinical, bulk and compartment-specific transcriptomic, proteomic, and bioimaging data from human specimens. Specially, they identified the TNF α -mediated inflammatory response associated with low JUNB expression destabilizes CLA neoplastic identity by promoting a CLA-to-BL transition in PDAC. Certainly this is a great resource for the community, but the insights drawn from this data are a bit overstated given that there is no comprehensive support for the findings. I really believe that the authors need to clarify the following points.

Major points:

1. It is unclear which samples are being analyzed in each results section. Maybe there can be better clarity of which samples were analyzed with what and how they are integrated/related to each other.
2. In Fig 1e, the authors indicated that GATA6 expression was higher in JUNB^{high} than JUNB^{low} regions within the same patient with patient-paired analysis. It is better to provide the representative whole slide images of IHC for GATA6 and JUNB in serial sections.
3. In Fig 3, the authors hypothesized that HDACs may be involved in JUNB-mediated transcriptional repression of inflammatory BL-specific lineage signatures. However, the authors only select HDAC1 for further analysis. Please clarify it. In addition, the authors only analyzed HDAC1 cooperates with JUNB in transcriptional repression, it is hard to conclude that JUNB restricts BL pro-inflammatory programs via HDAC-mediated repression.
4. In Fig 4, the authors observed a trend towards higher immune infiltrations in cJUN-OE CLA-derived tumors. Why the author directly focuses on the changes of macrophages although the authors indicated the relationship between TNF α and macrophages with references. It is better to observe the changes of multiple immune cells in the microenvironment, and further changes in macrophages are reasonable. Further in Fig 5, the relationship between cJUN and TNF was introduced by references, without any experimental data support.
5. In Fig.5g, after TNF α treatment, both T cells and B cells were decreased in PDAC patients with a high expression of the JUNB repression signature genes, the authors only observed the change of T cells, how about the B cells?
6. The expression of JUNB and AP1 in PDAC patients with chemotherapy?

Minor points:

1. In line 68-69: Currently, the 5-year survival rate for pancreatic cancer is 12%.
2. In line 164, should be fig.1f-h, not extended data.
3. In Line 388 and 391, the abbreviation of gemcitabine (GEM) should appear once.

Reviewer #2 (Remarks to the Author): Expert in pancreatic cancer functional genomics, therapy,

preclinical models, and epigenetics; co-reviewed with Reviewer #3

In the manuscript by Klein and Tu et al., titled "Spatial Tumor Immune Heterogeneity Facilitates Subtype Co-existence and Therapy Response via AP1 Dichotomy in Pancreatic Cancer," the authors have conducted a comprehensive investigation using preclinical models, multi-center clinical data, and diverse omics analyses to explore the interplay between neoplastic intrinsic AP1 transcription factors and extrinsic CD68+ macrophages in pancreatic ductal adenocarcinoma (PDAC). Their analyses encompassed ATAC-seq, ChIP-seq, and RNA-seq, revealing JUNB/AP1- and HDAC-mediated epigenetic programs influencing pro-inflammatory immune signatures in tumor cells. This dichotomy in AP1 signaling was identified as a potential driver of intratumoral subtype co-existence and immunosuppressive tumor microenvironment with T cell exclusion.

The study delved into the role of TNF- α + macrophages and their association with a reactive phenotype, impacting CD8+ T cell infiltration. Importantly, the authors suggest that combining anti-TNF- α immunotherapy and chemotherapy could potentially reduce macrophage counts and promote CD3+/CD8+ T cell infiltration, leading to improved survival in preclinical models. These findings propose a reciprocal interdependence between neoplastic and microenvironmental factors in shaping PDAC heterogeneity and aggressiveness. However, despite the authors' efforts to organize a coherent rationale, the manuscript is highly disorganized and lacks data-driven reasoning. Additionally, most results are not novel and, in some cases, contradictory with the current literature without sufficient justification for this discrepancy. Moreover, the results do not support most of the conclusions and are redundant in the classical/basal-like subtypes. Lastly, the in vitro and in vivo experiments lack coherence mainly due to the constant shift in cell types depending on the result.

Comments:

1. It is unclear why the authors focused their analysis on JUNB and cJUN without a data-driven approach. The authors must clarify.
2. The authors argue that JUNB is related to the classical subtype and cited two references. However, JUNB's function is associated with proteins with which it interacts and post-translational modifications. Therefore, it could promote a differentiation state or EMT (doi: 10.1083/jcb.201109045, doi:10.1186/s13059-022-02800-0). The authors must clarify this point and define the specific protein complex associated with JUNB and the classical subtype.
3. The authors performed most of their experiments using CAPAN 1 and 2 and defined these cells as classical. However, they did not specify the criteria or methodology used to subtype the cells. Additionally, restricting the analysis to only two cell lines could generate bias in the interpretation of the results, mostly because CAPAN1 has alterations in BRCA2, representing a low proportion of PDAC patients (5%). The authors must include other types of cells such as BXPC3 in order to confirm their observations regarding JUNB and the classical phenotype.
4. The authors concluded that GATA6 and JUNB have a positive correlation. However, the results were not significant. Therefore, such affirmation is not accurate. The authors must correct this point and show in the manuscript the correlation coefficient and the P-value for all cohorts when comparing GATA6 and JUNB. They must underline the fact that the correlations were not significant.
5. The cohort of 105 patients from the TMA must be further characterized in the manuscript. Specifically, the source (metastatic, resectable), treatment status at the time of the biopsy, and the tumor subtype. For the TMA, the authors need to clarify for each picture the % of epithelial and stroma cells and indicate which cells are considered positive. Additionally, defining the criteria for JUNB high and low (method and

cutoff) is necessary. Lastly, the authors must report the number of cores per patient.

6. The sources of each set of patients must be included in the manuscript (105 TMA, 23 IF...). Also, clinical data concerning the set of patients is necessary in supplementary data to fully understand the presented results.

7. The authors need to add more information about the FACS experiment. How was the data processed to select the epithelial cell population?

8. The authors claim that CLA showed the highest expression of JUNB, however BL-B displays similar levels. Performing additional statistics comparing CLA with the other subtypes is required.

9. The authors must clarify the number of samples assessed. Additionally, using only 4 samples per group with the observed variations could generate biases in the conclusions. The authors must include additional samples per each group. Lastly, the authors claim that classical cells are sensitive to chemotherapy. This is not an original observation; other authors have pointed out this fact (doi: 10.1158/1078-0432.CCR-19-146).

10. The authors must clarify in the ChIP-seq experiment the cells used. Additionally, integrating publicly available data with novel data, even if they come from the same cell, could generate biases in the analysis associated with a batch effect. The authors must describe their approach to solve this point.

11. Defining the meaning of "potential downstream enhancer of GATA6" in the ChIP-seq experiment is essential

12. The authors performed siRNA on JUNB and argued that the silencing of this gene promotes an invasive state. However, they did not perform functional analysis of the cells with and without siRNA to validate that affirmation.

13. The authors used the full TCGA cohort (177 patients); however, this cohort contains normal, inflamed pancreas, and non-PDAC neoplasms (doi: 10.3390/cancers11010126). It is required that the authors elucidate this point.

14. The inferences between macrophage infiltration and the levels of JUNB are unclear. The authors claim that patients with high JUNB (Classical) display low levels of macrophages, whereas those with low JUNB (basal-like) display high levels of macrophages. This conclusion is barely supported by TCGA, and the other cohorts do not display the same patterns. The different cohorts are not comparable on this subject due to lack of information concerning the scale or the different types of clusters. Additionally, the type of macrophages must be evaluated. They must include more than one deconvolution algorithm.

15. The authors need to illustrate how the JUNB repression signature was generated.

16. Separating the results from TCGA, QCMG, and Puleo when analyzing the overall survival, DFS and JUNB repression signature (5.g) is fundamental. Also, include the number at risk and which P-value (log-rank test?) is reported in the KM. Additionally, the authors must perform a multivariate Cox analysis with clinical variables per cohort. Lastly, they must demonstrate the novel contribution of their JUNB-based classification compared to the other transcriptomic subtypes, such as PurIST.

17. The authors performed the siRNA experiment only in CAPAN1. They must expand the panel of cells to evaluate the extent of their conclusions. Additionally, the authors must be consistent in the panel of cells in each experiment. The authors shift from CAPAN1 (siRNA) to CAPAN2 (luciferase and in vivo) depending on the experiment.

18. Why did the authors solely focus on HDAC1 and no other HDACs, such as HDAC3, which main target is H3k27ac. A pathway enrichment analysis is not enough to justify an experimental decision, mainly because this analysis pointed out other potential targets.

19. It is imperative for the authors to clarify how the overexpression of cJUN "overexpressed HA-tagged

cJUN (cJUN-OE) and then orthotopically implanted empty vector control (EV) or cJUN-OE cells” generates a CLA tumor (JUNB high) in the in vivo experiments. It is unclear in the proposed mechanism.

20. The authors must describe the histological differences between cJUN overexpressed tumors and the control.

21. In figure 4, the authors must clarify the type of macrophage detected.

22. In figure 5d, the authors seem to have inverted TNF- α and VC for the GSEA of CLA related gene sets.

23. The authors used TNF- α to modulate PDAC phenotype. They found that TNF- α induces EMT and basal-like phenotype. These observations have already been published (doi: 10.1158/0008-5472.CAN-07-5704). Lastly, they must define the type of mice used in each experiment in the main text.

Additionally, it is crucial to illustrate how these results are related to JUNB results and macrophage recruitment.

24. The authors must include in the anti-TNF α + Gem experiments, the anti-TNF α and GEM alone controls to evaluate the actual synergy of the combinatory treatment.

Reviewer #3 (Remarks to the Author): Expert in pancreatic cancer functional genomics, therapy, preclinical models, and epigenetics; co-reviewed with Reviewer #2

Reviewer #4 (Remarks to the Author): Expert in cancer multi-omics, bioinformatics, and pancreatic cancer

Review Report: Spatial Tumor Immune Heterogeneity Facilitates Subtype Co-existence and Therapy Response via AP1 Dichotomy in Pancreatic Cancer

Summary:

This manuscript investigates the role of AP1 transcription factor dichotomy (JUNB vs. cJUN) in driving intratumoral heterogeneity in pancreatic ductal adenocarcinoma (PDAC). The study combines preclinical models, clinical data, and multi-omics analyses to explore how AP1 factors, alongside CD68+ macrophages, influence tumour subtype co-existence and response to therapy. The findings suggest that JUNB/AP1- and HDAC-mediated epigenetic programs suppress pro-inflammatory immune signatures, favouring a therapy-responsive classical subtype, while CD68+/TNF- α + macrophages promote a basal-like PDAC phenotype.

Having reviewed the paper three times, I appreciate the authors' thorough analysis and report on the AP1 dichotomy's role in driving intratumoral heterogeneity in PDAC. Each review has deepened my understanding of their comprehensive work. While I have comments and suggestions, I believe they are

more suited for future research directions. The current study stands as a complete and substantial contribution in its field, and it would be inappropriate (borderline insulting) for me to recommend additional experiments for this already comprehensive study.

Major Comments:

1. **Comparative Analysis:** The work would significantly benefit from expanding its comparative analysis to include the role of AP1 in relation to other pivotal transcription factors and epigenetic mechanisms known to influence intratumoural heterogeneity in PDAC, such as NF- κ B, STAT3, HIF-1 α , EZH2, and DNMTs. This broader examination would provide valuable context, situating the study within the extensive landscape of cancer biology. Could the authors discuss on this to offer readers a more comprehensive understanding of the intricate regulatory networks driving PDAC heterogeneity and potentially unveil new therapeutic targets or biomarkers.

2. **Enhancement of Methodological Transparency:** The integration of diverse data types is commendably robust; however, detailing the selection criteria for clinical samples and ensuring the reproducibility of bioinformatics analyses would substantially strengthen the manuscript. The authors are encouraged to deposit pre-processed data into a recognised repository and share the code used for generating key figures and findings (where necessary). This step would not only bolster the study's credibility but also facilitate future research by allowing others to validate and extend the work presented.

3. **Therapeutic Implications:** The discussion on potential therapeutic strategies targeting the AP1 dichotomy and macrophage signalling pathways is intriguing. Could the authors expand this discussion on how these findings could be translated into clinical trials or drug development would be valuable.

4. **Statistical Rigour:** Expanding on the "Statistical analysis" and statistical methods used to analyse the data would enhance the manuscript's methodological rigour. This could also include stating the approach utilised for multiple comparison whenever possible.

Other suggestions

1. **Experimental Validation:** Include additional experimental validation of the key findings, particularly the functional roles of JUNB and cJUN in PDAC. This could involve using CRISPR/Cas9 for gene editing to directly assess the impact of these transcription factors on tumor phenotype and immune response.

2. **Patient Diversity:** Expand the dataset to include a wider range of PDAC patients, considering variables such as disease stage, treatment history, and genetic background. This diversity could help validate the findings across a broader patient population.

Reviewer #5 (Remarks to the Author): Expert in tumour immune microenvironment, immunogenomics, immunology, macrophages, and pancreatic cancer; co-reviewed with Reviewer #6

In this study, Klein et al. investigate the mechanisms underlying intratumoral subtype heterogeneity in pancreatic ductal adenocarcinoma. Combining different data on human PDAC specimens, they identify

the dichotomy of AP1 transcription factors and macrophage regionalization as neoplastic cell-intrinsic and -extrinsic drivers of PDAC subtype heterogeneity respectively.

PDAC is a highly heterogeneous disease. A comprehensive description of intratumoral heterogeneity of PDAC subtypes and its impact on clinical outcome is still lacking, thus this paper is relevant to the field. The manuscript is well-written and combines valuable -omics data to explore the role of AP1 TFs and macrophages in shaping PDAC subtype heterogeneity. Unfortunately, some major concerns need to be addressed prior to recommending this study for publication, as there are key findings poorly supported by the data.

Major comments:

1. The authors associate JUNB with classical subtype identity in PDAC patients. However, clear and significant data proving expression of JUNB in classical-like neoplastic cells are missing:

a. The authors focus their attention on proving the co-expression of JUNB and GATA6 as proxy of CL subtype, and in some analyses, the correlation between GATA6 and JUNB is not very evident (Extended figure 1a). GSEA using CL signatures strengthens the findings, but the author should also report GSEA using BL subtype signatures.

Moreover, GSEA of JUNB bound regions using regions associated with classical/basal-like genes or using as gene sets the acetylated regions specific for low/high grade PDAC cells (Diaferia) could confirm that JUNB preferentially binds to regions associated with aggressive phenotype.

Finally, the authors should test if JUNB repression signature is expressed at different levels in classical vs basal like cell lines and if its expression anti-correlates with JUNB expression.

b. The IHC data are important to quantify the intratumoral co-existence of JUNB-high and -low area. However it is unclear why the other patients were excluded from the correlation analysis. How is the protein expression of GATA6 in patients not showing intratumor heterogeneity and does it confirm the positive correlation with JUNB? The authors analyzed 105 PDAC patients for expression of JUNB and GATA6 in malignant cells. However, it is not clear how this analysis was computed. Could the authors perform double staining to check co-expression of JUNB-GATA6 in the annotated tumor cells?

c. The expression of JUNB in non-tumor cells may compromise the results. As shown in fig 1p the majority of JUNB-expressing cells seems to be composed by non-malignant cells. In lines 168-169 the authors say that they queried transcriptome and proteome of PDAC samples epithelium- or stroma-enriched by laser-capture microdissection, however in Fig.1I it is unclear which data are correlated. The correlation between JUNB and GATA6 is reported for both epithelium- and stroma-enriched samples together? Is it possible to perform correlation on epithelium- or stroma-enriched samples separately to see if correlation is occurring only in the epithelial region?

2. The authors claim that JUNB is a key molecule for maintaining CL subtype, but it would be interesting to further analyze JUNB expression and co-expression with GATA6 in PDAC patients at different stages and upon different treatments. This is potentially very interesting:

a. Is JUNB expression changing in the different stages of PDAC? Fig.1O contains data from PDAC patients stage I-IV.

b. How do therapies impact on JUNB expression, CL/BL state and TNF expression?

The conclusions drawn by the authors on the expression of JUNB upon therapies are unclear. After chemo or neo-adjuvant therapy the % of tumor cells co-expressing JUNB and GATA6 decreases and the majority of JUNB+ cells are GATA6-negative. Does it imply that JUNB expression dissociate from classical-

like phenotype upon chemo or neoadjuvant therapy? Is it possible that GEM induces TNF production thus promoting suppressive TME and limiting its activity?

It would be of notice a deeper analysis, since in the final part of the paper the authors use Gemcitabine+aTNF to improve mice survival. In this experiment, control groups are missing: mice should be treated with GEM and aTNFa alone.

3. The authors claim that JUNB-mediated transcriptional repression of inflammatory BL-specific lineage signature is mediated by HDAC1. The ChIP-qPCR and IP experiments supporting this finding are performed on a restricted number of inflammatory genes, which are not enough to conclude that there is HDAC-mediated transcriptional repression. Genome-wide analysis on HDAC occupancy should be performed to assess concomitant binding of JUNB and HDAC1 in inflammatory genes, in presence and absence of JUNB to confirm the transcriptional repression of inflammatory genes by HDAC-mediated deacetylation. In addition to that, lines 259-261 referring to Fig.3C-E should be clarified. The authors say that all the reported genes are repressed since they display strong JUNB binding in absence of H3K27ac. However, it looks like there is some binding of JUNB in regions where H3K27ac is present near cJUN gene and to a lesser extent near IL1A gene.

4. The authors link high JUNB expression to reduced macrophage recruitment. The reduced macrophage infiltration is an important point in the manuscript, and it should be addressed more in depth. The data reported examine the correlation between JUNB expression and markers of monocytic lineage, relying on bulk RNA-seq scores. However, this analysis does not consider the fact that JUNB is expressed by multiple cell types and that even within tumor cells JUNB expression is heterogeneous (a key point of the study is that subtypes with different JUNB/cJUN expression co-exist within the same tumor). Other analyses would be more suitable to prove this finding, for example reanalysis of single-cell RNA-seq data, IF or IHC stainings. Additionally, in the data presented in Fig.5G, in patients stratified based on the JUNB repression signature score no striking difference is observed in the relative MCPcounter for the Monocytic lineage.

5. The authors claim that TNFa+ macrophage promote subtype coexistence by destabilizing CLA identity and promoting BL state. However major points need to be addressed:

a. TNFa+ macrophages are not shown. There is no identification of macrophages expressing TNF but only correlative analysis of macrophage content and TNFhigh regions. A double positive staining for CD68 and TNF in human and an intracellular FC analysis of macrophages expressing TNF in mice should be performed. This is particularly important since in figure 6D ctrl tumors show that the majority of TNF+ cells are CD45-.

b. How do cJUN areas recruit macrophages? Since JUNB silencing increase CK recruiting myeloid cells, are these cJUN areas producing higher levels of CCL2? Are cJUN OE cells producing higher levels of CCL2 then empty ones?

c. Does TNF induce basal like genes? In figure 5D the authors could perform GSEA using also BL signatures.

6. TNFa+ macrophages mark reactive stroma with low T cells

a. Figure 5F NMRI-nu/nu mice are immunodeficient mice lacking T cells, nevertheless the authors show a decrease recruitment of T cells in tumor-bearing mice treated with Tnf. Can you explain? Moreover, this

is in contrast with the increased expression of Cxcl9 and Cxcl10, key cytokines for recruitment of CD8+ T cells, in cancer cells upon JUNB silencing (Fig.3A)

7. Which is the effect of TNF in tumor vs stroma/immune cells?

Minor comments:

- In lines 203-204 the authors state that JUNB binds on a potential downstream enhancer of GATA6. However, in Fig.2B the peak of JUNB binding does not have the marker of active enhancers H3K27ac. How do they explain this?

- Unclear references: line 164 (Extended data Figure 1f-h); line 254 (Fig.2p,q)

- Increase dimension of characters for p-values in graphs and always explicit the comparisons the statistical analyses are referred to

- The authors should clarify in the methods how they defined the TNF-high, -int and -low regions. This should be clearer.

Reviewer #6 (Remarks to the Author): Expert in tumour immune microenvironment, immunogenomics, and bioinformatics; co-reviewed with Reviewer #5

Point-To-Point Response to Reviewer's Comments

We thank to the reviewers for their comments and suggestions. In response, we have conducted additional experiments and included new datasets to thoroughly address the points raised by the reviewers. Furthermore, we have restructured the manuscript to strengthen our core findings and comprehensively address the reviewers' suggestions. We believe that the inclusion of additional data and the reviewers' insights have enhanced our study. The following experiments have been incorporated into our findings in response to the reviewers' comments/suggestions:

Main Figures:

- Fig. 1e,f: JUNB and GATA6 co-expression and quantification in epithelial (panCK+) cells in chemotherapy-naïve PDAC patients.
- Fig. 2j: Correlation of directly JUNB-repressed genes in CAPAN1 cells (Fig. 2h) with JUNB itself in a panel of 46 PDAC patients of the Cancer Cell Line Encyclopedia (CCLE) to define a generalized, refined JUNB repression signature.
- Fig. 2k,l: Survival analysis for JUNB repression signature in PDAC patients, now using the refined JUNB repression signature and an additional cohort (Zhou et al. 2023)¹.
- Fig. 3k,l: ChIP-seq analysis for HDAC1 with JUNB-targeting siRNA or control, overlapped with JUNB binding regions; GREAT analysis of the overlapping regions.
- Fig. 3m,n: Immunoblot analysis for JUNB, cJUN, and CCL2 in additional CLA-like cell lines HPAF-II and CFPAC-1.
- Fig. 4i-n: JUNB/cJUN hotspot analysis in cJUN-OE orthotopic *in vivo* model for CD163 (M2 macrophages), CD86 (M1 macrophages), and CCL2.
- Fig. 5g: Updated MCPcounter analysis results for CD8 T cells, B lineages, and monocytic lineages in PDAC patient data using the refined JUNB repression signature.
- Fig. 5h: CIBERSORTx analysis for CD8 T cell, B cell, and macrophage populations in PDAC patient data using the refined JUNB repression signature.
- Fig. 5n,o: IHC analysis and quantification for CD163 (M2 macrophages) in orthotopically transplanted CAPAN1 tumors treated with TNF- α or VC for three weeks.

Extended Data Figures:

- ExtDataFig. 1a: Large overview areas for JUNB and GATA6 co-expression in epithelial (panCK+) cells in chemotherapy-naïve PDAC patients (as in **Fig. 1e,f**).
- ExtDataFig. 1b,c: JUNB and GATA6 co-expression in epithelial (panCK+) cells in orthotopically transplanted low grade/CLA CAPAN1 and high grade/BL MiaPaCa2 cells *in vivo*.
- ExtDataFig. 1d: Representative CK19 and JUNB staining in PMCC TMA cohort, now exemplifying manual classification of epithelial tumor-specific JUNB on the basis of the CK19 staining.
- ExtDataFig. 2e-g: Invasion assay and quantification of additional CLA-like cell line CFPAC-1 with JUNB-targeting siRNA or control, as well as confirmatory immunoblot.
- ExtDataFig. 2k: Gene Ontology analysis for the refined JUNB repression signature.
- ExtDataFig. 2l-o: Survival analysis for JUNB repression signature in PDAC patients for each individual cohort, using the refined JUNB repression signature and the additional cohort (Zhou et al. 2023)¹.

-
- ExtDataFig. 2p,q: Updated stage and grade data in PDAC patient data using the refined JUNB repression signature, now additionally including the cohort of the Bailey lab (Zhou et al. 2023)¹ for stage and the Puleo cohort for grade.
- ExtDataFig. 3e,f: ChIP-seq analysis for HDAC1 as well as H3K27ac with JUNB-targeting siRNA or control.
- ExtDataFig. 3g,h: ChIP-seq analysis for H3K27ac with JUNB-targeting siRNA or control, overlapped with JUNB binding regions; GREAT analysis of the overlapping regions.
- ExtDataFig. 3k: Immunoblot analysis for JUNB, cJUN, and CCL2 in CAPAN1.
- ExtDataFig. 3l,m: Reporter assay for cJUN promoter with JUNB-OE in additional CLA-like cell lines HPAF-II and CFPAC-1.
- ExtDataFig. 5a: CIBERSORTx analysis for M2 macrophages in stromal compartment of virtually microdissected RNA-seq data of orthotopically transplanted CAPAN1 tumors treated with TNF- α or VC for three weeks.
- ExtDataFig. 5b: Updated MCPcounter analysis results for all other immune cell types not included in **Fig. 5g** in PDAC patient data using the refined JUNB repression signature.
- ExtDataFig. 5c: CIBERSORTx analysis for all other immune cell types not included in **Fig. 5h** in PDAC patient data using the refined JUNB repression signature.

REVIEWER COMMENTS

Reviewer #1 (Remarks to the Author):

Klein et al. report a novel spatially regulated dichotomy in the AP1 transcriptional programs (JUNB vs. cJUN) that shapes regional subtype identity in PDAC via combining preclinical models, multi-center clinical, bulk and compartment-specific transcriptomic, proteomic, and bioimaging data from human specimens. Specially, they identified the TNF α -mediated inflammatory response associated with low JUNB expression destabilizes CLA neoplastic identity by promoting a CLA-to-BL transition in PDAC. Certainly this is a great resource for the community, but the insights drawn from this data are a bit overstated given that there is no comprehensive support for the findings. I really believe that the authors need to clarify the following points.

We thank the reviewer for their positive and thoughtful comments. In response to the reviewer's comments, we have been able to clarify several important points and included new datasets (e.g. Co-IP for JUNB-HDAC1/2/3, HDAC1/H3K27ac-ChIP-seq, CIBERSORTx and macrophage expression analysis) in the revised manuscript, as detailed below.

Major points:

1. It is unclear which samples are being analyzed in each results section. Maybe there can be better clarity of which samples were analyzed with what and how they are integrated/related to each other.

We thank the reviewers for bringing this to our attention. We acknowledge that adding labels to the cohorts in the figures, as well as explicitly mentioning them in the text, would enhance clarity. As a result, we have now included UMG (University Medical Center Göttingen), DKFZ (Deutsches Krebsforschungszentrum), and PMCC (Princess Margaret Cancer Centre) labels at the appropriate places. Moreover, we have included clinical data of these patient cohorts in the Methods section. Additionally, we have expanded the methods section for the analyzed data from the aforementioned datasets to provide better clarity. We have also made sure to appropriately label all publicly available data used in this study in the figures and the figure legends.

2. In Fig 1e, the authors indicated that GATA6 expression was higher in JUNB^{high} than JUNB^{low} regions within the same patient with patient-paired analysis. It is better to provide the representative whole slide images of IHC for GATA6 and JUNB in serial sections.

The findings of the PMCC cohort were obtained by analyzing tissue microarrays (TMA) from a large group of patient samples (n=105). Indeed, we here analyzed JUNB and GATA6 in near-adjacent but not, as requested, in serial sections. In order to visualize the expression of GATA6 in both high and low JUNB regions within larger tissue areas, we have now conducted co-expression analysis of JUNB and GATA6 in human PDAC patients and orthotopic-derived CLA or BL tumors. We have now included these new data sets from the UMG PDAC cohort (n=32) to demonstrate the co-expression of JUNB and GATA6 in single, epithelial (panCK+) cells using triple-IF with panCK, JUNB, and GATA6 in the corresponding sections (data now presented in **Fig.1e,f and ExtDataFig. 1a**, see **P2P-Fig. 1a,b**). In addition, new analyses in low-grade/CLA-like and BL tumors derived from pancreatic orthotopic tumors also showed that only the low-grade/CLA-like tumors show JUNB:GATA6 double-positive cells (specifically in the epithelial (panCK+) compartment; now included in **ExtDataFig. 1b,c**, see **P2P-Fig. 1c,d**). These data

demonstrates a strong relationship between the expression of JUNB and GATA6, as an increased JUNB-positive fraction correlates with high GATA6:JUNB double-positive neoplastic compartments.

Point-to-Point Response Figure 1. a, IF for JUNB, GATA6, and pan-cytokeratin (panCK) in resection tissue of therapy-naïve PDAC patients at representative region with high, intermediate, and low

epithelial JUNB expression in the University Medical Center Göttingen (UMG) cohort. Epithelial area is overlaid on greyscale images in magenta, based on panCK+ cell classification by QuPath. In the overlay, blue: DAPI, green: JUNB, magenta: panCK, yellow: GATA6. Scale bar 50 μ m. b, Quantification of a for JUNB+ and GATA6:JUNB double-positive epithelial (panCK+) cells. Linear regression with 95% CI, as well as Spearman's R and associated P value. n=32. c, As in a, for orthotopically transplanted CAPAN1 and MiaPaCa2 cells into NMRI-Foxn1nu/nu mice. Scale bar: overview, 200 μ m; insert, 50 μ m. d, Quantification of c for per-animal average percentage of GATA6:JUNB+ epithelial (panCK+) cells with mean \pm s.d. shown. CAPAN1, n=7; MiaPaCa2, n=8. Student's t-test with Welch's correction.

3. *In Fig 3, the authors hypothesized that HDACs may be involved in JUNB-mediated transcriptional repression of inflammatory BL-specific lineage signatures. However, the authors only select HDAC1 for further analysis. Please clarify it.*

We fully acknowledge that we did not adequately explain our rationale for focusing on HDAC1, and we have made this clearer in the revised version (please see page 9, line 261-275). Our main intention was to identify the JUNB-mediated repressor chromatin complexes that negatively regulate BL pro-inflammatory signatures. In our study, we observed HDAC target gene signatures that were enriched upon JUNB silencing (RNA-seq data; **Fig. 3f**). Here, HDAC1 was of special interest, as it has been highlighted in a comprehensive study as crucial determinant of the PDAC subtype heterogeneity (**ref**²⁷, as cited in the previous draft). Furthermore, multiple studies have shown the importance of Class I HDACs (HDAC1, 2, 3, 8) in PDAC^{3,4}. In addition, we only observed an interaction of JUNB with HDAC1 (**Fig. 3g,h**) but not with HDAC2 and HDAC3 on experimental levels (**P2P-Fig. 2a,b**); these were our primary reasons for focusing on HDAC1.

(cont) In addition, the authors only analyzed HDAC1 cooperates with JUNB in transcriptional repression, it is hard to conclude that JUNB restricts BL pro-inflammatory programs via HDAC-mediated repression.

Thank you for bringing up this important point. We have now conducted HDAC1 and H3K27ac ChIP-seq after siRNA against JUNB, reported now in **Fig. 3k,l**, and **ExtDataFig. 3e-h**. Our new findings further corroborate that JUNB recruits HDAC1 to the genome, as we observe almost exclusively loss of HDAC1 occupancy upon JUNB silencing. Out of the approximately 12,000 siCtrl-exclusive HDAC1 regions identified, 1454 overlap with JUNB binding regions we previously identified (**Fig. 3k**, see **P2P-Fig. 2c**). These overlapping JUNB-HDAC1 regions are enriched for inflammatory TNF- α signaling pathways (**Fig. 3l**, see **P2P-Fig. 2d**). Additionally, we found a high number of JUNB binding regions gaining H3K27ac, a mark of activation, on TNF- α signaling pathways upon siJUNB as well (**ExtDataFig. 3g,h**, see **P2P-Fig. 2e,f**), which altogether strongly supports the suppression of pro-inflammatory signaling by JUNB-HDAC1 in low-grade/CLA-like cells.

In addition to this new data, we have edited the text to more adequately delineate our initial rationale for focusing on HDAC1 in the current version of the manuscript (please see page 9, line 261-275).

Point-to-Point Response Figure 2. **a,b**, Immunoblot for JUNB, HDAC2, HDAC3, and β -actin after JUNB pull-down, IgG isotype control or input in CAPAN1 (**a**) and CAPAN2 (**b**). $n=3$. **c,d**, ChIP-seq analysis for JUNB in control cells and HDAC1 with siJUNB or siCtrl. **c**, Overlap of JUNB binding regions and regions where HDAC1 is significantly lost upon siJUNB (“HDAC1_DOWN”). **d**, GREAT analysis of the overlapping regions of **c** with $-\log_{10}(P_{adj})$ indicated. Hallmark (H) and curated (C2) signature collections of the Molecular Signature Database (MSigDB) are shown. **e**, Overlap of JUNB binding regions in control cells and regions where H3K27ac is significantly gained upon siJUNB (“H3K27ac_UP”). **f**, GREAT analysis of the overlapping region of **e** with $-\log_{10}(P_{adj})$ indicated. Hallmark (H) and curated (C2) signature collections of the Molecular Signature Database (MSigDB) are shown.

4. In Fig 4, the authors observed a trend towards higher immune infiltrations in cJUN-OE CLA-derived tumors. Why the author directly focuses on the changes of macrophages although the authors indicated the relationship between TNF α and macrophages with references. It is better to observe the changes of multiple immune cells in the microenvironment, and further changes in macrophages are reasonable. Further in Fig 5, the relationship between cJUN and TNF was introduced by references, without any experimental data support.

We understand the reviewer’s concerns and have added significant experimental data which supports a specific role of macrophages in this context, beyond our previous studies referenced in the original manuscript. As per reviewers’ suggestions, we have now performed comprehensive immunostaining analysis for distinct macrophages, as well as MCPcounter and CIBERSORTx analyses on patient data to understand the specific types and overall changes in immune cell populations in our study. This combined data from all cohorts has revealed that while other immune populations are largely unchanged or increasing along the JUNB repression signature, JUNB acts as a negative regulator specifically of M2 macrophages. Notably, high JUNB repression signature is associated with high monocytic lineage scores (**Fig. 5g**), and in particular high M2 macrophage infiltration (**Fig. 5h**), further supporting the rise in inflammatory signaling upon JUNB loss (see **P2P-Fig. 3g,h**).

We further clarified this observation by comprehensive immunostaining analysis for distinct macrophage populations. In particular, we have analyzed cJUN-overexpressing tumors to include further IHC/IF data for M1 and M2 macrophages and CCL2, now presented in **Fig. 4i-n**, see **P2P-Fig. 3a-f**. This clearly confirmed that the activation of

the cJUN-CCL2 axis leads to the recruitment of M2 macrophages (CD163+) to cJUN hotspot areas. Our results show that CCL2+ cells are located significantly closer to cJUN hotspots (310.2 μm) than to JUNB hotspots (516.4 μm), consistent with the data presented in **ExtDataFig. 4c-e**, which indicates an increase in CD68+/TNF- α + macrophages in cJUN-overexpressing tumors. Overall, our findings support that the loss of JUNB triggers a cJUN-CCL2 feed-forward loop, recruiting TNF- α + macrophages. This destabilizes the epithelial state and promotes a BL-inflammatory program.

Point-to-Point Response Figure 3. **a-f**, Distance analysis based on IHC for CD163, CD86, and IF for CCL2 in HA-cJUN-OE tumors. **a,c,e**, Distance analysis of CD163+ (**a**), CD86+ (**c**), or CCL2+ (**e**) cells to JUNB or HA-cJUN hotspots. Scatter plots show each individual cell, with mean \pm s.d. Student's t-test with Welch's correction. **a**, $n=7691$ CD163+ cells from $n=4$ tumors, with a total of $n=654297$ cells analyzed. **c**, $n=8384$ CD86+ cells from $n=4$ tumors, with a total of $n=632792$ cells analyzed. **e**, $n=14925$ CCL2+ cells from $n=5$ tumors, with a total of $n=745208$ cells analyzed. **b,d,f**, As in **a,c,e**, showing the shortest distances of each cell towards both the HA-cJUN and JUNB hotspots. **g**, Relative MCPcounter scores for the indicated lineages in 652 patients of the TCGA, QCMG, Puleo, and Zhou cohort, separated into quartiles based on the JUNB repression signature score. MCPcounter scores were min-max normalized and standardized to the mean of the lower JUNB repression signature score group for merging of the different cohorts. Mean \pm s.d. shown. **h**, As in **g**, but applying CIBERSORTx for deconvolution. Mean \pm s.d. for CIBERSORTx percentages shown.

5. In Fig.5g, after TNF α treatment, both T cells and B cells were decreased in PDAC patients with a high expression of the JUNB repression signature genes, the authors only observed the change of T cells, how about the B cells?

In reference to the MCPcounter analysis of the JUNB repression signature in PDAC patients (**Fig. 5g**), the reviewer correctly notes that low JUNB repression signature in patients exhibit higher T and B cell infiltration. We have corroborated this observation by using CIBERSORTx, which also demonstrates elevated levels of CD8 T cells, naïve and memory B cells (**Fig. 5h**, see **P2P-Fig. 3h**), as well as naïve and activated memory CD4 T cells (**ExtDataFig. 5c**, see **P2P-Fig. 4a**) in patients with low JUNB repression signature scores.

We therefore proceeded to validate the changes in B cells (marked by CD20 and normalized by CK19-negative TME area) in the TMA cohort. However, we did not observe any significant associations with JUNB expression or TNF- α (**P2P-Fig. 4b,c**). Consequently, we have chosen not to pursue further investigation into B cell populations.

Point-to-Point Response Figure 4. **a**, CIBERSORTx deconvolution percentages for the indicated lineages in 652 patients of the TCGA, QCMG, Puleo, and Zhou cohort, separated into quartiles based on the JUNB repression signature score. Mean \pm s.d. shown. **b,c** IHC analysis in 105 PDAC patients of the Princess Margaret Cancer Centre (PMCC) for epithelial JUNB expression (**b**) and TNF- α expression (**c**). Quantification for CD20 in JUNB^{low} (low) and JUNB^{high} (high) expression per sample (**b**) or in TNF- α ^{low} (low), TNF- α ^{intermediate} (int), and TNF- α ^{high} (high) expression (**c**).

6. The expression of JUNB and AP1 in PDAC patients with chemotherapy?

We agree that expression of JUNB and GATA6 in therapy-induced plasticity is extremely interesting and had thus started to assemble a matched chemo-naïve and post-chemotherapy patient cohort, as included in the original manuscript. However, obtaining patient tissue that had undergone neo-adjuvant chemotherapy before resection is not standard practice in our centers, making it a challenging task. We could therefore not obtain sufficient numbers of matched patient samples as requested by reviewer #2. At the same time, a recent study by the Peter Bailey lab (Zhou et al. 2023), published during the submission of our manuscript, compared the specificity of neo-adjuvant therapies (FOLFIRINOX vs. Gemcitabine) in PDAC subtype plasticity and its impact on patient prognosis. This study demonstrated subtype co-existence and the induction of an inflammatory immune microenvironment that largely depended on the choice of neo-adjuvant therapies. These findings emphasize the necessity of a comprehensive, large

scale investigation into the choice of chemotherapies in subtype plasticity. Therefore, we removed the respective data from our currently small cohort and will build this resource for future, detailed investigation of chemotherapy in JUNB and GATA6-mediated epithelial plasticity.

Instead, we have now utilized Fig. 1 to provide a clean and comprehensive evaluation of the important relationship between GATA6 and JUNB expression (in chemo-naïve PDAC patients). We expanded the cohort of chemo-naïve resected patients stained for GATA6, JUNB, and panCK (to mark the neoplastic compartment), which revealed a strong association between neoplastic epithelial (panCK+) JUNB and the amount of neoplastic epithelial GATA6:JUNB double-positive cells, now included in **Fig. 1e,f** and **ExtDataFig. 1a**, see **P2P-Fig. 1**. We are convinced that this refined focus will provide greater clarity of our proposed study and help elucidate this crucial point.

Minor points:

1. *In line 68-69: Currently, the 5-year survival rate for pancreatic cancer is 12%.*

We thank the reviewer for pointing this out. In Germany, we typically use an approximation of the US survival rate (12%) and Germany (10%). However, since we referenced the US data, we will adjust it to 12%. We apologize for the oversight.

2. *In line 164, should be fig.1f-h, not extended data.*

Thank you. The line should have read **Figure 1f-h** in the previous manuscript version. In the revised manuscript, this data has changed position, but the reference to it in the text has been corrected.

3. *In Line 388 and 391, the abbreviation of gemcitabine (GEM) should appear once.*

We have now removed the second explanation of the abbreviation.

Reviewer #2 (Remarks to the Author): Expert in pancreatic cancer functional genomics, therapy, preclinical models, and epigenetics; co-reviewed with Reviewer #3

In the manuscript by Klein and Tu et al., titled "Spatial Tumor Immune Heterogeneity Facilitates Subtype Co-existence and Therapy Response via AP1 Dichotomy in Pancreatic Cancer," the authors have conducted a comprehensive investigation using preclinical models, multi-center clinical data, and diverse omics analyses to explore the interplay between neoplastic intrinsic AP1 transcription factors and extrinsic CD68+ macrophages in pancreatic ductal adenocarcinoma (PDAC). Their analyses encompassed ATAC-seq, ChIP-seq, and RNA-seq, revealing JUNB/AP1- and HDAC-mediated epigenetic programs influencing pro-inflammatory immune signatures in tumor cells. This dichotomy in AP1 signaling was identified as a potential driver of intratumoral subtype co-existence and immunosuppressive tumor microenvironment with T cell exclusion.

The study delved into the role of TNF- α macrophages and their association with a reactive phenotype, impacting CD8+ T cell infiltration. Importantly, the authors suggest that combining anti-TNF- α immunotherapy and chemotherapy could potentially reduce macrophage counts and promote CD3+/CD8+ T cell infiltration, leading to improved survival in preclinical models. These findings propose a reciprocal interdependence between neoplastic and microenvironmental factors in shaping PDAC heterogeneity and aggressiveness. However, despite the authors' efforts to organize a coherent rationale, the manuscript is highly disorganized and lacks data-driven reasoning. Additionally, most results are not novel and, in some cases, contradictory with the current literature without sufficient justification for this discrepancy. Moreover, the results do not support most of the conclusions and are redundant in the classical/basal-like subtypes. Lastly, the in vitro and in vivo experiments lack coherence mainly due to the constant shift in cell types depending on the result.

We thank the reviewer for their valuable feedback and constructive suggestions. We have carefully considered all concerns raised, included new datasets, made extensive edits to increase coherence, and stated the specific advances provided by this study more clearly.

Before addressing the individual responses, we would like to clarify any concerns about the novelty of our findings. After careful review, we believe it would be beneficial to restate our specific focus and outline the new insights provided by the study (please see below). We are building on previous work, such as the identification of subtypes in PDAC, their varying chemo-response, and intratumoral co-existence, by aiming to provide the underlying cause for this intratumoral subtype co-existence and its associated immune diversity, thus offering a much-needed mechanistic basis for it.

Collectively, we identified that:

- a) PDAC patients have intratumoral heterogeneity in AP1 expression, CD68+ macrophages and CD8+ T cell within individual patients;
- b) The classical-associated JUNB/AP1 transcriptional program keeps inflammatory immune response low via HDAC-mediated epigenetic repression;
- c) Loss of JUNB function is significantly linked to worse PDAC patients outcome, associated with an increased BL inflammatory immune response;
- d) Co-existing cJUN-high PDAC cells destabilize classical neoplastic state in the local neighborhood via a CD68+/TNF- α + macrophage-driven inflammatory response; and
- e) Combined anti-TNF- α immunotherapy and standard chemotherapy can reduce CD68+/TNF- α + macrophage infiltration and restore CD3+/CD8+ T cell infiltration and improve survival.

Comments:

1. *It is unclear why the authors focused their analysis on JUNB and cJUN without a data-driven approach. The authors must clarify.*

We fully acknowledge that the rationale for studying the JUNB and cJUN transcription factors was not sufficiently clear in the original manuscript and thank the reviewer for pointing this out. Rather than a profiling-driven approach, this is indeed a hypothesis-driven study that builds on previous work and aims to address key unanswered questions arising from recent studies on intratumoral subtype co-existence. In our previous research, we used a data-driven method to identify signature pathways associated with PDAC subtype heterogeneity. Our results indicated that the AP1 pathway was significantly enriched in tumors of BL PDAC patients⁵. We have now included the following lines in the revised results section (please see page 6, line 139-145) to provide a better rationale:

“In our previous study, the JUN/AP1 pathway was found to significantly influence the subtype identity of PDAC through tumor cell-intrinsic and extrinsic mechanisms. Due to their ability to integrate extrinsic inflammatory signals and intrinsic transcriptional programs, the AP1 transcription factors (TF) JUNB and cJUN are particularly important in the context of intratumoral subtype plasticity in PDAC. In this study, the initial focus was on JUNB/AP1, as it has been implicated in the identity of low-grade or CLA-like PDAC in previous research, including our own findings.” In this study, the initial focus was on JUNB/AP1, as it has been implicated in the identity of low-grade or CLA-like PDAC in previous research, including our own findings^{16,21,23,34,35}.

In the previous study, we demonstrated that the BRD4-cJUN/AP1-driven transcriptional network recruits tumor-associated inflammatory macrophages, which is crucial for the maintenance of basal-like subtype identity⁵. However, the underlying mechanisms, especially with regard to loss of a classical identity, and their specific relation to intratumoral subtype co-occurrence had remained unclear. In this study, we are reporting a spatially regulated dichotomy in the AP1 TF program (JUNB vs. cJUN) that governs subtype co-existence/hybrid state in PDAC via both tumor cell-intrinsic and extrinsic mechanisms. In brief, we show how CLA PDAC cells maintain their epithelial state and how the inflammatory microenvironment then modulate this cellular program, such as via TNF α + macrophages. Specifically, we expose the connection between the loss of CLA-associated JUNB/AP1 transcriptional program via increased inflammatory immune response in the local neighborhood, leading to the destabilization of CLA-like neoplastic identity.

We have thoroughly revised the manuscript, incorporating new in vitro and in vivo data analysis as well as reorganizing the content to clearly elucidate our overall study.

2. *The authors argue that JUNB is related to the classical subtype and cited two references. However, JUNB's function is associated with proteins with which it interacts and post-translational modifications. Therefore, it could promote a differentiation state or EMT (doi: 10.1083/jcb.201109045, doi:10.1186/s13059-022-02800-0). The authors must clarify this point and define the specific protein complex associated with JUNB and the classical subtype.*

We thank reviewer for their thoughtful comment. As the reviewer noted, the AP1 transcription factors JUNB and cJUN play highly context-dependent roles. In neoplastic PDAC cells, we and others have demonstrated that JUNB expression is highly specific to low-grade, CLA-like PDAC tumors, while it is absent in high-grade PDAC or PDX-derived cell lines⁵⁻⁸. Conversely, cJUN is absent in low-grade, CLA-like PDAC and expressed in high-grade PDAC tumors⁵.

In this study, we found that JUNB loss, either through gene silencing or through external TNF- α stimulation, significantly reduces the JUNB-mediated epithelial differentiation program, triggering invasive and pro-inflammatory properties. In line with these findings, our CHIP experiments (**Fig.2 a-f**) revealed JUNB binding over GATA6, FOXA1 and HNF1B, as well as enrichment of motifs of these and other CLA-associated TFs in the JUNB binding sites, which are involved in maintaining the low-grade/CLA-like subtype of PDAC^{6,7} (see **P2P-Fig. 5a**). The question of whether JUNB physically interacts with other TFs remains open and future studies should prioritize identifying additional JUNB-associated partner TFs that provide insights into the JUNB/AP1 partner complex in CLA-like PDAC cells. Notably, we demonstrate that JUNB physically interacts with HDAC1 to repress its target gene signatures. Furthermore, we have performed HDAC1 and H3K27ac CHIP-seq upon JUNB silencing, strongly indicating the significance of HDAC1 and JUNB interactions in repressing pro-inflammatory gene signatures (now included in **Fig. 3k,l** and **ExtDataFig. 3e-h**, see **P2P-Fig. 5b-e**).

In the Results and Discussion section, we have included the following for better clarity:

Results:

“TFs rely on additional epigenetic co-regulators to exert their regulatory functions on lineage gene expression. A previous study identified key epigenetic regulators, such as HDAC1, as crucial in determining PDAC subtype heterogeneity²⁷. Particularly, in the gene signatures directly repressed by JUNB (see Fig. 2i, ExtDataFig. 2k) and in GSEA in JUNB silencing transcriptome data (Fig. 3f), HDAC target signatures were found to be enriched. Therefore, we hypothesized that HDAC1 may be involved in JUNB-mediated transcriptional repression of BL-associated inflammatory lineage signatures.”

Discussion:

“Notably, the maintenance of a CLA-like epithelial state or the suppression of pro-inflammatory factors by JUNB requires the epigenetic co-regulator HDAC1. Our study emphasizes the importance of previous work²⁷, which has demonstrated how the interaction of epigenetic co-regulators like HDAC1 with lineage-specific TFs can affect PDAC heterogeneity.”

Point-to-Point Response Figure 5. **a**, Analysis of motif enrichment in JUNB-bound sites by CHIP-seq in CAPAN1 cells published previously⁵. Negative \log_{10} adjusted P values are indicated. **b**, Overlap of

JUNB binding regions and regions where HDAC1 is significantly lost upon siJUNB (“HDAC1_DOWN”). **c**, GREAT analysis of the overlapping regions of **b** with $-\log_{10}(P_{adj})$ indicated. Hallmark (H) and curated (C2) signature collections of the Molecular Signature Database (MSigDB) are shown. **d**, Overlap of JUNB binding regions in control cells and regions where H3K27ac is significantly gained upon siJUNB (“H3K27ac_UP”). **e**, GREAT analysis of the overlapping region of **d** with $-\log_{10}(P_{adj})$ indicated. Hallmark (H) and curated (C2) signature collections of the Molecular Signature Database (MSigDB) are shown.

3. The authors performed most of their experiments using CAPAN 1 and 2 and defined these cells as classical. However, they did not specify the criteria or methodology used to subtype the cells. Additionally, restricting the analysis to only two cell lines could generate bias in the interpretation of the results, mostly because CAPAN1 has alterations in BRCA2, representing a low proportion of PDAC patients (5%). The authors must include other types of cells such as BXPC3 in order to confirm their observations regarding JUNB and the classical phenotype.

We agree. Replicating the full spectrum of classical to basal-like identity seen in human PDAC with cell lines is near-impossible due to the inherent artificial nature of these systems and we value the reviewer’s suggestion to enhance the robustness of our findings by expanding the number of cell lines in our analysis.

Our initial selection of PDAC cell lines and their-associated subtype-specificity was largely based on the previous comprehensive PDAC subtype-specific *in vitro* and *in vivo* studies^{5-7,9,10}. Therefore, we opted to use CAPAN1 and CAPAN2 as the primary established cell lines to study classical-like subtype identity because they have been extensively used and characterized as low-grade, well-differentiated PDAC at both the transcriptional and functional levels^{5-7,9,10}. Moreover, CAPAN1 and CAPAN2 cell lines results in the formation of low-grade/well-differentiated tumor histology in immunocompromised mouse models^{5,10}. In addition, we had included a patient-derived xenograft (PDX) cell line (GCDX62) in the present study, which mimics CLA-like phenotypic identity both on molecular (high JUNB/GATA6/CDH1) and functional levels (low invasion rate) as shown previously⁵.

To assess the CLA status of available cell lines, we analyzed the RNA expression data of the Cancer Cell Line Encyclopedia (CCLE)^{11,12} and performed gene set variation analysis for the Collisson¹³, Moffitt¹⁴ and Notta¹⁵ CLA(-A/B) signatures for a panel of 46 pancreatic cancer cell lines (**P2P-Fig. 6**). This confirmed that our main models of the BL identity (PANC1, MIAPACA2) were strongly depleted in CLA-related signatures, while CAPAN1 and CAPAN2 fell into the CLA half of the spectrum (please see **P2P-Fig. 6**). We then selected additional CLA models. We initially pursued the reviewer’s suggestion of using the additional cell line BXPC3 for CLA subtype confirmation. However, although they do enrich for CLA signatures (**P2P-Fig. 6**), we soon refrained from using BXPC3 as these cells harbor wild-type KRAS and were reported to actually also display key properties of the basal-like PDAC subtype, such as high expression of TP63¹⁶⁻¹⁸, a master regulator of the BL identity in PDAC. Additionally, Tonelli et al.¹⁹ demonstrated that orthotopically transplanted BXPC3 cells form a poorly differentiated tumor histology.

In the revised version of the manuscript, we have now included the cell lines CFPAC-1 and HPAF-II as additional models representing CLA well-differentiated PDAC. We repeated all key experiments, including transwell invasion assay, and immunoblots for inflammatory markers (e.g. CCL2) after JUNB silencing, as well as reporter assay for the cJUN promoter in CFPAC-1 and HPAF-II cell lines. Importantly, the results from these experiments further confirm the control of BL-associated pro-inflammatory immune and invasive functions by JUNB in CFPAC-1 and HPAF-II cell lines.

Point-to-Point Response Figure 6. Gene set variance analysis (GSVA) scores for Notta¹⁵, Moffitt¹⁴, and Collisson¹³ CLA gene signatures in 46 PDAC cell lines of the Cancer Cell Line Encyclopedia. Cell lines used in this study are indicated.

4. The authors concluded that *GATA6* and *JUNB* have a positive correlation. However, the results were not significant. Therefore, such affirmation is not accurate. The authors must

correct this point and show in the manuscript the correlation coefficient and the P-value for all cohorts when comparing GATA6 and JUNB. They must underline the fact that the correlations were not significant.

We apologize for the lack of clarity. As the reviewer points out correctly, we showed correlation analyses of GATA6 and JUNB in RNA-seq data from tumor-bulk of three cohorts, which was not significant. As we have mentioned in the text, this is likely confounded by stromal expression of these factors, as our subsequent analyses of epithelial-specific expression consistently show significant association of GATA6 and JUNB across multiple cohorts and analyses types. This includes analyses of

- resected PDAC patient tumors
 - o manually annotated epithelial (CK19+) regions of TMAs (**Fig. 1g-m**)
 - o multiplex staining of the neoplastic (panCK+) compartment, which is new data in response to Reviewer #1 (data now presented in **Fig.1e,f** and **ExtDataFig. 1a**, see **P2P-Fig. 7a,b**)
- orthotopic-derived CLA or BL tumors (new data now included in **ExtDataFig.1b,c**, see **P2P-Fig. 7c,d**)
- flow-cytometry-sorted epithelial (EPCAM+) cells (**Fig. 1c**),
- PDAC microdissected epithelia (**Fig. 1n**)

We acknowledge this detour was clearly confusing and have thus decided to remove the bulk sequencing data altogether (previous version of the manuscript **ExtDataFig. 1a-c**). We hope this modification will help to better convey the central finding, which is the strong association of JUNB and GATA6 in neoplastic epithelial PDAC cells. We appreciate the reviewer's feedback on the disorganization of the data, and we hope that our re-organization addresses this concern.

Point-to-Point Response Figure 7. a, IF for JUNB, GATA6, and pan-cytokeratin (panCK) in resection tissue of therapy-naïve PDAC patients at representative region with high, intermediate, and low epithelial JUNB expression in the University Medical Center Göttingen (UMG) cohort. Epithelial area is overlaid on greyscale images in magenta, based on panCK⁺ cell classification by QuPath. In the overlay, blue: DAPI, green: JUNB, magenta: panCK, yellow: GATA6. Scale bar 50 μ m. **b**, Quantification of **a** for JUNB⁺ and GATA6:JUNB double-positive epithelial (panCK⁺) cells. Linear regression with 95% CI, as well as Spearman's *R* and associated *P* value. *n*=32. **c**, As in **a**, for orthotopically transplanted CAPAN1 and MiaPaCa2 cells into NMRI-*Foxn1*^{nu/nu} mice. Scale bar: overview, 200 μ m; insert, 50 μ m. **d**, Quantification of **c** for per-animal average percentage of GATA6:JUNB⁺ epithelial (panCK⁺) cells with mean \pm s.d. shown. CAPAN1, *n*=7; MiaPaCa2, *n*=8. Student's t-test with Welch's correction.

5. The cohort of 105 patients from the TMA must be further characterized in the manuscript. Specifically, the source (metastatic, resectable), treatment status at the time of the biopsy, and the tumor subtype.

We thank the reviewer for this suggestion. We have included the following paragraph in the Methods section to provide additional details about the TMA cohort:

“The PDAC tissue microarray (TMA) cohort comprises resectable tumor specimens from 105 treatment-naïve patients diagnosed with PDAC. These included 50 female and 65 male patients, 9 stage I and 96 stage II, with a mean age at diagnosis of 65.8 years (age range 42-84 years). These tumors were obtained from the UHN Biospecimens Program after collection at Princess Margaret Cancer Centre (Toronto, Canada) and have been previously discussed in studies^{41,52,53}. All patients provided written informed consent for the molecular characterization of their tumor samples and for follow-up on their clinical information. Out of the 105 patients, specimens from the Tissue Microarray were utilized in the study. To create the TMA, a pathologist identified the optimal area for coring

reviewed sections from paraffin blocks. After marking the areas, 1.2 mm tissue cores were manually punched and transferred into recipient paraffin blocks. In addition to the tumor samples, cores of benign pancreatic, renal, pulmonary, and hepatic tissues were included for control purposes and to aid in TMA slide orientation. Multiple tumor cores were arrayed (2-4-fold redundancy) from the paraffin blocks and spread across multiple slides for the final tissue microarray. Additionally, for a subset of n = 98 patients, tumor specimens were processed by Laser Capture Microdissection (LCM) to enable epithelial-enriched RNAseq profiling. This analysis revealed that n = 36 tumors exhibited a basal-like profile, while n = 62 exhibited a classical profile, as determined using the clustering method described previously⁵.”

(cont.) For the TMA, the authors need to clarify for each picture the % of epithelial and stroma cells and indicate which cells are considered positive. Additionally, defining the criteria for JUNB high and low (method and cutoff) is necessary.

As outlined above, all analyses were focused on epithelial cells and in this specific regard, malignant epithelia had been manually annotated before analysis. Accordingly, stromal cells were not included. We have included the following paragraph in the Methods section to provide additional details about IHC stain quantification:

“IHC staining was quantified using QuPath bioimage analysis software. Individual TMA cores were registered, and patient identifiers were superimposed for analysis. Malignant epithelial regions were manually annotated for JUNB, Cytokeratin 19, CDH1 and GATA6 expression (see also Extended Data Fig 1d), while the Simple Tissue Detection tool was used to select tissue for TNF- α expression analysis. Residual normal pancreas epithelium, nerves, large blood vessels, tissue folds and stain artifact were manually excluded. Pixel-based detection parameters were set and optimized for each stain. ‘Positive pixel percentage’ within the annotated epithelial regions was determined as the fraction of positive pixels within all detected pixels and as indicated, either averaged across all cores per patient (patient level). For display and statistical testing, samples were then stratified into groups (‘high’, ‘intermediate’, ‘low’) via the top, intermediate and bottom 33% of the obtained JUNB and TNF- α positive pixel percentage values, respectively.”

(cont.) Lastly, the authors must report the number of cores per patient.

We have now included the following paragraph in the Methods detailing the used TMA:

“To create the TMA, a pathologist identified the optimal area for coring reviewed sections from paraffin blocks. After marking the areas, 1.2 mm tissue cores were manually punched and transferred into recipient paraffin blocks. In addition to the tumor samples, cores of benign pancreatic, renal, pulmonary, and hepatic tissues were included for control purposes and to aid in TMA slide orientation. Multiple tumor cores were arrayed (two- to four-fold redundancy) from the paraffin blocks and spread across multiple slides for the final tissue microarray.”

6. The sources of each set of patients must be included in the manuscript (105 TMA, 23 IF...). Also, clinical data concerning the set of patients is necessary in supplementary data to fully understand the presented results.

Thank you kindly for bringing this to our attention. In response to the reviewer's suggestion, we have added the following information about the clinical details for

patients from both the TMA cohort of 105 patients (PMCC cohort) and the now expanded IF dataset of 32 patients (UMG cohort).

“This UMG cohort included 32 chemotherapy-naïve, resected primary PDAC cases, with 23 female and 9 male patients, 5 stage I, 19 stage II, and 8 stage III, with a mean age at diagnosis of 72.2 years (age range 49-86 years).”

and

“The PDAC tissue microarray (TMA) cohort from Princess Margaret Cancer Centre (PMCC) comprises resectable primary pancreatic tumor specimens from 105 treatment-naïve patients diagnosed with PDAC. These included 50 female and 65 male patients, 9 stage I and 96 stage II, with a mean age at diagnosis of 65.8 years (age range 42-84 years).”

7. The authors need to add more information about the FACS experiment. How was the data processed to select the epithelial cell population?

We apologize for not clarifying this in the manuscript text. The RNA-seq data utilized in this study has been previously published^{5,20}. Consequently, we have only briefly addressed the data acquisition and processing in the "Patient data analysis and study approval" section of the Methods (please see **page, 21**, line 616-623). To enhance clarity, we have now segregated the FACS data into a separate section within the Methods.

“The compartment-sorted patient transcriptome data has been published previously²². In brief, tumor tissue of untreated patient with partial pancreateoduodenectomy at the Department of General, Visceral and Transplantation Surgery, University of Heidelberg (HIPO-project approved by the ethical committee of the University of Heidelberg; case number S-206/2011 and EPZ-Biobank Ethic Vote no. 301/2001) were subjected to fluorescence-activated cell sorting with compartment-specific markers (for epithelial EPCAM⁺/CD45⁻/CD31⁻) and subsequently RNA-sequenced in the sorted fractions.”

We apologize if the information regarding this experiment was ambiguous in the Methods previously.

8. The authors claim that CLA showed the highest expression of JUNB, however BL-B displays similar levels. Performing additional statistics comparing CLA with the other subtypes is required.

We have now conducted individual pairwise comparisons for all combinations of Fig. 1o of the previous manuscript, in addition to the Kruskal-Wallis test reported previously (**P2P-Fig. 8**). Therein, we observed a significant difference of CLA-A to both BL-A as well as BL-B, as in the manuscript text (previous version line 184). Please note that we are showing this analysis for the reviewer only as we have decided to remove the data of **Fig. 1o** from the revised manuscript, in alignment with further comments from other reviewers. **Fig. 1** is now specifically focused on the relationship of JUNB with GATA6 in neoplastic epithelial cells.

Point-to-Point Response Figure 8. JUNB expression in LCM-enriched epithelia of resectable patients and COMPASS trial advanced patients, classified for the Notta/Chan-Seng-Yue¹⁵ PDAC subtypes. Kruskal-Wallis test and Student's t-test with Welch's correction. n=486.

9. *The authors must clarify the number of samples assessed. Additionally, using only 4 samples per group with the observed variations could generate biases in the conclusions. The authors must include additional samples per each group.*

We understand the reviewer's concern about the limited sample size for **Fig. 1p,q** of the previous version of the manuscript, which is also in line with other reviewers' comments. We initially aimed to analyze the expression of JUNB and GATA6 in therapy-induced plasticity using a matched chemo-naïve and post-chemotherapy patient cohort. However, obtaining patient tissue that had undergone neo-adjuvant chemotherapy before resection is challenging, as it is not standard practice in our centers. We were not able to obtain much larger number of matched patient samples. In addition, a recent study by the Peter Bailey lab (Zhou et al. 2023)¹, published during the submission of our manuscript, compared the specificity of neo-adjuvant therapies (FOLFIRINOX vs. Gemcitabine) in PDAC subtype plasticity and its impact on patient prognosis. This study demonstrated subtype co-existence and the induction of an inflammatory immune microenvironment that largely depended on the choice of neo-adjuvant therapies. These findings emphasize the necessity of a comprehensive, large-scale investigation into the choice of chemotherapies in subtype plasticity. Therefore, we decided to remove the respective data from our currently small cohort and will build this resource for future, detailed investigation of chemotherapy in JUNB and GATA6-mediated epithelial plasticity. Removal of this data does not alter the overall study conclusion; rather, we are convinced it provides greater clarity for our proposed study.

Instead, we followed the suggestion to provide additional evidence for the relationship of GATA6 and JUNB in chemo-naïve resected PDAC patients, which is a key point of this study. We expanded the cohort of chemo-naïve resected patients (n=32) stained for GATA6, JUNB, and panCK in the neoplastic compartment, which revealed a strong association between neoplastic epithelial (panCK+) JUNB and the amount of neoplastic epithelial GATA6:JUNB double-positive cells, as shown above in P2P-Fig. 7a,b (page, 15/16), and now included in **Fig.1e,f** and **ExtDataFig. 1a** of the manuscript.

(cont.) Lastly, the authors claim that classical cells are sensitive to chemotherapy. This is not an original observation; other authors have pointed out this fact (doi: 10.1158/1078-0432.CCR-19-146).

We apologize for the confusion regarding the data presented in **Fig. 1p,q** of the previous version of the manuscript. As the reviewer mentioned, it is well known that CLA identity is associated with higher chemotherapy sensitivity compared to BL tumors. Originally, the data in **Fig. 1p,q** was intended to illustrate that GATA6:JUNB double-positive neoplastic cells are more abundant in chemotherapy-naïve patients and are partially diminished after treatment. We regret if this was not clearly stated in the manuscript. However, as mentioned above, the respective data and any associated statements have been removed in the revised manuscript.

10. The authors must clarify in the ChIP-seq experiment the cells used. Additionally, integrating publicly available data with novel data, even if they come from the same cell, could generate biases in the analysis associated with a batch effect. The authors must describe their approach to solve this point.

Thank you for raising this important point. CAPAN1 cells were used for both the JUNB ChIP-seq experiment and the RNA-seq experiment after JUNB silencing. We understand the reviewer's concern about potential batch effects when comparing our data to publicly available data and are aware of this issue. In fact, to minimize such biases, the statistical analyses for ChIP-seq and RNA-seq were conducted separately within their own biological replicates and the results of these independent analyses were integrated, such as JUNB binding regions and differentially expressed genes upon siJUNB, not the raw data. Furthermore, both data sets were generated in our lab using the same cells and culture conditions, which further minimizes variability.

11. Defining the meaning of "potential downstream enhancer of GATA6" in the ChIP-seq experiment is essential

We apologize for the lack of clarity in **Fig. 2b**. Previous studies have shown that enrichment of H3K4me1 strongly correlates with enhancer regions, while H3K4me3 enrichment is associated with promoter regions or transcription start site (TSS)²¹. We consider the region downstream of the GATA6 gene body to be a potential enhancer, as it is marked by H3K27ac as well as, importantly, a peak for H3K4me1, as shown in **P2P-Fig. 9**. H3K4me3, a marker of TSS regions, is not binding at the enhancer region. Therefore, we only characterize this as a "potential" enhancer, as we currently lack direct evidence of this region looping to the promoter of GATA6, or any functional validation of enhancer activity.

Point-to-Point Response Figure 9. Representative ChIP-seq tracks for JUNB published previously⁵ as well as publicly available data⁶ for H3K27ac, H3K4me3, and H3K4me1 for the GATA6 locus.

12. The authors performed siRNA on JUNB and argued that the silencing of this gene promotes an invasive state. However, they did not perform functional analysis of the cells with and without siRNA to validate that affirmation.

We had performed transwell invasion experiments with silenced JUNB in both CAPAN2 and GCDX62 cell lines. This experiment was mentioned in line 220ff and the results were presented in **ExtDataFig. 2b-g** in the previous draft, see **P2P-Fig. 10a-f**.

In response to the reviewer's suggestions to include additional CLA cell lines, we also conducted transwell invasion assay experiments in CFPAC-1 cells to increase the number of CLA subtype cell lines (**P2P-Fig. 10g-i**). The results, now included in **ExtDataFig. 2e-g**, showed increased invasiveness upon JUNB loss.

Point-to-Point Response Figure 10. a-i, Transwell invasion assay for CAPAN2 (a-c), GCDX62 (e-g), and CFPAC-1 (h-j) with siJUNB or siCtrl. a,d,g, Immunoblot for JUNB and β -actin after siJUNB or siCtrl in CAPAN2 (b), GCDX62 (e), and CFPAC-1 (h), validating silencing for the invasion assay. n=3. b,e,h, DAPI staining of invaded CAPAN2 (b), GCDX62 (e), or CFPAC-1 (h) cells. Scale bar 100 μ m. c,f,i, Quantification of b,e,h, for number of invaded cells. Average counts per FOV with mean \pm s.d. shown. c,i, n=6 inserts from n=3 independent experiments. f, n=7 inserts from n=4 independent experiments. Student's t-test with Welch's correction.

13. The authors used the full TCGA cohort (177 patients); however, this cohort contains normal, inflamed pancreas, and non-PDAC neoplasms (doi: 10.3390/cancers11010126). It is required that the authors elucidate this point.

We are grateful to the reviewer for highlighting this point. In line with their suggestion, we have confined the analysis of the TCGA cohort to the 150 curated PDAC cases as detailed by Nicolle *et al.*²². The curated TCGA patient cohort was used in all relevant figures, including **Fig. 2k,l**, **Fig. 5g,h**, **ExtDataFig. 2l,p,q**, **ExtDataFig. 5b,c**. The exclusion of non-PDAC cases has not altered the outcome of the analysis; our key message remains consistent.

14. *The inferences between macrophage infiltration and the levels of JUNB are unclear. The authors claim that patients with high JUNB (Classical) display low levels of macrophages, whereas those with low JUNB (basal-like) display high levels of macrophages. This conclusion is barely supported by TCGA, and the other cohorts do not display the same patterns. The different cohorts are not comparable on this subject due to lack of information concerning the scale or the different types of clusters. Additionally, the type of macrophages must be evaluated. They must include more than one deconvolution algorithm.*

This is an excellent point. We have now performed comprehensive immunostaining analysis, MCPcounter and CIBERSORTx analysis address reviewer's concerns. As previously mentioned, we mainly used a combination of MCPcounter and marker expression to assess macrophages. In response to the reviewer's feedback, we have conducted CIBERSORTx analysis on the patient data to understand the specific types and overall changes in macrophage populations associated with the JUNB repression signature.

Our analysis of the combined data from all cohorts has revealed highest levels of M2 macrophages in patients with high JUNB repression signature scores, while other macrophage populations are largely unaffected (now included in **Fig. 5h**, see **P2P-Fig. 11a**). Experimentally, we have now conducted an analysis of CD86 (M1) and CD163 (M2) macrophages in relation to JUNB and cJUN hotspot areas in orthotopic cJUN-overexpressing CLA-like tumors (now included in **Fig. 4i-l**, see **P2P-Fig. 11b-e**). This confirmed close association of M2 macrophages with cJUN areas, while further away from JUNB hotspots. The opposite was found true for M1 macrophages.

Additionally, we have performed M2 macrophage infiltration analysis by IHC in our TNF- α -treated CLA-like tumors (now included in **Fig. 5n,o**, see **P2P-Fig. 11f,g**), as well as CIBERSORTx analysis in the corresponding virtually microdissected stromal expression profile (now included in **ExtDataFig. 5a**, see **P2P-Fig. 11h**). These analyses have demonstrated significantly lower M2 macrophages in the JUNB-high vehicle control samples compared to JUNB-low TNF- α -treated tumors. Further details are also discussed in the response in section 21 below.

Point-to-Point Response Figure 11. **a**, CIBERSORTx deconvolution percentages for the indicated lineages in 652 patients of the TCGA, QCMG, Puleo, and Zhou cohort, separated into quartiles based on the JUNB repression signature score. Mean \pm s.d. shown. **b-e**, Distance analysis based on IHC for CD163 and CD86 in HA-cJUN-OE tumors. **b,d**, Distance analysis of CD163⁺ (**b**) and CD86⁺ (**d**) cells to JUNB or HA-cJUN hotspots. Scatter plots show each individual cell, with mean \pm s.d. Student's t-test with Welch's correction. **b**, n=7691 CD163⁺ cells from n=4 tumors, with a total of n=654297 cells analyzed. **d**, n=8384 CD86⁺ cells from n=4 tumors, with a total of n=632792 cells analyzed. **c,e**, As in **b,d**, showing the shortest distances of each cell towards both the HA-cJUN and JUNB hotspots. **f**, CD163 IHC staining in orthotopically transplanted CAPAN1 tumors treated with TNF- α or VC. Arrows indicate positive cells. Scale bar: overview, 100 μ m; insert, 25 μ m. **g**, Quantification of **f** for per-animal percentage of CD163⁺ cells with mean \pm s.d. shown. n=7. Student's t-test with Welch's correction. **h**, CIBERSORTx analysis for M2 macrophages in stromal compartment of virtually microdissected RNA-seq data of orthotopically transplanted CAPAN1 tumors treated with TNF- α or VC for three weeks. VC, n=2; TNF- α , n=3. Student's t-test with Welch's correction.

15. The authors need to illustrate how the JUNB repression signature was generated.

The JUNB repression signature used in the new version of the manuscript was generated in two steps. First, JUNB ChIP-seq regions were assigned to genes using rGREAT, and these were overlapped with the genes that are significantly upregulated upon JUNB silencing. Then, the correlation of the resulting 146 genes to JUNB itself was analyzed in a panel of 46 PDAC cell lines of the Cancer Cell Line Encyclopedia

(CCLE), retaining only those which had a negative Spearman's R , for the final 37-gene JUNB repression signature. We have updated the methods to make this clearer:

“For integration of RNA- and CHIP-seq data, consensus CHIP-seq peaks were annotated to genes using the R package rGREAT²³ v3.0.0, and their fold change in the corresponding RNA-seq extracted. Gene ontology (GO) analysis was then performed for the significantly up- or downregulated genes of this subset using the clusterProfiler package as above. The overlap of JUNB-bound genes (as per rGREAT) with genes that are significantly upregulated upon siJUNB in the RNA-seq were subsequently analyzed for their correlation to JUNB in 46 pancreatic cancer cell lines of the Cancer Cell Line Encyclopedia (CCLE, Stransky et al. Nature. 2015, Nusinow et al. Cell. 2020) dataset, which was downloaded from cBioPortal. Those genes that had an inverse association with JUNB (negative Spearman's R) were termed the *JUNB repression signature*; the full list is available in **Supplementary Table 3.**”

16. Separating the results from TCGA, QCMG, and Puleo when analyzing the overall survival, DFS and JUNB repression signature (5.g) is fundamental. Also, include the number at risk and which P-value (log-rank test?) is reported in the KM. Additionally, the authors must perform a multivariate Cox analysis with clinical variables per cohort. Lastly, they must demonstrate the novel contribution of their JUNB-based classification compared to the other transcriptomic subtypes, such as PurlST.

As per the reviewer's suggestions, we have performed individual survival analyses and included all relevant transcriptomic subtyping schemes. We have, however, not yet included this data into the revised manuscript as we would like to point out that the JUNB repression signature was not designed as a prognostic tool for clinical application. Rather, it is a metric of key biological program that underlies the establishment of prognostic tumor subtypes. We used the survival data in patients to validate our in vitro findings on the cell biological effects of this program in the clinical setting, which we believe is thereby well supported. Nevertheless, if the reviewer prefers, we will include the multivariate analyses in the manuscript.

The results are provided below and show individual survival analyses for the different cohorts in **ExtDataFig. 21-o** (see **P2P-Fig. 12a-d**), as well as the number of at risk patients for all survival curves (**Fig. 2k,i, ExtDataFig. 21-o**). Multivariate analyses were performed for clinical variables, subtype signatures, and, as requested, PurlST (**P2P-Fig. 12e-g**). Overall, these show a significant prognostic power of the JUNB repression signature in the multivariate analysis with clinical parameters (**P2P-Fig. 12e**) and in combination with PurlST (**P2P-Fig. 12f**). In the multivariate analysis with subtype signatures, we did not see significant prognostic power in any of the factors (including PurlST), except for Collisson_CLA (**P2P-Fig. 12g**). This may likely be due to collinearity of these signatures.

Point-to-Point Response Figure 12. a-d, Overall survival, numbers at risk, and hazard ratio in TCGA (a, n=150), QCMG (b, n=96), Puleo (c, n=288) and Zhou (d, n=85) patients stratified by JUNB repression signature score. Top: Kaplan-Meier survival analysis for the lower/upper quartiles and mid group for JUNB repression signature scores. Median survival (ms) is indicated. Log-rank test. Bottom: Cox proportional hazard. Hazard ratio (to lower quartile) with 95% CI. P values are shown right. **e-g**, Multivariate regression models for JUNB repression signature scores (“JUNB_repressed”) and clinical variables as well as PurIST scores and Notta¹⁵, Moffitt¹⁴, and Collisson¹³ subtype signatures. Hazard ratio with 95% CI is indicates. P values are shown right.

17. The authors performed the siRNA experiment only in CAPAN1. They must expand the panel of cells to evaluate the extent of their conclusions. Additionally, the authors must be consistent in the panel of cells in each experiment. The authors shift from CAPAN1 (siRNA) to CAPAN2 (luciferase and in vivo) depending on the experiment.

In response to the reviewer’s previous comment 12, we want to highlight that we have already conducted JUNB silencing in three different CLA-like PDAC cell lines: CAPAN1, GCDX62, and CAPAN2. Additionally, we have now included JUNB silencing in two more CLA-like cell lines, CFPAC1 and HPAF-II. The following key experiments have been presented and included in the manuscript for these cell lines: i) transwell invasion assay in GCDX62, CFPAC1 and CAPAN2 (**ExtDataFig. 2b-j**), ii) luciferase assay in CFPAC1, HPAF-II and CAPAN2 (**Fig. 3o, ExtDataFig. 3l,m**), iii) WB analysis for cJUN or CCL2 in CAPAN1, GCDX62, CFPAC1 and CAPAN2 cell lines (**Fig. 3m,n, ExtDataFig. 3i-k**), iv) HDAC1 and JUNB co-immunoprecipitation (**Fig. 3g,h**) and cJUN overexpression in CAPAN1 and CAPAN2 cell lines (**Fig. 3p,q**) and v) macrophage (M1 and M2) staining in CAPAN1 (TNF- α -treated; **Fig. 5n,o**) and CAPAN2 (cJUN-overexpressed; **Fig. 4**) derived orthotopic pancreatic tumors.

We appreciate the reviewer's feedback on expanding cell lines, and we believe that the additional new results in expanded CLA-like cell lines strengthen our study.

18. Why did the authors solely focus on HDAC1 and no other HDACs, such as HDAC3, which

main target is H3k27ac. A pathway enrichment analysis is not enough to justify an experimental decision, mainly because this analysis pointed out other potential targets.

We apologize for not providing adequate rationale for focusing on HDAC1 in our study. Our main intention was to identify the JUNB-mediated repressor complex that enhances the mechanistic insight of our study. We observed HDAC target gene signatures that were enriched upon JUNB silencing (Fig. 3f). Multiple studies have shown the importance of Class I HDACs (HDAC1, 2, 3, 8) in pancreatic cancer^{3,4}. In particular, HDAC1 has been highlighted in a comprehensive study as crucial factor in determining PDAC subtype plasticity (ref²⁷, which was already cited in the previous draft). Additionally, we only observed an interaction of JUNB with HDAC1 (Fig. 3g,h) but not with HDAC2 and HDAC3 (P2P-Fig. 13 a,b), which is one of the primary reasons for focusing on HDAC1. We have now edited the text and included an additional citation in the current version of the manuscript (please see page, 9/10; line 261-290).

Additionally, we have now conducted HDAC1 and H3K27ac ChIP-seq after siRNA against JUNB, reported now in Fig. 3k,l, and ExtDataFig. 3e-h. Our findings support the idea that JUNB recruits HDAC1 to the genome, as we observe almost exclusively loss of HDAC1 occupancy upon JUNB silencing. Out of the approximately 12,000 siCtrl-exclusive HDAC1 regions identified, 1454 overlap with JUNB binding regions we previously identified (Fig. 3k, see P2P-Fig. 13c). These overlapping JUNB-HDAC1 regions are enriched for inflammatory TNF- α signaling pathways (Fig. 3l, see P2P-Fig. 13d). Additionally, we found a high number of JUNB binding regions gaining H3K27ac, a mark of activation, on TNF- α signaling pathways upon siJUNB as well (ExtDataFig. 3g,h, see P2P-Fig. 13e,f), which supports the suppression of pro-inflammatory signaling by JUNB-HDAC1 in low-grade/CLA-like cells. We have now edited the text to provide adequate rationale in the current version of the manuscript (please see page 9/10, line 261-290).

Point-to-Point Response Figure 13, Immunoblot for JUNB, HDAC2, HDAC3, and β -actin after JUNB pull-down, IgG isotype control or input in CAPAN1 (a) and CAPAN2 (b). n=3. c,d, ChIP-seq analysis for JUNB in control cells and HDAC1 with siJUNB or siCtrl. c, Overlap of JUNB binding regions and regions where HDAC1 is significantly lost upon siJUNB ("HDAC1_DOWN"). d, GREAT analysis of the overlapping regions of c with $-\log_{10}(P_{adj})$ indicated. Hallmark (H) and curated (C2) signature collections of the Molecular Signature Database (MSigDB) are shown. e, Overlap of JUNB binding regions in

control cells and regions where H3K27ac is significantly gained upon siJUNB (“H3K27ac_UP”). **f**, GREAT analysis of the overlapping region of **e** with $-\log_{10}(P_{adj})$ indicated. Hallmark (H) and curated (C2) signature collections of the Molecular Signature Database (MSigDB) are shown.

19. It is imperative for the authors to clarify how the overexpression of cJUN “overexpressed HA-tagged cJUN (cJUN-OE) and then orthotopically implanted empty vector control (EV) or cJUN-OE cells” generates a CLA tumor (JUNB high) in the *in vivo* experiments. It is unclear in the proposed mechanism.

Firstly, we want to make it clear that the cJUN-OE model does not produce a CLA tumor with high levels of JUNB. This unique model was used to gain insight into how the cJUN-CCL2-driven local inflammatory environment (via TNF- α) disrupts the CLA-neoplastic state (resulting in reduced JUNB and GATA6) and promotes subtype-coexistence. According to our main findings, this diverse model presents a valuable opportunity to explore potential regional differences through both cell-intrinsic and extrinsic mechanisms, which could help explain why we generally observe either a cJUN-high or JUNB-high state, in line with the antithetical effects on inflammation induced by these AP1 factors. Although the cJUN-OE construct induces a general elevation in cJUN, in the *in vivo* analysis we could see that not every single tumor cell is positive for the HA-tagged cJUN, even within individual ducts (see **ExtDataFig. 3a**). This allowed us to perform the shown “hotspot analysis”, investigating areas of high cJUN and high JUNB expression separately.

Indeed, we found that cJUN hotspot areas are surrounded by CCL2+ cells, which, in turn, lead to the recruitment CD68+ and CD163+ M2 macrophages (now included in **Fig. 4i-n**, see **P2P-Fig. 11b-e** above (page, 22) and **P2P-Fig. 14 a-b**).

Point-to-Point Response Figure 14. a,b, Distance analysis based on IF for CCL2 in HA-cJUN-OE tumors. **a**, Distance analysis of CCL2+ cells to JUNB or HA-cJUN hotspots. Scatter plots show each individual cell, with mean \pm s.d. Student’s t-test with Welch’s correction. $n=14925$ CCL2+ cells from $n=5$ tumors, with a total of $n=745208$ cells analyzed. **b**, As in **a**, showing the shortest distances of each cell towards both the HA-cJUN and JUNB hotspots.

20. The authors must describe the histological differences between cJUN overexpressed tumors and the control.

In the original version of the manuscript, we have only noted that there may be an increased immune infiltration visible in the HE (line, 315f). We have now moved this to an earlier point in the results as follows:

“Intriguingly though, not all ductal cells in the HA-cJUN-OE tumors showed cJUN expression (**Extended Data Fig. 4a**). There was not obvious tumor histological differences between the groups, however, we interestingly observed a trend towards

higher immune infiltrations (**Fig. 4b**). This was further supported by an increase in CD45⁺/CD68⁺ and TNF- α ⁺/CD68⁺ macrophages in cJUN-OE CLA-derived tumors (**Extended Data Fig. 3c-e**), in line with the pro-inflammatory effects of cJUN. As JUNB attenuated expression of cJUN (see **Fig. 3k-m**), we next assessed whether cJUN^{low} areas in this heterogeneous tumor model exhibited high JUNB expression.”

21. In figure 4, the authors must clarify the type of macrophage detected.

We thank reviewer for this important comment. The data presented in **ExtDataFig. 4c-e** showed increased CD45⁺/CD68⁺ and CD68⁺/TNF- α ⁺ macrophages in the cJUN-OE PDAC tumors compared to empty vector controls. As mentioned in the response to section 14 above, we have now included IHC for the CD86⁺ M1 macrophages and the CD163⁺ M2 macrophages in our hotspot analysis (now included in **Fig. 4i-l**, see **P2P-Fig. 11b-e**), as we have previously shown for CD68⁺ macrophages (**Fig. 4g,h**). Consistent with the general association of pro-tumor effects with M2-like CD168⁺ macrophages, we observed that these macrophages are attracted to cJUN hotspot regions (average distance 359.7 μ m) and generally slightly further away from JUNB hotspots (403.5 μ m). Conversely, the M1-like CD86⁺ macrophages, which are known to exhibit anti-tumor effects, show the opposite trend, being found closer to JUNB hotspots (459.9 μ m) and further away from cJUN areas (606.8 μ m). Additionally, we also examined the infiltration of CD163⁺ cells in TNF- α -treated CAPAN1-derived orthotopic tumors, where we had observed significantly reduced protein expression of GATA6, JUNB, and E-cadherin (**Fig. 5h-l**). Upon TNF- α treatment, there were strong infiltrations of CD163⁺ macrophages (now included in **Fig. 5n,o**, see **P2P-Fig. 11f-g**). These results highlight the negative impact of M2-like TNF- α ⁺ macrophages in promoting the BL inflammatory state.

22. In figure 5d, the authors seem to have inverted TNF- α and VC for the GSEA of CLA related gene sets.

The labeling for TNF- α and VC in **Fig. 5d** is not inverted. The running enrichment score is negative, indicating that the shown gene sets of CLA subtype signatures are enriched in the VC samples in the comparison TNF- α vs VC. We apologize if this was not easily visible in the graph. Therefore, we have now added vertical lines indicating an enrichment score of 0 in all enrichment plots, so that the direction of the enrichment is more clearly visible.

23. The authors used TNF- α to modulate PDAC phenotype. They found that TNF- α induces EMT and basal-like phenotype. These observations have already been published (doi: 10.1158/0008-5472.CAN-07-5704). Lastly, they must define the type of mice used in each experiment in the main text. Additionally, it is crucial to illustrate how these results are related to JUNB results and macrophage recruitment.

We agree with the reviewer's assessment that this study²⁴ has shown a general involvement of TNF- α in EMT progression in PDAC cells. However, TNF- α had not yet been described as driver of co-existence of subtypes. Additionally, the influence of inflammatory TNF- α ⁺ macrophages in subtype plasticity and co-existence were unclear. Furthermore, the relationship between the dichotomous JUN/AP1 expression in neoplastic cells and its association with subtype plasticity, immune recruitment, and influence on TNF- α signaling at a spatial level had not been described in the literature before. We here presented the novel role of TNF- α in a unique regulatory network in PDAC subtype co-existence. In this context, it is important to note that we have

furthermore clearly identified inflammatory CD68+ macrophages as the source of TNF- α in the TME, not neoplastic cells as reported in the previous study²⁴. In addition, our data shows that exogenous TNF- α negatively regulates CLA lineage-specific marker expression (GATA6, JUNB, and E-cadherin) both in orthotopic tumors and in cell lines, which may enforce subtype mixture by destabilizing CLA-like phenotypic state and thus promote BL aggressive phenotypic state (please see **Fig. 5j-m**). On the mechanistic level, our findings suggest that in a CLA-like state, loss of JUNB induces the cJUN-CCL2 feed-forward loop, which, in turn, recruits TNF α + macrophages that enforce destabilization of the epithelial state and thus promote a BL inflammatory program. This study highlights the role of intrinsic JUNB signaling in counteracting pro-inflammatory processes and inhibiting macrophage infiltration, TNF- α signaling, and inducing T cell recruitment, which ultimately translates to better patient survival. Together, our findings substantially increase our understanding of the cross-talk and interconnectedness of intrinsic subtype identity in neoplastic tumor cells and extrinsic signaling (such as TNF- α + macrophages) cascades in the TME, with prognostic and therapeutic implications for PDAC.

As requested by the reviewer, the details of the mice strain are provided in the figure legends and methods section and are also depicted in the figures.

24. The authors must include in the anti-TNF α + Gem experiments, the anti-TNF α and GEM alone controls to evaluate the actual synergy of the combinatory treatment.

Thank you for raising this important point. In our study of the syngeneic KPCbl6 orthotopic model, we decided to investigate the effects of combining anti-TNF- α with gemcitabine therapy, as well as a vehicle control (VC) group. This decision was based on several factors. Most importantly, findings from previous and unpublished experiments that indicated no significant survival benefits from either anti-TNF- α ⁵ or gemcitabine (unpublished, see **P2P-Fig. 15**) monotherapy in the KPC syngeneic model. These findings are consistent with a recent study²⁵, where authors reported that anti-TNF- α monotherapy (ms=25d) or gemcitabine + paclitaxel chemotherapy (ms=25d) compared to vehicle control group (ms=24d) in a KPCbl6 orthotopic model did not improve survival, which might partially due to the high aggressiveness of these models where mice die as early as 3 weeks after tumor implantation. In sum, we were required to follow animal welfare regulations to minimize unnecessary suffering for animals in groups that previously showed no effect. In addition, we reasoned that clinical trials would require a chemo-backbone so that this experimental setup bears additional preclinical value.

We have already included the following sentences in our results section (page 14; line 432-436), along with appropriate references:

*“Anti-TNF- α monotherapy is not effective in aggressive PDAC⁵. Similarly, gemcitabine (GEM) chemotherapy alone or in combination with paclitaxel is essentially ineffective in *Kras*^{G12D};*p53*^{R172H};*Pdx1-Cre* (KPC)-derived murine PDAC models^{25,26}. Thus, we tested whether combination of gemcitabine (GEM) with TNF- α inhibition may enhance treatment response.”*

Point-to-Point Response Figure 15. KPC cells were orthotopically implanted into syngeneic C57BL6/J mice and treated with gemcitabine (Gem) chemotherapy, or vehicle control. Kaplan-Meier survival analysis with median survival is indicated. Log-rank test shows no significant (ns) difference.

Reviewer #3 (Remarks to the Author): Expert in pancreatic cancer functional genomics, therapy, preclinical models, and epigenetics; co-reviewed with Reviewer #2

Reviewer #4 (Remarks to the Author): Expert in cancer multi-omics, bioinformatics, and pancreatic cancer

Review Report: Spatial Tumor Immune Heterogeneity Facilitates Subtype Co-existence and Therapy Response via AP1 Dichotomy in Pancreatic Cancer

Summary:

This manuscript investigates the role of AP1 transcription factor dichotomy (JUNB vs. cJUN) in driving intratumoral heterogeneity in pancreatic ductal adenocarcinoma (PDAC). The study combines preclinical models, clinical data, and multi-omics analyses to explore how AP1 factors, alongside CD68+ macrophages, influence tumour subtype co-existence and response to therapy. The findings suggest that JUNB/AP1- and HDAC-mediated epigenetic programs suppress pro-inflammatory immune signatures, favouring a therapy-responsive classical subtype, while CD68+/TNF- α + macrophages promote a basal-like PDAC phenotype.

Having reviewed the paper three times, I appreciate the authors' thorough analysis and report on the AP1 dichotomy's role in driving intratumoral heterogeneity in PDAC. Each review has deepened my understanding of their comprehensive work. While I have comments and suggestions, I believe they are more suited for future research directions. The current study stands as a complete and substantial contribution in its field, and it would be inappropriate (borderline insulting) for me to recommend additional experiments for this already comprehensive study.

We truly appreciate the reviewer's positive and gracious comments and are thankful for their helpful suggestions, all of which have been incorporated into the revised manuscript as detailed below.

Major Comments:

1. Comparative Analysis: The work would significantly benefit from expanding its comparative analysis to include the role of AP1 in relation to other pivotal transcription factors and epigenetic mechanisms known to influence intratumoural heterogeneity in PDAC, such as NF- κ B, STAT3, HIF-1 α , EZH2, and DNMTs. This broader examination would provide valuable context, situating the study within the extensive landscape of cancer biology. Could the authors discuss on this to offer readers a more comprehensive understanding of the intricate regulatory networks driving PDAC heterogeneity and potentially unveil new therapeutic targets or biomarkers.

We appreciate the reviewer's valuable suggestion. The important role of epigenetic modulators and their associated TFs in PDAC heterogeneity has been extensively studied by Lomberk and colleagues²⁷. This study suggested the dynamic role of epigenetic co-regulators in controlling super enhancers of the PDAC subtype-specific lineage TFs (e.g. GATA6, ELF3) and its associated neoplastic phenotype. Importantly, DNA methyltransferases, EZH2 (histone methyltransferase), and HDAC1 are highly abundant in the neoplasms and are tightly regulated by both cell-intrinsic (TF-driven) or by microenvironmental signals. This aligns with our study, as we have shown how JUNB recruits HDAC1 to repress BL-proinflammatory gene signatures critical for the low-grade/CLA-like phenotypic state. We have now performed ChIP-seq analysis for HDAC1 and H3K27ac after JUNB silencing which further elucidates this point (now included in **Fig. 3k,l** and **ExtDataFig. 3e-h**, see **P2P-Fig. 16**). We have included the following sentence in the Results and Discussion:

Results:

"TFs rely on additional epigenetic co-regulators to exert their regulatory functions on lineage gene expression. A previous study identified key epigenetic regulators, such as HDAC1, as crucial in determining PDAC subtype heterogeneity²⁷. Particularly, in the gene signatures directly repressed by JUNB genes (see **Fig. 2l, **ExtDataFig. 2K**) and**

in GSEA in JUNB silencing transcriptome data, HDAC target signatures were found to be upregulated (**Fig. 3F**). Therefore, we hypothesized that HDAC1 may be involved in JUNB-mediated transcriptional repression of BL-associated inflammatory lineage signatures.”

Discussion:

“Specifically, the TNF- α -mediated inflammatory response associated with low JUNB expression destabilizes CLA neoplastic identity by promoting a CLA-to-BL transition via epigenetic transcriptional reprogramming in PDAC. Notably, the maintenance of a CLA-like epithelial state or the suppression of pro-inflammatory factors by JUNB requires the epigenetic co-regulator HDAC1. Our study emphasizes the importance of previous research²⁷, which has demonstrated how the interaction of epigenetic co-regulators like HDAC1 with lineage-specific TFs can affect PDAC heterogeneity.”

The reviewer’s suggestion on the therapeutic targets: We do not currently have specific drugs to target AP1 (cJUN or JUNB) TFs, but there are epigenetic inhibitors like HDAC inhibitors available. We think it is important to be cautious when targeting CLA subtype tumors. In the Discussion section, we have included a paragraph about future targeting strategies involving a combination of anti-TNF- α and immunotherapies. Unfortunately, we couldn’t explore these questions in depth due to word count restrictions. We have thoroughly discussed the role of subtype-specific lineage TFs and chromatin regulatory functions in subtype maintenance in another review article²⁸, which we have cited in this manuscript. We apologize to the reviewer for this limitation.

Point-to-Point Response Figure 16. a, Overlap of JUNB binding regions and regions where HDAC1 is significantly lost upon siJUNB (“HDAC1_DOWN”). **b**, GREAT analysis of the overlapping regions of **a** with $-\log_{10}(P_{adj})$ indicated. Hallmark (H) and curated (C2) signature collections of the Molecular Signature Database (MSigDB) are shown. **c**, Overlap of JUNB binding regions in control cells and regions where H3K27ac is significantly gained upon siJUNB (“H3K27ac_UP”). **d**, GREAT analysis of the overlapping region of **c** with $-\log_{10}(P_{adj})$ indicated. Hallmark (H) and curated (C2) signature collections of the Molecular Signature Database (MSigDB) are shown.

2. Enhancement of Methodological Transparency: The integration of diverse data types is commendably robust; however, detailing the selection criteria for clinical samples and ensuring the reproducibility of bioinformatics analyses would substantially strengthen the manuscript. The authors are encouraged to deposit pre-processed data into a recognised repository and share the code used for generating key figures and findings (where necessary). This step would not only bolster the study's credibility but also facilitate future research by allowing others to validate and extend the work presented.

We thank reviewer for their thoughtful suggestion. In response to queries from other reviewers, we have included additional details and clarifications in the Methods section regarding the analysis and overall composition of the clinical and experimental datasets in both patients and murine models. All raw and processed bioinformatics data (such as ATAC-seq, CHIP-seq, and RNA-seq) relevant to this study have been deposited in a repository. Please refer to the Data Availability section for further details. Regarding the code used, we did not utilize any in-house developed analysis tools. All tools employed are publicly available, and their usage has been delineated in the Methods section.

3. Therapeutic Implications: The discussion on potential therapeutic strategies targeting the AP1 dichotomy and macrophage signalling pathways is intriguing. Could the authors expand this discuss on how these findings could be translated into clinical trials or drug development would be valuable.

We thank reviewer for their thoughtful suggestion. This study identified the interactions between neoplastic intrinsic AP1 TFs (JUNB/AP1 and cJUN/AP1) and TNF- α + macrophages in driving the co-existence of subtypes or an aggressive and immunosuppressive PDAC. However, pharmacological targeting of AP1 has not been successful so far. We strongly believe that targeting TNF- α signaling could be an alternative and successful approach in targeting PDAC tumors. We have made the following addition to the Discussion detailing possible translation to clinical application in PDAC:

“In accordance, the PRINCE trial suggests that higher levels of CD4+/8+ T cells are associated with a better response to immune checkpoint inhibition combined with chemotherapy. Elevated TNF- α signaling has a negative effect on therapy response in metastatic PDAC patients, emphasizing its role in immunosuppression. Combining anti-TNF- α therapy with chemotherapy has shown significant benefits in metastatic lung cancer patients²⁹. This approach is now being used with nivolumab and anti-TNF- α therapy in resectable lung cancer patients (NCT04991025). Future studies targeting TNF- α pathways combined with immunotherapies may offer important therapeutic options in PDAC therapy.”

4. Statistical Rigour: Expanding on the “Statistical analysis” and statistical methods used to analyse the data would enhance the manuscript's methodological rigour. This could also include stating the approach utilised for multiple comparison whenever possible.

Thank you for this suggestion. We have further detailed the statistical tests used in the revised Statistical Analysis section, including the built-in correction methods in the bioinformatic tools such as Bonferroni for rGREAT, Benjamini-Hochberg for DESeq2 and clusterProfiler.

Other suggestions
1. Experimental Validation: Include additional experimental validation of the key findings,

particularly the functional roles of JUNB and cJUN in PDAC. This could involve using CRISPR/Cas9 for gene editing to directly assess the impact of these transcription factors on tumor phenotype and immune response.

This is an interesting concept for future experiments. Thus far, we have utilized lentiviral overexpression of JUN/AP1 TFs to examine tumor phenotype and innate immune response. Future studies will involve targeting cJUN or JUNB using CRISPR/Cas9-editing to investigate tumor plasticity and immune response through orthotopic tumors.

2. Patient Diversity: Expand the dataset to include a wider range of PDAC patients, considering variables such as disease stage, treatment history, and genetic background. This diversity could help validate the findings across a broader patient population.

We thank the reviewer for raising this point. We have expanded our analysis to include an additional patient cohort published by Zhou et al.¹ for the JUNB repression signature study. This has allowed us to increase the number of cohorts from three to four and add 97 additional patients (85 with survival data), bringing the total to 652 patients with expression data and 619 with survival information. We have also included data from 32 resected patient tumors for JUNB and GATA6 analysis (UMG cohort), in addition to the existing 105 TMA resected patient datasets (PMCC cohort).

Reviewer #5 (Remarks to the Author): Expert in tumour immune microenvironment, immunogenomics, immunology, macrophages, and pancreatic cancer; co-reviewed with Reviewer #6

In this study, Klein et al. investigate the mechanisms underlying intratumoral subtype heterogeneity in pancreatic ductal adenocarcinoma. Combining different data on human PDAC specimens, they identify the dichotomy of AP1 transcription factors and macrophage regionalization as neoplastic cell-intrinsic and -extrinsic drivers of PDAC subtype heterogeneity respectively.

PDAC is a highly heterogeneous disease. A comprehensive description of intratumoral heterogeneity of PDAC subtypes and its impact on clinical outcome is still lacking, thus this paper is relevant to the field.

The manuscript is well-written and combines valuable -omics data to explore the role of AP1 TFs and macrophages in shaping PDAC subtype heterogeneity. Unfortunately, some major concerns need to be addressed prior to recommending this study for publication, as there are key findings poorly supported by the data.

We thank the reviewer for their positive and gracious comments. In our revised manuscript, we have included new datasets to address the reviewer's in-depth analysis of the AP1/HDAC1-driven pro-inflammatory program and macrophage functions in PDAC subtype plasticity. This additional information provides a more comprehensive understanding as outlined below.

Major comments

1. The authors associate JUNB with classical subtype identity in PDAC patients. However, clear and significant data proving expression of JUNB in classical-like neoplastic cells are missing:

a. The authors focus their attention on proving the co-expression of JUNB and GATA6 as proxy of CL subtype, and in some analyses, the correlation between GATA6 and JUNB is not very evident (Extended figure 1a). GSEA using CL signatures strengthens the findings, but the author should also report GSEA using BL subtype signatures.

We apologize for not clarifying this effectively. Our intent was not to imply that JUNB is a classical subtype identity marker/proxy. Instead, our current data and prior studies support the view that JUNB act as a key regulator of a CLA-like epithelial differentiation state^{5,6,7}, largely restricted to low-grade/CLA-like epithelial neoplastic states. We have now included additional data sets, orthotopic tumors, and the expression and functions of JUNB in previous (GCDX62, CAPAN1, and CAPAN2) as well as newly added low-grade/CLA-like cell lines (CFPAC1 and HPAFII), which further strengthen JUNB's function in maintaining an epithelial neoplastic state. As pointed out by the reviewer, we observed downregulation of BL subtype genes such as TP63 and MYC upon JUNB silencing (please see **Supplementary Table 3**), but we did not observe a significant negative correlation between JUNB expression and BL subtype signatures using GSEA.

While the correlation of JUNB and GATA6 in epithelial cells is consistently strong across multiple data sets, indeed, in bulk tumor data (**ExtDataFig. 1a-c** in the previous manuscript), it is not significant. This is likely due to expression of JUNB or GATA6 in non-neoplastic cells. To address both this concern and comments by other reviewers, we are now presenting only epithelial-specific JUNB/GATA6 expression data in **Fig. 1**. We have now expanded cohort of chemotherapy-naive, resected PDAC patients, where we investigated the co-expression of JUNB and GATA6 in the neoplastic epithelial compartment (panCK+) in both human PDAC patients and orthotopic-derived CLA or BL tumors. We have now included these new data sets from the UMG PDAC cohort (n=32) to demonstrate the co-expression of JUNB and GATA6 in single, epithelial (panCK+)

cells using triple-IF with panCK, JUNB, and GATA6 in the corresponding sections (data now presented in **Fig.1e,f** and **ExtDataFig. 1a**, see **P2P-Fig. 17a,b**). In addition, new analyses in low-grade/CLA-like and BL tumors derived from pancreatic orthotopic tumors also showed that only the low-grade/CLA-like tumors show JUNB:GATA6 double-positive cells (specifically in the epithelial (panCK+) compartment; now included in **ExtDataFig. 1b,c**, see **P2P-Fig. 17c,d**). These data demonstrates a strong correlation between the expression of JUNB and GATA6 double-positive in neoplastic compartments. In addition, we do see a significant association of JUNB and GATA6 in PDAC microdissected epithelia (**Fig. 1n**), flow-cytometry-sorted epithelial (EPCAM+) cells (**Fig. 1c**), manually confirmed epithelial (CK19+) regions of TMAs (**Fig. 1g-m**).

Altogether, we have decided to exclude the bulk RNA expression data and instead expand on the connection between epithelial-specific JUNB and GATA6 expression. Additionally, we have refined the text to incorporate the term "CLA-like epithelial state" or "phenotypic state" in place of "CLA identity" within relevant sentences. We apologize for any inconvenience caused to the reviewers.

Point-to-Point Response Figure 17. a, IF for JUNB, GATA6, and pan-cytokeratin (panCK) in resection tissue of therapy-naive PDAC patients at representative region with high, intermediate, and low epithelial JUNB expression in the University Medical Center Göttingen (UMG) cohort. Epithelial area is overlaid on greyscale images in magenta, based on panCK⁺ cell classification by QuPath. In the overlay, blue: DAPI, green: JUNB, magenta: panCK, yellow: GATA6. Scale bar 50 μ m. **b**, Quantification of **a** for JUNB⁺ and GATA6:JUNB double-positive epithelial (panCK⁺) cells. Linear regression with 95% CI, as well as Spearman's *R* and associated *P* value. *n*=32. **c**, As in **a**, for orthotopically transplanted CAPAN1 and MiaPaCa2 cells into NMRI-*Foxn1*^{nu/nu} mice. Scale bar: overview, 200 μ m; insert, 50 μ m. **d**, Quantification of **c** for per-animal average percentage of GATA6:JUNB⁺ epithelial (panCK⁺) cells with mean \pm s.d. shown. CAPAN1, *n*=7; MiaPaCa2, *n*=8. Student's t-test with Welch's correction.

(cont.) Moreover, GSEA of JUNB bound regions using regions associated with classical/basal-like genes or using as gene sets the acetylated regions specific for low/high grade PDAC cells (Diaferia) could confirm that JUNB preferentially binds to regions associated with aggressive phenotype.

Regarding the association of JUNB-bound regions and CLA/BL genes, we do see a near-significant enrichment of Moffitt and Collisson CLA signatures, but likewise the Notta BL-B signatures, indicating that it may function both through activation of CLA genes and repression of BL-B (**P2P-Fig. 18**). With a much higher significance though, JUNB binds to hallmark genes of the TNF- α signaling pathway, underlining the importance of JUNB in repression of inflammatory pathways that in turn promote BL identity.

Point-to-Point Response Figure 18. a, GREAT analysis in JUNB-bound sites by CHIP-seq in CAPAN1 cells published previously⁵. Hallmark signatures of the Molecular Signature Database, as well as Notta¹⁵, Moffitt¹⁴, and Collisson¹³ subtype signatures are indicated. Negative $\log_{10} q$ values are indicated.

(cont.) Finally, the authors should test if JUNB repression signature is expressed at different levels in classical vs basal like cell lines and if its expression anti-correlates with JUNB expression.

We thank the reviewer for this helpful remark. Indeed, our original JUNB repression signature of 146 genes, although utilizing genes that are upregulated upon JUNB silencing in CAPAN1 cells, did not exhibit anti-correlation with JUNB itself in expression data of a wider range of 46 PDAC cell lines of the Cancer Cell Line Encyclopedia (CCLE), as many of the individual genes correlate with JUNB as well. Therefore, we have refined the signature through this analysis to define a more broadly applicable JUNB repression signature by only including those genes that were negatively associated with JUNB in the CCLE data (now included in **Fig. 2j**, see **P2P-Fig. 19a**). All analysis was redone using this refined 37 gene signature, which does negatively correlate with JUNB expression in CCLE (**P2P-Fig. 19b**). This refined JUNB repression signature yielded largely the same results as the original one, such as enrichment of TNF- α signaling pathways (now included in **ExtDataFig. 2k**, see **P2P-Fig. 19c**), survival benefit of low signature expressing patients (updated and additional individual cohorts in **Fig. 2k,l**, **ExtDataFig. 2l-o**, see **P2P-Fig. 19d**), and diverging TME composition (updated in **Fig. 5g**, **ExtDataFig. 5b**), including in additional CIBERSORTx analysis (now included in **Fig. 5h**, **ExtDataFig. 5c**, see **P2P-Fig. 19e**).

Point-to-Point Response Figure 19. **a**, Spearman correlation of genes of significantly upregulated, JUNB-bound genes following JUNB silencing in CAPAN1 with JUNB in 46 PDAC cell lines of the Cancer Cell Line Encyclopedia (CCLE). Negatively associated genes (red) form the refined *JUNB repression signature*. **b**, Correlation analysis for JUNB and Gene Set Variance Analysis (GSVA) scores of the *JUNB repression signature* in 46 PDAC cell lines of the CCLE. Linear regression with 95% CI, as well as Spearman's R and associated P value. **c**, Gene ontology analysis of the *JUNB repression signature* with $-\log_{10}(q\text{-value})$ indicated. Hallmark (H) and curated (C2) signature collections of the Molecular Signature Database (MSigDB) are shown. **d**, Overall survival, numbers at risk, and hazard ratio in TCGA ($n=150$), Puleo ($n=288$), QCMG ($n=96$), and Zhou ($n=85$) patients stratified by *JUNB repression signature* score. Top: Kaplan-Meier survival analysis for the lower/upper quartiles ($n=155$ each) and mid group ($n=309$) for *JUNB repression signature* scores. Median survival (ms) is indicated. Log-rank test. Bottom: Cox proportional hazard. Hazard ratio (to lower quartile) with 95% CI. P values are shown right. **e**, CIBERSORTx deconvolution percentages for the indicated lineages in 652 patients of the TCGA, QCMG, Puleo, and Zhou cohort, separated into quartiles based on the *JUNB repression signature* score. Mean \pm s.d. shown.

b. The IHC data are important to quantify the intratumoral co-existence of JUNB-high and -low area. However it is unclear why the other patients were excluded from the correlation analysis.

We apologize for the confusion about the data presented in **Fig 1b-d** of the original manuscript. We have now revised the Methods section to clarify the approach (**please see page 20, line 572-615**). Actually, all patients were included in the observations. We found that 32.4% of patients exhibited intratumoral co-existence of both JUNB-high and JUNB-low regions. This led us to consider the average across all cores per patient, as

well as individual JUNB-high and JUNB-low cores. Yet, all patients, regardless of heterogeneous or homogeneous JUNB expression, were included in the analyses presented in **Fig. 1g-m**.

(cont.) How is the protein expression of GATA6 in patients not showing intratumor heterogeneity and does it confirm the positive correlation with JUNB?

GATA6 also exhibits intratumoral heterogeneity, as do many other markers when compared across multiple cores per patient. This previously led to us to develop a toolset for analytically accessing spatial heterogeneity captured in standard TMA setups³⁰. And, our study shows that the intratumor heterogeneity of GATA6 and JUNB are associated, as demonstrated in **Fig. 1j**, where regions within the same tumor with high JUNB levels exhibit higher GATA6 expression compared to regions with low JUNB levels.

(cont.) The authors analyzed 105 PDAC patients for expression of JUNB and GATA6 in malignant cells. However, it is not clear how this analysis was computed. Could the authors perform double staining to check co-expression of JUNB-GATA6 in the annotated tumor cells?

Thank you for prompting this clarification. In our study of 105 patients, we stained JUNB and GATA6 on proximal sections and analyzed GATA6 expression in JUNB-high versus JUNB-low cores. Specifically, we looked at cores with top vs bottom 33% of epithelial JUNB expression levels, which we quantified using pixel-based detection of the IHC colorimetric product. We have now included the following detailed description of the stain quantification methods in the Methods section:

“IHC staining was quantified using QuPath bioimage analysis software. Individual TMA cores were registered, and patient identifiers were superimposed for analysis. Malignant epithelial regions were manually annotated for JUNB, Cytokeratin 19, CDH1 and GATA6 expression (see also Extended Data Fig 1d), while the Simple Tissue Detection tool was used to select tissue for TNF- α expression analysis. Residual normal pancreas epithelium, nerves, large blood vessels, tissue folds and stain artifact were manually excluded. Pixel-based detection parameters were set and optimized for each stain. ‘Positive pixel percentage’ within the annotated epithelial regions was determined as the fraction of positive pixels within all detected pixels and as indicated, either averaged across all cores per patient (patient level). For display and statistical testing, samples were then stratified into groups (‘high’, ‘intermediate’, ‘low’) via the top, intermediate and bottom 33% of the obtained JUNB and TNF- α positive pixel percentage values, respectively.”

As requested by the reviewer, and as mentioned above in response to comment 1a, we examined the co-expression of JUNB and GATA6 in the neoplastic epithelial compartment (panCK+) in chemo-naive PDAC patients (n=32) and in orthotopic-derived PDAC tumors (now included in **Fig. 1e,f, ExtDataFig. 1a-c**, please see **P2P-Fig. 17**). Our results show a strong association of GATA6 and JUNB in the neoplastic compartments (panCK+) of resected PDAC patient tumors as well as in orthotopic-derived low-grade/CLA-like tumors (**Fig. 1e,f, ExtDataFig. 1a-c**, please see **P2P-Fig. 17**).

c. The expression of JUNB in non-tumor cells may compromise the results. As shown in fig 1p the majority of JUNB-expressing cells seems to be composed by non-malignant cells. In lines 168-169 the authors say that they queried transcriptome and proteome of PDAC samples epithelium- or stroma-enriched by laser-capture microdissection, however in Fig. 11 it is unclear which data are correlated. The correlation between JUNB and GATA6 is reported for both epithelium- and stroma-enriched samples together? Is it possible to perform correlation on

epithelium- or stroma-enriched samples separately to see if correlation is occurring only in the epithelial region?

We thank the reviewer for bringing this to our attention. We had indeed presented the combination of both epithelium and stroma and apologize for the oversight. We then restricted the analysis in the LCM-enriched data to epithelial samples (n=30) and do no longer observe a significant correlation between GATA6 and JUNB protein (**P2P-Fig. 20**). This contrasts with the otherwise strong correlations we observed by IHC (patient data from 105 TMA, **Fig. 1g-m**), multiplex-IF (panCK+/GATA6+/JUNB+, in n=32 UMG patient data, **Fig. 1e,f** and **ExtDataFig. 1a**) and orthotopic tumors (**ExtDataFig. 1b,c**), FACS-sorted epithelial-specific transcriptome datasets (n=31) (**Fig. 1c**). We furthermore do see a significant association in the LCM-RNA-seq data in a larger cohort of patients (**Fig. 1n**) and a noticeable trend in the RNA expression of the cohort (**P2P-Fig. 20, right**). From this we altogether conclude that variation and sample size (n=30) in the epithelial LCM/proteome analysis may mask the correlation which otherwise presents consistently across multiple cohorts. We have decided to exclude both protein and RNA data in order to enhance the clarity and flow of the manuscript. Thank you again for raising this important point.

Point-to-Point Response Figure 20. Correlation analysis of RNA (*left*) and protein (*right*) expression for JUNB and GATA6 in laser-capture microdissection-enriched human PDAC tumor epithelia. Linear regression, as well as Spearman's *R* and associated *P* value.

2. The authors claim that JUNB is a key molecule for maintaining CL subtype, but it would be interesting to further analyze JUNB expression and co-expression with GATA6 in PDAC patients at different stages and upon different treatments. This is potentially very interesting:
a. Is JUNB expression changing in the different stages of PDAC? Fig. 1O contains data from PDAC patients stage I-IV.

We appreciate the reviewer's concerns. It is indeed an intriguing question whether the expression of JUNB and GATA6 changes at different stages of the disease and during therapy. In keeping with additional new analyses addressing how JUNB impacts the overall clinical outcomes in PDAC patients (response to Reviewer 2), we now also analyzed the relationship of JUNB repression signature with tumor stage and furthermore tumor grade utilizing three public RNA-seq cohorts. These analysis indicate an increase in Stage I patients with JUNB expression in the upper quartile in their

respective cohorts (lower quartile 3.57%, mid 8.77%, upper 13.95% Stage I; **P2P-Fig. 21a**). To address this, we suggested the use of the JUNB repression signature, which should be more robust to stromal influence by quantifying a larger set of genes instead of a single factor. With the JUNB repression score, there is a stage-dependent increase in the JUNB repression scores (**ExtDataFig. 2b**, see **P2P-Fig. 21b**).

We think that to study the co-expression of GATA6 and JUNB through multiplex-IF in neoplastic cells (panCK+/GATA6+/JUNB+) during the progression of the disease or in response to therapy, we would need larger sets of patient data specific to each stage, including samples from resected tumors and advanced stage tumors as well as from tumors that have been treated with therapy. The JUNB:GATA6 expression data presented in our study primarily originates from patients who underwent R0 resection. Thus, the limited number of patients did not allow us to assess stage-dependent expression of JUNB/GATA6/panCK+ in the neoplastic epithelial cells (panCK+).

Point-to-Point Response Figure 21. **a**, AJCC stages in TCGA, QCMG, and Zhou cohorts combined, stratified by lower quartile (lowerQ), upper quartile (upperQ) and mid JUNB expression. **b**, JUNB repression signature scores in AJCC stages in TCGA, QCMG, and Zhou cohorts combined.

b. How do therapies impact on JUNB expression, CL/BL state and TNF expression? The conclusions drawn by the authors on the expression of JUNB upon therapies are unclear. After chemo or neo-adjuvant therapy the % of tumor cells co-expressing JUNB and GATA6 decreases and the majority of JUNB+ cells are GATA6-negative. Does it imply that JUNB expression dissociate from classical-like phenotype upon chemo or neoadjuvant therapy? Is it possible that GEM induces TNF production thus promoting suppressive TME and limiting its activity?

This is a good point raised by the reviewer. Originally, we hypothesized that therapy-induced inflammatory signals, such as TNF- α , might negatively influence the expression of JUNB and GATA6. This hypothesis was later confirmed in our TNF- α -treated *in vitro* and preclinical models.

We initially aimed to analyze the expression of JUNB and GATA6 in therapy-induced plasticity using a matched chemo-naïve and post-chemotherapy patient cohort. However, obtaining patient tissue that had undergone neo-adjuvant chemotherapy before resection is not standard practice in our centers, making it challenging. Since we did not obtain a large number of matched patient samples as requested by other reviewers, we have decided to remove this data. Instead, we have focused on the GATA6:JUNB double-positivity in chemo-naïve resected PDAC patients. We now expanded the cohort of chemo-naïve resected patients stained for GATA6, JUNB, and panCK in the neoplastic compartment, which revealed a strong association between neoplastic epithelial (panCK+) JUNB and the amount of neoplastic epithelial GATA6:JUNB double-positive cells, as mentioned above (**P2P-Fig. 17 a.b**). This refined focus will provide greater clarity of our proposed study and help elucidate this crucial point.

In addressing our initial questions, a recent study by the Peter Bailey lab¹, published during the submission of our manuscript, compared the specificity of neo-adjuvant therapies (FOLFIRINOX vs. Gemcitabine) in PDAC subtype plasticity and its impact on patient prognosis. This study demonstrated subtype plasticity and the induction of an inflammatory immune microenvironment that largely depended on the choice of neo-adjuvant therapies. These findings emphasize the necessity of a comprehensive investigation into the choice of chemotherapies in subtype plasticity. Therefore, we believe it is crucial to conduct further investigation involving a matched chemo-naïve and post-chemotherapy patient cohort to examine the impact of JUNB and GATA6-mediated epithelial plasticity.

(cont.) It would be of notice a deeper analysis, since in the final part of the paper the authors use Gemcitabine+aTNF to improve mice survival. In this experiment, control groups are missing: mice should be treated with GEM and aTNFa alone.

In our study of the syngeneic KPCbl6 orthotopic model, we decided to investigate the effects of combining anti-TNF- α with gemcitabine therapy, as well as a vehicle control (VC) group. This decision was based on findings from previous and unpublished experiments that indicated no significant survival benefits from either anti-TNF- α ⁵ or gemcitabine (unpublished, see **P2P-Fig. 22**) monotherapy in the KPC syngeneic model. Therefore, we followed animal welfare regulations to minimize unnecessary suffering for animals in groups that previously showed no effect. These findings are consistent with the most recent study²⁵, where authors reported that anti-TNF- α monotherapy (ms=25d) or gemcitabine + paclitaxel chemotherapy (ms=25) compared to vehicle control group (ms=24) in a KPCbl6 orthotopic model did not improve survival.

It is important to note that in syngeneic KPC models, animals died as soon as 2 weeks of tumor implantation. Syngeneic KPC tumors are associated with the development of fibrotic stroma and dense extracellular matrix, which likely impedes the delivery of anti-TNF- α monotherapy or gemcitabine therapy to the tumor site. Therefore, it is conceivable that a combination treatment could yield better results. We have already included the following sentences in our results section, along with appropriate references:

“Anti-TNF- α monotherapy is not effective in aggressive PDAC⁵. Similarly, gemcitabine (GEM) chemotherapy alone or in combination with paclitaxel is essentially ineffective in *Kras*^{G12D};*p53*^{R172H};*Pdx1-Cre* (KPC)-derived murine PDAC models^{25,26}. Thus, we tested whether combination of gemcitabine (GEM) with TNF- α inhibition may enhance treatment response.”

Point-to-Point Response Figure 22. KPC cells were orthotopically implanted into syngeneic C57BL/6/J mice and treated with gemcitabine (Gem) chemotherapy, or vehicle control. Kaplan-Meier survival analysis with median survival is indicated. Log-rank test shows no significant (ns) difference.

3. The authors claim that JUNB-mediated transcriptional repression of inflammatory BL-specific lineage signature is mediated by HDAC1. The ChIP-qPCR and IP experiments supporting this finding are performed on a restricted number of inflammatory genes, which are not enough to conclude that there is HDAC-mediated transcriptional repression. Genome-wide analysis on HDAC occupancy should be performed to assess concomitant binding of JUNB and HDAC1 in inflammatory genes, in presence and absence of JUNB to confirm the transcriptional repression of inflammatory genes by HDAC-mediated deacetylation.

We thank reviewer for their thoughtful comment. We have previously restricted the analysis to a select number of target genes to prove a possible mechanism of JUNB-mediated repression. As the reviewer points out, a genome-wide analysis by ChIP-seq is needed to elucidate this mechanism in more detail. Hence, we have now performed HDAC1 and H3K27ac ChIP-seq after siRNA against JUNB or control siRNA, now included in **Fig. 3k,l**, and **ExtDataFig. 3e-h**, see **P2P-Fig. 23**. In line with our hypothesis of JUNB recruiting HDAC1 to the genome, we could identify a genome-wide loss of HDAC1 occupancy upon siJUNB. Of the approx. 12,000 siCtrl-exclusive HDAC1 regions identified, 1454 overlap with JUNB binding regions identified by us previously (**Fig. 3k**). These overlapping JUNB-HDAC1 regions highly enrich for TNF- α signaling pathways in a GREAT analysis, including the Hallmark TNF- α signature (**Fig. 3l**). In line, we identified a number of differentially bound regions of H3K27ac after siJUNB. Among these regions with increased H3K27ac, 3589 regions overlapped with JUNB binding regions (**ExtDataFig. 3g**). Again, these overlapping regions highly enriched for TNF- α and other inflammatory signaling pathways as well as EMT (**ExtDataFig. 3g**). These results validate our targeted approach shown previously on the genome-wide level, indicating HDAC1 in the presence of JUNB as a negative regulator of BL pro-inflammatory program.

Point-to-Point Response Figure 23. **a**, Overlap of JUNB binding regions and regions where HDAC1 is significantly lost upon siJUNB (“HDAC1_DOWN”). **b**, GREAT analysis of the overlapping regions of **a** with $-\log_{10}(P_{adj})$ indicated. Hallmark (H) and curated (C2) signature collections of the Molecular Signature Database (MSigDB) are shown. **c**, Overlap of JUNB binding regions in control cells and regions where H3K27ac is significantly gained upon siJUNB (“H3K27ac_UP”). **d**, GREAT analysis of the overlapping region of **c** with $-\log_{10}(P_{adj})$ indicated. Hallmark (H) and curated (C2) signature collections of the Molecular Signature Database (MSigDB) are shown.

(cont.) In addition to that, lines 259-261 referring to Fig.3C-E should be clarified. The authors say that all the reported genes are repressed since they display strong JUNB binding in absence of H3K27ac. However, it looks like there is some binding of JUNB in regions where H3K27ac is present near cJUN gene and to a lesser extent near IL1A gene.

As the reviewer correctly points out, there are regions where JUNB is also binding to acetylated regions in the genome and may activate genes. Based on our integrated ChIP- and RNA-seq data (**Fig. 2g-i**), as well as functionally luciferase reporter (**Fig. 3o**, **ExtDataFig. 3l,m**) and immunoblot data (**Fig. 3m,n**, **ExtDataFig. 3i-k**), we believe that JUNB predominantly exerts a repressive effect on genes associated with inflammatory processes. There are also instances where it induces gene expression, possibly in conjunction with other epithelial TF such as GATA6 (see **Fig. 2a-f**). We posit that for cJUN expression, enhancer-mediated activation of the gene is pivotal, as we have previously observed⁵. Although the promoter of cJUN exhibits H3K27 acetylation, the enhancer region where JUNB binding occurs does not, resulting in increased expression upon depletion of JUNB. It is likely that the promoter is maintained in a poised state, enabling rapid triggering of cJUN expression in response to specific stimuli, such as TNF- α .

4. The authors link high JUNB expression to reduced macrophage recruitment. The reduced macrophage infiltration is an important point in the manuscript, and it should be addressed more in depth. The data reported examine the correlation between JUNB expression and markers of monocytic lineage, relying on bulk RNA-seq scores. However, this analysis does not consider the fact that JUNB is expressed by multiple cell types and that even within tumor cells JUNB expression is heterogeneous (a key point of the study is that subtypes with different JUNB/cJUN expression co-exist within the same tumor). Other analyses would be more suitable to prove this finding, for example reanalysis of single-cell RNA-seq data, IF or IHC stainings. Additionally, in the data presented in Fig.5G, in patients stratified based on the JUNB repression signature score no striking difference is observed in the relative MCPcounter for the Monocytic lineage.

The reviewer raises a valid point regarding the need for a more detailed examination of the specific types of macrophages affected by the AP1 TF dichotomy. We now performed CIBERSORTx analysis of the aggregated patient data from all cohorts in relation to the refined JUNB repression signature. This analysis has uncovered an increase in M2 macrophages and a decrease in naïve and memory B cells and CD8+ T cells in patients with higher JUNB repression signature scores (now included in **Fig. 5h**, see **P2P-Fig. 24a**). This suggests that loss of JUNB-mediated repression encourages a macrophage-inflamed environment and T and B cell exclusion. Notably, recent studies have emphasized the potential association of M2 macrophages with T cell exclusion, poor prognosis, and their correlation to the BL subtype state in PDAC patients^{1,31,32}. Interestingly, the refined JUNB repression signature also led to the finding of increased monocytic lineage scores in the MCPcounter analysis (updated data in **Fig. 5g**, see **P2P-Fig. 24b**), indicating that the improvement in the signature removed irrelevant genes that may have masked this effect.

On the experimental front, we have now conducted an IHC analysis of CD86 (M1) and CD163 (M2) macrophage infiltration in the JUNB+ hotspot area in orthotopic cJUN-overexpressing tumors (now included in **Fig. 4i-i**, see **P2P-Fig. 24c-f**). Our findings revealed that CD168+ M2 macrophages are attracted to cJUN hotspot regions (359.7 μm), while being generally slightly further away from JUNB hotspots (403.5 μm). Conversely, M1-like CD86+ macrophages, known for exhibiting anti-tumor effects³², were found closer to JUNB hotspots (459.9 μm) and further away from cJUN areas (606.8 μm).

In addition, we have now investigated the infiltration of CD163+ macrophages in TNF- α -treated CLA-derived orthotopic tumors, where we previously observed a significant reduction in protein expression of GATA6, JUNB, and E-cadherin (**Fig. 5i-m**). In these tumors, we noted a substantial infiltration of CD163+ M2 macrophages upon TNF- α treatment (now included in **Fig. 5n,o**, see **P2P-Fig. 24g,h**), emphasizing the impact of M2 macrophages in promoting the BL inflammatory state. This observation was additionally corroborated by CIBERSORTx analysis in the corresponding virtually microdissected stromal expression profile (now included in **ExtDataFig. 5a**, see **P2P-Fig. 24i**). We have cited all the mentioned references in the revised manuscript.

Point-to-Point Response Figure 24. **a**, CIBERSORTx deconvolution percentages for the indicated lineages in 652 patients of the TCGA, QCMG, Puleo, and Zhou cohort, separated into quartiles based on the JUNB repression signature score. Mean \pm s.d. shown. **b**, As in **a**, for relative MCPcounter scores. MCPcounter scores were min-max normalized and standardized to the mean of the lower JUNB repression signature score group for merging of the different cohorts. Mean \pm s.d. shown. **c-f**, Distance

analysis based on IHC for CD163 and CD86 in HA-cJUN-OE tumors. **c,e**, Distance analysis of CD163⁺ (**c**) and CD86⁺ (**e**) cells to JUNB or HA-cJUN hotspots. Scatter plots show each individual cell, with mean \pm s.d. Student's t-test with Welch's correction. **c**, n=7691 CD163⁺ cells from n=4 tumors, with a total of n=654297 cells analyzed. **e**, n=8384 CD86⁺ cells from n=4 tumors, with a total of n=632792 cells analyzed. **d,f**, As in **c,e**, showing the shortest distances of each cell towards both the HA-cJUN and JUNB hotspots. **g**, CD163 IHC staining in orthotopically transplanted CAPAN1 tumors treated with TNF- α or VC. Arrows indicate positive cells. Scale bar: overview, 100 μ m; insert, 25 μ m. **h**, Quantification of **g** for per-animal percentage of CD163⁺ cells with mean \pm s.d. shown. n=7. Student's t-test with Welch's correction. **i**, CIBERSORTx analysis for M2 macrophages in stromal compartment of virtually microdissected RNA-seq data of orthotopically transplanted CAPAN1 tumors treated with TNF- α or VC for three weeks. VC, n=2; TNF- α , n=3. Student's t-test with Welch's correction.

5. The authors claim that TNF α + macrophage promote subtype coexistence by destabilizing CLA identity and promoting BL state. However major points need to be addressed: a. TNF α + macrophages are not shown. There is no identification of macrophages expressing TNF but only correlative analysis of macrophage content and TNFhigh regions. A double positive staining for CD68 and TNF in human and an intracellular FC analysis of macrophages expressing TNF in mice should be performed. This is particularly important since in figure 6D ctrl tumors show that the majority of TNF+ cells are CD45-.

We appreciate the reviewer's concerns regarding the presence of TNF- α + macrophages in our model. The data presented in the manuscript in **ExtData Fig. 4c-e** shows a strong increase in TNF- α +/CD68+ double-positive macrophages in the cJUN-OE PDAC tumors. Additionally, we have now performed additional IF staining for CCL2 included in the cJUN/JUNB hotspot analysis in the same model (now included in **Fig.4m,n**, see **P2P-Fig. 25a,b**). Our new findings suggest that CCL2+ cells, potentially recruiting CD68+/TNF- α + macrophages, are significantly closer to cJUN hotspots (310.2 μ m) compared to JUNB hotspots (516.4 μ m). Using the cJUN/JUNB hotspot analysis, we have further confirmed that the CD168+ M2 macrophages are attracted to cJUN hotspot regions (359.7 μ m), while being generally slightly further away from JUNB hotspots (403.5 μ m) in orthotopic cJUN-overexpressing tumors (now included in **Fig. 4i-l**, see **P2P-Fig. 24c-f**).

Upon reviewing the suggestion from the reviewer to conduct FC analysis, we realize that they meant to refer to Fig. 7d instead of Fig. 6d, which showcases TMA tissues. The reviewer's suggestion to perform FC analysis in KPCbl6 tumors (in **Fig.7d**) is valuable. However, due to technical limitations, all of our orthotopic tumors and human PDAC specimens are frozen, which prevents us from conducting flow cytometry analysis. Nevertheless, through single-cell RNA and IHC analysis, we and others have demonstrated that CD68 macrophages exclusively express TNF- α in PDAC tumors^{5,35}.

We have chosen for most analyses to show a correlation between regional TNF- α expression and macrophages, as TNF- α can be secreted into the microenvironment and bind to receptors on other cells than the macrophages to induces TNF- α signaling, such as in the neoplastic epithelial cells. This explains why we see a majority of cells in e.g. **Fig. 7d** not showing double-positivity with CD68. Nonetheless we have quantified CD45/TNF- α double-positive cells (**Fig. 7f**).

Point-to-Point Response Figure 25. a,b, Distance analysis based on IF for CCL2 in HA-cJUN-OE tumors. **a,** Distance analysis of CCL2⁺ cells to JUNB or HA-cJUN hotspots. Scatter plots show each individual cell, with mean \pm s.d. Student's t-test with Welch's correction. $n=14925$ CCL2⁺ cells from $n=5$ tumors, with a total of $n=745208$ cells analyzed. **b,** As in **a,** showing the shortest distances of each cell towards both the HA-cJUN and JUNB hotspots.

b. How do cJUN areas recruit macrophages? Since JUNB silencing increase CK recruiting myeloid cells, are these cJUN areas producing higher levels of CCL2? Are cJUN OE cells producing higher levels of CCL2 than empty ones?

The reviewer correctly points out that cJUN transcriptionally regulates the production of CCL2 in PDAC cells. CCL2 then recruits macrophages in the tumor microenvironment (TME), as demonstrated in our previous study⁵. As mentioned above, we have now included an IF analysis of CCL2 in the in orthotopic cJUN-overexpressing tumors (see **P2P-Fig. 25**). Our results revealed that CCL2⁺ cells are significantly closer to cJUN hotspots (310.2 μ m) in comparison to JUNB hotspots (516.4 μ m) (please see **P2P-Fig. 25a,b**).

c. Does TNF induce basal like genes? In figure 5D the authors could perform GSEA using also BL signatures.

TNF- α treatment indeed induces expression of BL genes. In our study, we have demonstrated that KRT14, TP63, VIM, and KRT5, representative markers of the BL identity are upregulated following TNF- α treatment in CLA CAPAN1 cells in vitro (**Fig. 5a**). However, in the *in vivo* TNF- α experiment where we show reduced CLA signatures in the tumor cell transcriptome (**Fig. 5d**), BL signatures did not show statistically significant changes.

6. TNFa+ macrophages mark reactive stroma with low T cells

a. Figure 5F NMRI-nu/nu mice are immunodeficient mice lacking T cells, nevertheless the authors show a decrease recruitment of T cells in tumor-bearing mice treated with Tnf. Can you explain?

The athymic NMRI^{nu/nu} mice have a significant deficiency in T cells, but they do have a small number of mature T cells that can become cytotoxic when activated by IL-2, as previously reported^{33,34}. However, we anticipate a very low presence of T cells. In **Fig. 5f**, the analysis is based on signature expression normalized across samples as z-scores. Therefore, these changes may indicate minor differences in the original tumor. CIBERSORTx analysis of the same data did not show any significant difference in any of the T cell populations, although there seems to be a decreasing trend in CD8 T cells (**P2P-Fig. 26**). However, in this model, the only population that exhibited a significant change in response to TNF- α treatment *in vivo* were M2 macrophages, which aligns with

the overall data presented in this study, as mentioned above (**P2P-Fig. 24i**). Additionally, in immunocompetent KPC tumors, a recent study has revealed that TNF- α -mediated signaling via macrophages leads to a substantial reduction in CD8+ T cell infiltration in the PDAC TME³⁵. These findings are consistent with the results we observed in human TMA as well as in KPCbl6-derived tumors. We thank the reviewer for pointing this out. We have included the citation in the manuscript text.

Point-to-Point Response Figure 26. CIBERSORTx analysis for CD8 T cells in stromal compartment of virtually microdissected RNA-seq data of orthotopically transplanted CAPAN1 tumors treated with TNF- α or VC for three weeks. VC, n=2; TNF- α , n=3. Student's t-test with Welch's correction.

(cont.) Moreover, this is in contrast with the increased expression of Cxcl9 and Cxcl10, key cytokines for recruitment of CD8+ T cells, in cancer cells upon JUNB silencing (Fig. 3A)

We acknowledge the concerns raised by the reviewers. The role of CXCL9/10 in PDAC is intricate and not fully understood. Our data in **Ext. Data. Fig. 3d** and **Fig. 2b** shows a significant enrichment of IFN- γ response signatures as well as upregulation of CXCL9/10 upon JUNB silencing. Importantly, the chemokines CXCL9/10 are associated with IFN- α/γ signaling events, which have been linked to the squamous/BL PDAC subtype (ref²⁰; citation already included in the manuscript). Moreover, our analysis has demonstrated higher expression levels of both CXCL9 and CXCL10 in high-grade tumors in the TCGA and QCMG cohorts, with CXCL10 being particularly enriched in squamous (i.e., basal-like) tumors (**P2P-Fig. 27**). In line with our observations, Gao and Cheng et al.³⁶ demonstrated that CXCL9 suppresses cytotoxic T cell function in PDAC. Nie et al.³⁷ showed that CXCL10 is associated with Tregs and negatively associated with CD8+ T cells. Further data showed that CXCL10 is linked to Tregs and poor survival³⁸. Therefore, we believe that both CXCL9 and CXCL10 may promote PDAC aggressiveness. However, it remains an open area for future research to determine whether and how JUNB or TNF- α -mediated transcriptional regulation of CXCL9/10 contributes to immunosuppression in PDAC.

Point-to-Point Response Figure 27. a-f, Expression of CXCL9 (a,c,e) and CXCL10 (b,d,f) stratified by tumor grade in TCGA (a,b) and QCMG (c,d), as well as PDAC subtypes in QCMG (e,f). Data was accessed and subtype calls used from R2 Genomics platform (<https://hgserver1.amc.nl/>). Student's t-test with Welch's correction.

7. Which is the effect of TNF in tumor vs stroma/immune cells?

The effects of TNF- α on tumor and stromal/immune cells are a very interesting subject of study. Our research primarily focuses on how TNF- α affects tumor cells and the JUN/AP1 dichotomy, and its associated changes in the immune microenvironment. We showed that treating CLA-like tumor cells with exogenous TNF- α destabilizes their epithelial differentiation state by reducing their lineage-specific transcription regulators. Additionally, TNF- α induces a cJUN-CCL2 feed-forward loop, leading to the recruitment of TNF- α + macrophages that promote an immunosuppressive program. Our analysis suggests that TNF- α affects macrophage polarization and leads to an increase in M2-like macrophages. In line with our findings, a recent study found that TNF- α secreted by macrophages suppresses the expression of IL-33 in epithelial neoplastic cells, which is critical in recruiting CD103+ dendritic cells and eliciting a cytotoxic CD8+ T cell response in PDAC³⁵. Importantly, consistent with our findings, they show that targeting TNF- α in macrophages restores cytotoxic T cell response in metastatic PDAC³⁵. Additional signature analysis in the stroma-specific compartment showed an enrichment in EMT-related genes upon TNF- α treatment. This may indicate an elevated presence of cancer-associated fibroblasts, as these are highly expressing EMT genes³⁹. This is consistent with a previous study⁴⁰, as well as the MCPcounter data shown in **ExtDataFig. 5b**. Together, these studies support the conclusion that TNF- α is involved in immunosuppression in PDAC.

Minor comments:

- In lines 203-204 the authors state that JUNB binds on a potential downstream enhancer of GATA6. However, in Fig.2B the peak of JUNB binding does not have the marker of active enhancers H3K27ac. How do they explain this?

We apologize for the lack of clarity in **Fig. 2b**. Previous studies have shown that enrichment of H3K4me1 strongly correlates with enhancer regions, while H3K4me3 enrichment is associated with promoter regions or transcription start site (TSS)²¹. We consider the region downstream of the GATA6 gene body to be a potential enhancer, as it is marked by H3K27ac as well as, importantly, a peak for H3K4me1, as shown in **P2P-Fig. 28**. H3K4me3, a marker of TSS regions, is not binding at the enhancer region. Therefore, we only characterize this as a "potential" enhancer, as we currently lack direct evidence of this region looping to the promoter of GATA6, or any functional validation of enhancer activity.

Point-to-Point Response Figure 28. Representative ChIP-seq tracks for JUNB published previously (Tu et al) as well as publicly available data for H3K27ac, H3K4me3, and H3K4me1 (Diaferia) for the GATA6 locus.

- Unclear references: line 164 (Extended data Figure 1f-h); line 254 (Fig.2p,q)

We apologize for the oversights. The lines should have read **Figure 1f-h** and **Fig. 2k**, respectively, in the previous manuscript version. In the revised manuscript, both pieces of data have changed position, but their reference in the text has been corrected.

- Increase dimension of characters for p-values in graphs and always explicit the comparisons the statistical analyses are referred to

As requested, we have increased the font sizes where necessary. The figures comparing two groups indicate which groups are referred to in the analysis.

- The authors should clarify in the methods how they defined the TNF-high, -int and -low regions. This should be clearer.

We have made adjustments to the Methods section regarding the analysis of the IHC patient TMA analysis to ensure clarity.

Reviewer #6 (Remarks to the Author): Expert in tumour immune microenvironment, immunogenomics, and bioinformatics; co-reviewed with Reviewer #5

REFERENCES

1. Zhou, X. *et al.* Persister cell phenotypes contribute to poor patient outcomes after neoadjuvant chemotherapy in PDAC. *Nat Cancer* **4**, 1362–1381 (2023).
2. Shinke, G. *et al.* Role of histone deacetylase 1 in distant metastasis of pancreatic ductal cancer. *Cancer Sci* **109**, 2520–2531 (2018).
3. Cai, M.-H. *et al.* Depletion of HDAC1, 7 and 8 by Histone Deacetylase Inhibition Confers Elimination of Pancreatic Cancer Stem Cells in Combination with Gemcitabine. *Sci Rep* **8**, 1621 (2018).
4. Roca, M. S. *et al.* HDAC class I inhibitor domatinostat sensitizes pancreatic cancer to chemotherapy by targeting cancer stem cell compartment via FOXM1 modulation. *J Exp Clin Cancer Res* **41**, 83 (2022).
5. Tu, M. *et al.* TNF- α -producing macrophages determine subtype identity and prognosis via AP1 enhancer reprogramming in pancreatic cancer. *Nat Cancer* **2**, 1185–1203 (2021).
6. Diaferia, G. R. *et al.* Dissection of transcriptional and *cis* regulatory control of differentiation in human pancreatic cancer. *The EMBO Journal* **35**, 595–617 (2016).
7. Milan, M. *et al.* FOXA2 controls the cis-regulatory networks of pancreatic cancer cells in a differentiation grade-specific manner. *EMBO J* **38**, e102161 (2019).
8. Milan, M., Diaferia, G. R. & Natoli, G. Tumor cell heterogeneity and its transcriptional bases in pancreatic cancer: a tale of two cell types and their many variants. *The EMBO Journal* **40**, e107206 (2021).
9. Porter, R. L. *et al.* Epithelial to mesenchymal plasticity and differential response to therapies in pancreatic ductal adenocarcinoma. *Proceedings of the National Academy of Sciences* **116**, 26835–26845 (2019).
10. Krebs, N. *et al.* Axon guidance receptor ROBO3 modulates subtype identity and prognosis via AXL-associated inflammatory network in pancreatic cancer. *JCI Insight* **7**, e154475 (2022).
11. The Cancer Cell Line Encyclopedia Consortium & The Genomics of Drug Sensitivity in Cancer Consortium. Pharmacogenomic agreement between two cancer cell line data sets. *Nature* **528**, 84–87 (2015).
12. Nusinow, D. P. *et al.* Quantitative Proteomics of the Cancer Cell Line Encyclopedia. *Cell* **180**, 387–402.e16 (2020).
13. Collisson, E. A. *et al.* Subtypes of pancreatic ductal adenocarcinoma and their differing responses to therapy. *Nat Med* **17**, 500–503 (2011).
14. Moffitt, R. A. *et al.* Virtual microdissection identifies distinct tumor- and stroma-specific subtypes of pancreatic ductal adenocarcinoma. *Nat Genet* **47**, 1168–1178 (2015).
15. Chan-Seng-Yue, M. *et al.* Transcription phenotypes of pancreatic cancer are driven by genomic events during tumor evolution. *Nat Genet* **52**, 231–240 (2020).
16. Bailey, P. *et al.* Genomic analyses identify molecular subtypes of pancreatic cancer. *Nature* **531**, 47–52 (2016).
17. Somerville, T. D. *et al.* Squamous trans-differentiation of pancreatic cancer cells promotes stromal inflammation. *Elife* **9**, e53381 (2020).
18. Hamdan, F. H. & Johnsen, S. A. DeltaNp63-dependent super enhancers define molecular identity in pancreatic cancer by an interconnected transcription factor network. *Proc. Natl. Acad. Sci. U.S.A.* **115**, (2018).
19. Tonelli, C. *et al.* A mucus production programme promotes classical pancreatic ductal adenocarcinoma. *Gut* **73**, 941–954 (2024).
20. Espinet, E. *et al.* Aggressive PDACs Show Hypomethylation of Repetitive Elements and the Execution of an Intrinsic IFN Program Linked to a Ductal Cell of Origin. *Cancer Discov* **11**, 638–659 (2021).
21. Sharifi-Zarchi, A. *et al.* DNA methylation regulates discrimination of enhancers from promoters through a H3K4me1-H3K4me3 seesaw mechanism. *BMC Genomics* **18**, 964 (2017).

-
22. Nicolle, R. *et al.* Prognostic Biomarkers in Pancreatic Cancer: Avoiding Errata When Using the TCGA Dataset. *Cancers* **11**, 126 (2019).
 23. Gu, Z. & Hübschmann, D. *rGREAT*: an R/bioconductor package for functional enrichment on genomic regions. *Bioinformatics* **39**, btac745 (2023).
 24. Egberts, J.-H. *et al.* Anti-Tumor Necrosis Factor Therapy Inhibits Pancreatic Tumor Growth and Metastasis. *Cancer Research* **68**, 1443–1450 (2008).
 25. Bianchi, A. *et al.* Cell-Autonomous Cxcl1 Sustains Tolerogenic Circuitries and Stromal Inflammation via Neutrophil-Derived TNF in Pancreatic Cancer. *Cancer Discovery* **13**, 1428–1453 (2023).
 26. Mazur, P. K. *et al.* Combined inhibition of BET family proteins and histone deacetylases as a potential epigenetics-based therapy for pancreatic ductal adenocarcinoma. *Nat Med* **21**, 1163–1171 (2015).
 27. Lomberk, G. *et al.* Distinct epigenetic landscapes underlie the pathobiology of pancreatic cancer subtypes. *Nat Commun* **9**, 1978 (2018).
 28. Espinet, E., Klein, L., Puré, E. & Singh, S. K. Mechanisms of PDAC subtype heterogeneity and therapy response. *Trends in Cancer* **8**, 1060–1071 (2022).
 29. Paik, P. K. *et al.* Phase I trial of the TNF- α inhibitor certolizumab plus chemotherapy in stage IV lung adenocarcinomas. *Nat Commun* **13**, 6095 (2022).
 30. Aliar, K. *et al.* Hourglass, a rapid analysis framework for heterogeneous bioimaging data, identifies sex disparity in IL -6/ STAT3 -associated immune phenotypes in pancreatic cancer. *The Journal of Pathology* **261**, 413–426 (2023).
 31. Quaranta, V. *et al.* Macrophage-Derived Granulin Drives Resistance to Immune Checkpoint Inhibition in Metastatic Pancreatic Cancer. *Cancer Res* **78**, 4253–4269 (2018).
 32. Oh, K. *et al.* Coordinated single-cell tumor microenvironment dynamics reinforce pancreatic cancer subtype. *Nat Commun* **14**, 5226 (2023).
 33. Hünig, T. & Bevan, M. J. Specificity of cytotoxic T cells from athymic mice. *The Journal of experimental medicine* **152**, 688–702 (1980).
 34. ENVIGO. Athymic Nuce Mice.
 35. Dixit, A. *et al.* Targeting TNF- α -producing macrophages activates antitumor immunity in pancreatic cancer via IL-33 signaling. *JCI Insight* **7**, e153242 (2022).
 36. Gao, H.-F. *et al.* CXCL9 chemokine promotes the progression of human pancreatic adenocarcinoma through STAT3-dependent cytotoxic T lymphocyte suppression. *Aging* **12**, 502–517 (2020).
 37. Nie, Y., Liu, C., Liu, Q. & Zhu, X. CXCL10 is a prognostic marker for pancreatic adenocarcinoma and tumor microenvironment remodeling. *BMC Cancer* **23**, 150 (2023).
 38. Lunardi, S. *et al.* IP-10/CXCL10 induction in human pancreatic cancer stroma influences lymphocytes recruitment and correlates with poor survival. *Oncotarget* **5**, 11064–11080 (2014).
 39. Szabo, P. M. *et al.* Cancer-associated fibroblasts are the main contributors to epithelial-to-mesenchymal signatures in the tumor microenvironment. *Sci Rep* **13**, 3051 (2023).
 40. Adjuto-Saccone, M. *et al.* TNF- α induces endothelial–mesenchymal transition promoting stromal development of pancreatic adenocarcinoma. *Cell Death Dis* **12**, 649 (2021).
 41. Grünwald, B. T. *et al.* Spatially confined sub-tumor microenvironments in pancreatic cancer. *Cell* **184**, 5577-5592.e18 (2021).

REVIEWERS' COMMENTS

Reviewer #1 (Remarks to the Author):

I appreciate the Authors' effort to address points raised by myself and the other reviewers. There is no further comment.

Reviewer #2 (Remarks to the Author):

After reviewing the resubmitted manuscript, I am pleased to confirm that the authors have fully addressed my previous concerns and questions. Their responses were clear and effective in resolving the issues I raised, particularly with regard to the manuscript's coherence and focus.

The authors have made significant improvements in the organization and clarity of the paper. By refining the study's focus and highlighting the novel insights it provides, they have strengthened the narrative and rationale behind their findings. This has greatly improved the overall coherence of the manuscript, with a clearer emphasis on how the study builds on existing research while offering new mechanistic insights into intratumoral subtype co-existence and immune diversity in pancreatic ductal adenocarcinoma (PDAC).

The additional datasets and revisions introduced by the authors also contribute to the manuscript's scientific rigor. The consistency between their in vitro and in vivo models and the conclusions drawn is now much clearer. The focus on AP1 signaling and its role in tumor immune heterogeneity is now framed more effectively, with well-supported implications for therapy response.

Reviewer #4 (Remarks to the Author):

The authors have thoroughly and satisfactorily addressed all of my comments, providing clear and well-considered responses to each suggestion and concern raised during the review process.

Reviewer #5 (Remarks to the Author):

The revisions made by the authors have significantly enhanced the manuscript. By incorporating a substantial amount of additional data, they have improved the readability of the text and strengthened the overall message. However, one point that remains unclear is the exclusivity of TNF production by macrophages. While the authors now demonstrate an increase in TNF+ CD68+ cells in cJUN OE tumors, it is important to note that macrophages are not the sole producers of TNF in PDAC. Multiple studies have confirmed that neutrophils, T cells, and even cancer cells can also produce TNF (PMID: 36946782; PMID:

37914939). The authors should take care not to attribute the entire responsibility of TNF production to macrophages, and acknowledge the role of other cell types.

Reviewer #6 (Remarks to the Author):

REVIEWER COMMENTS

Reviewer #5 (Remarks to the Author):

The revisions made by the authors have significantly enhanced the manuscript. By incorporating a substantial amount of additional data, they have improved the readability of the text and strengthened the overall message. However, one point that remains unclear is the exclusivity of TNF production by macrophages. While the authors now demonstrate an increase in TNF+ CD68+ cells in cJUN OE tumors, it is important to note that macrophages are not the sole producers of TNF in PDAC. Multiple studies have confirmed that neutrophils, T cells, and even cancer cells can also produce TNF (PMID: 36946782; PMID: 37914939). The authors should take care not to attribute the entire responsibility of TNF production to macrophages, and acknowledge the role of other cell types.

We appreciate the reviewer's suggestions. As pointed out, other cell types may also secrete TNF- α . However, as demonstrated by us and others previously (Tu, Klein et al., Nature Cancer 2021; Dixit et al., JCI Insight 2022), macrophages are the main source of TNF- α in PDAC tumors. The aim of this study was not to elucidate the source of TNF- α , but rather to analyze the effects of TNF- α in subtype co-existence and plasticity. Our experiments confirm that these TNF- α + macrophages contribute to the subtype plasticity and disease aggressiveness. Nonetheless, we have expanded the following sentence in the introduction (lines 100-103) to acknowledge other cell types as potential sources of TNF- α , including citations to the two studies mentioned by the reviewer.

“Furthermore, TNF- α , secreted by macrophages, and other cell types, along with signaling events mediated by IFN- α/γ , can drive the BL subtype-specific transcriptional program and promote PDAC aggressiveness”